# Neural correlates and determinants of approach–avoidance conflict in the prelimbic prefrontal cortex

Jose A Fernandez-Leon[1][†][‡], Douglas S Engelke[1][†], Guillermo Aquino-Miranda[1][†], Alexandria Goodson[1,2], Maria N Rasheed[1], Fabricio H Do Monte[1,2]*

[1]Department of Neurobiology and Anatomy, The University of Texas Health Science Center, Houston, United States; [2]Rice University, Houston, United States

**Abstract** The recollection of environmental cues associated with threat or reward allows animals to select the most appropriate behavioral responses. Neurons in the prelimbic (PL) cortex respond to both threat- and reward-associated cues. However, it remains unknown whether PL regulates threat-avoidance vs. reward-approaching responses when an animals' decision depends on previously associated memories. Using a conflict model in which male Long–Evans rats retrieve memories of shock- and food-paired cues, we observed two distinct phenotypes during conflict: (1) rats that continued to press a lever for food (*Pressers*) and (2) rats that exhibited a complete suppression in food seeking (*Non-pressers*). Single-unit recordings revealed that increased risk-taking behavior in *Pressers* is associated with persistent food-cue responses in PL, and reduced spontaneous activity in PL glutamatergic (PL[GLUT]) neurons during conflict. Activating PL[GLUT] neurons in *Pressers* attenuated food-seeking responses in a neutral context, whereas inhibiting PL[GLUT] neurons in *Non-pressers* reduced defensive responses and increased food approaching during conflict. Our results establish a causal role for PL[GLUT] neurons in mediating individual variability in memory-based risky decision-making by regulating threat-avoidance vs. reward-approach behaviors.

*For correspondence:
fabricio.h.domonte@uth.tmc.edu

[†]These authors contributed equally to this work

**Present address:** [‡]Exact Sciences Faculty, CIFICEN (UNCPBA-CONICET-CICPBA) & INTIA (UNCPBA-CICPBA), Tandil, Argentina

**Competing interest:** The authors declare that no competing interests exist.

## Editor's evaluation

This paper offers a novel behavioural perspective showing how opposing motivational states interact to influence behaviour differentially across individuals. It uses a variety of cutting-edge tools to dissect the microcircuits of the prefrontal cortex. This report is novel, timely, and important. It will be of broad interest to neuroscientists studying fear, reward, motivation, and decision making and is relevant to understanding neural processes in stress and anxiety-related disorders.

## Introduction

The brain's ability to identify and discriminate cues associated with threat or reward allows organisms to respond appropriately to changes in the environment (*Schultz, 2015*; *Hu, 2016*). Animals respond to threatening cues with a series of defensive behaviors including avoidance responses that decrease their chances of being exposed to aversive outcomes (*McNaughton and Corr, 2014*; *Krypotos et al., 2015*; *Cain, 2019*). In contrast, reward cues have attractive and motivational properties that elicit approach behavior (*Robinson and Flagel, 2009*; *Morales and Berridge, 2020*). When animals are exposed to threat and reward cues simultaneously, an approach–avoidance conflict emerges, and decision-making processes are recruited to resolve the situation (*Kirlic et al., 2017*; *Barker et al., 2019*). While many studies have investigated the neural mechanisms that control threat-avoidance and reward-approach independently of each other, it is unclear how the brain uses previously learned

information to regulate the opposing behavioral drives of avoiding threats and seeking rewards during a conflict situation.

Neurons in the prelimbic (PL) subregion of the medial prefrontal cortex (mPFC) change their firing rates in response to cues that predict either threat or reward (*Baeg et al., 2001*; *Burgos-Robles et al., 2009*; *Burgos-Robles et al., 2013*; *Moorman and Aston-Jones, 2015*; *Dejean et al., 2016*; *Otis et al., 2017*). Accordingly, activity in PL neurons is necessary for the retrieval of both food- and threat-associated memories (*Sierra-Mercado et al., 2010*; *Courtin et al., 2014*; *Sangha et al., 2014*; *Do-Monte et al., 2015*; *Otis et al., 2017*). PL neurons are reciprocally connected with the basolateral nucleus of the amygdala (BLA) (*McDonald, 1991*; *Vertes, 2004*), a region implicated in the detection of threats or rewards (*Amir et al., 2015*; *Namburi et al., 2015*; *Beyeler et al., 2016*; *Zhang et al., 2020*). During a risky foraging task in rats, dynamic modifications in the activity of PL and BLA neurons correlate with the detection of imminent threats and the defensive readiness for action (*Kim et al., 2018*; *Kyriazi et al., 2020*). In addition, during a modified Pavlovian cue discrimination task involving footshocks as punishment, increased activity in the BLA–PL pathway is sufficient and necessary for the expression of freezing responses (*Burgos-Robles et al., 2017*), a passive form of defensive behavior. Conversely, inhibitory signaling in PL neurons correlates with threat-avoidance (*Diehl et al., 2018*), an active form of defensive behavior. While these studies suggest a potential role of PL during motivational conflict involving states of certainty (i.e., imminent threats), it is unknown whether changes in PL activity underlie the behavioral variability in approach–avoidance responses under states of uncertainty, when animal's decision depends entirely on the retrieval of previously associated memories. It is also unclear whether PL activity is necessary to coordinate appropriate behavioral responses during conflict, and if so, which subtypes of PL neurons govern the competing demands of approaching rewards vs. avoiding potential threats.

To address these questions, we designed an approach–avoidance conflict test that assesses the ability of rats to remember cues previously associated with either food or footshocks to make a behavioral decision. Using a combination of optogenetics and single-unit recordings, we investigated rats' individual variability in reward seeking and defensive responses during the conflict test and correlated their behaviors (e.g., freezing, avoidance, and risk-assessment) with the firing rate of photoidentified glutamatergic and GABAergic neurons in PL. We then examined the role of PL neurons in risky decision-making by optogenetically manipulating PL activity with high temporal resolution and cell-type specificity during the conflict test.

## Results

### Rats show individual variability in reward-seeking and defensive responses during the approach–avoidance conflict test

To investigate the motivational conflict between approaching rewards and avoiding potential threats, we established a behavioral model in which rats need to balance food seeking with conditioned defensive responses based on their memories of previously acquired cues. Food-restricted rats (18 g of chow per day) were initially placed in an operant box and trained to press a lever for sucrose in the presence of audiovisual cues that signaled the availability of food. Each lever press during the audiovisual cue presentation resulted in the delivery of a sucrose pellet into a nearby dish (see Methods for details). When rats reached 50% of discrimination during cued food seeking, they began lever pressing for sucrose preferentially during the audiovisual cues (*Figure 1—figure supplement 1A, B*). During the habituation day, rats were placed in an odor arena and familiarized with the food cues and the neutral odor amyl acetate (see Methods for details). Next, to pair the odor cue with an aversive stimulus, rats were exposed to an olfactory threat conditioning training (day 1). Animals were placed in an operant box (conditioning box; *Figure 1A*, left) previously connected to an olfactometer and habituated to one odor presentation (amyl acetate, 30 s) without footshock, followed by five odor presentations of the same odor that coterminated with an electrical footshock (0.7 mA, 1-s duration, 270–390-s intertrial intervals, *Figure 1A*, far-left). Food cues (30-s duration) were presented during the odor intervals to assess how threat conditioning alters lever-press responses. Rats showed robust defensive responses during the threat conditioning training, as evidenced by an increase in freezing (Shapiro–Wilk normality test, p < 0.001, Friedman test, Friedman statistic = 84.08, p < 0.001; Dunn's post hoc, p < 0.001) during the conditioned odor presentation (*Figure 1B*), and a decrease in lever

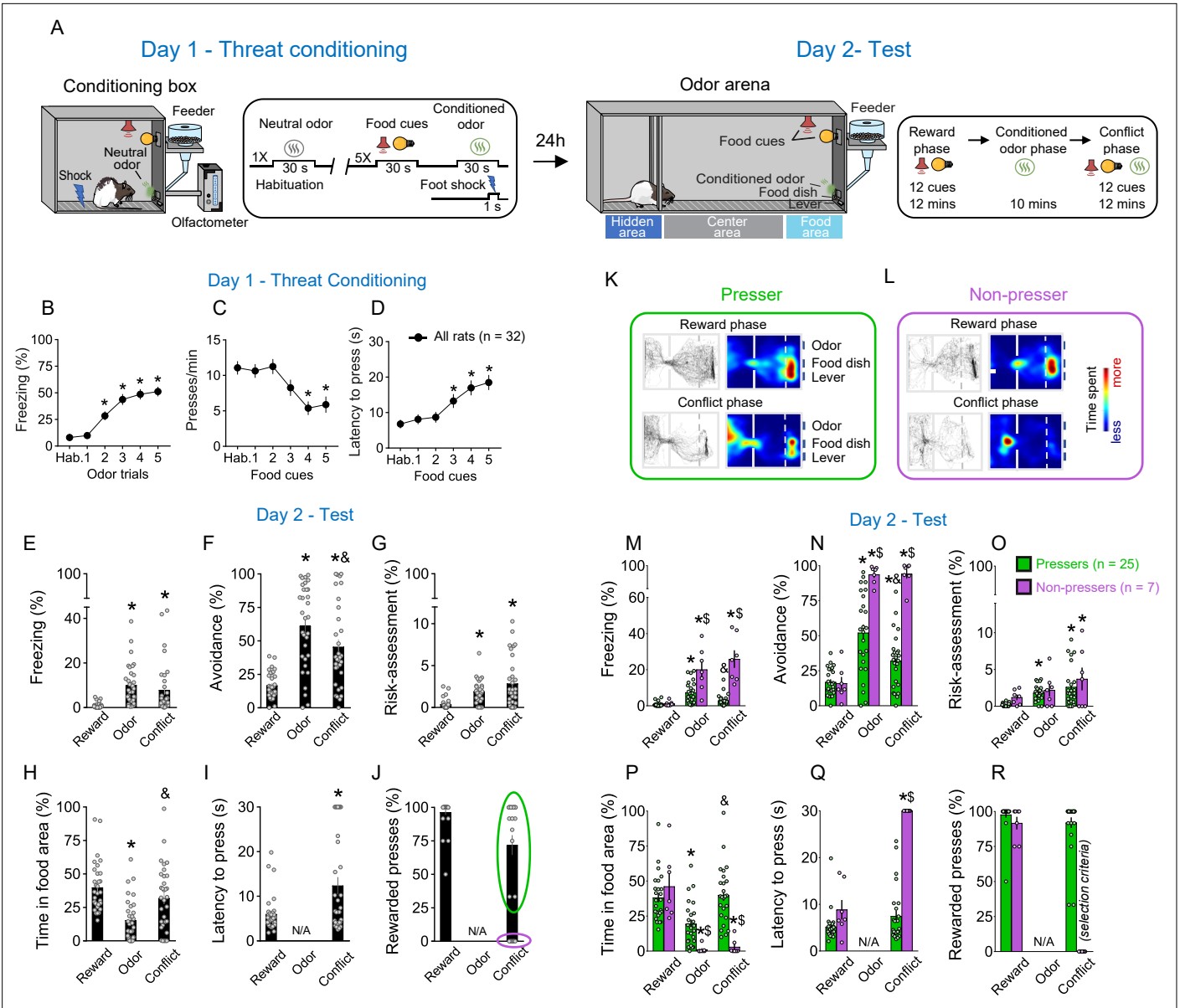

**Figure 1.** Rats show individual variability in reward-seeking responses during an approach–avoidance conflict test. (**A**) Schematic and timeline of the approach–avoidance conflict test. (**B–D**) Rats exhibited an increase in the percentage of time freezing (Shapiro–Wilk normality test, p < 0.001, Friedman test, Friedman statistic = 84.08, p < 0.001, Dunn's post hoc, p < 0.001) and a reduction in lever presses (Shapiro–Wilk normality test, p < 0.001, Friedman test, Friedman statistic = 35.11, p < 0.001, Dunn's post hoc, p < 0.001) with a higher latency to press the lever (Shapiro–Wilk normality test, p < 0.001, Friedman test, Friedman statistic = 29.45, p < 0.001, Dunn's post hoc, p < 0.001) during the olfactory threat conditioning session on day 1 (n = 32), when compared to before the shock. (**E–G**) Patterns of defensive responses and food seeking during the different phases (reward, odor, and conflict) of the test session on day 2. Rats showed an increase in defensive responses characterized by an augment in the percentage of time exhibiting (**E**) freezing (Shapiro–Wilk normality test, p < 0.05, Friedman test, Friedman statistic = 40.46, p < 0.001, Dunn's post hoc, p < 0.001), (**F**) avoidance (Shapiro–Wilk normality test, p < 0.05, Friedman test, Friedman statistic = 31.67, p < 0.001, Dunn's post hoc, p < 0.001), and (**G**) risk-assessment (Shapiro–Wilk normality test, p < 0.05, Friedman test, Friedman statistic = 29.86, p < 0.001, Dunn's post hoc, p < 0.001); and a decrease in the (**H**) percentage of time spent in the food area (Shapiro–Wilk normality test, p < 0.05, Friedman test, Friedman statistic = 32.19, p < 0.001, Dunn's post hoc, p < 0.001) during the odor presentation, when compared to the reward phase. Rats' defensive responses were significantly attenuated during the conflict phase as evidenced by a reduction in the percentage of time. (**F**) Avoiding the odor (p = 0.0031) and an increase in the percentage of time (**H**) approaching the food area (p < 0.001), when compared to the odor phase. (**I, J**) Two different behavioral phenotypes emerged during the conflict phase: rats that continued to press the lever (*Pressers*, green circle, n = 25) and rats that showed a complete suppression in lever pressing (*Non-pressers*, purple circle, n = 7). Rewarded presses were calculated as the percentage of the 12 cue trials in which rats pressed the lever. Representative tracks and heatmaps of time spent in each compartment of the arena for a (**K**) *Presser* or a (**L**) *Non-pressers* rat during the test session. (**M–R**) Patterns of defensive responses and food seeking during the different phases (reward, odor, and conflict) of the test session on day 2 after separating the animals into *Pressers* and *Non-pressers*.

*Figure 1 continued on next page*

*Figure 1 continued*

When compared to *Non-pressers*, *Pressers* showed reduced defensive responses characterized by an attenuation in the percentage of time exhibiting (**M**) freezing ($F_{(2, 60)} = 29.54$, $p < 0.001$) and (**N**) avoidance responses ($F_{(2, 60)} = 23.27$, $p < 0.001$), and an augment in the percentage of time (**P**) approaching the food area ($F_{(2, 60)} = 22.49$, $p < 0.001$) during both the odor and the conflict phases (Bonferroni post hoc test – odor phase, $p = 0.0453$; conflict phase, $p < 0.001$). (**Q**) *Non-pressers* showed increased latency to press the lever during the conflict phase when compared to the reward phase or to *Pressers* in the same phase ($F_{(1, 30)} = 55.14$, $p < 0.001$, Bonferroni post hoc test – all p's < 0.001). (**R**) The percentage of rewarded trials was used as a binary criterium for group classification. Data shown as mean ± standard error of the mean (SEM). One- or two-way analysis of variance (ANOVA) repeated measures followed by Bonferroni post hoc test, all *p's < 0.05 compared to the same group during the reward phase, all &p's < 0.05 compared to the same group during the odor phase, all $p's < 0.05 compared to *Pressers* during the same phase. All statistical analysis details are presented in *Source data 1*. See also *Figure 1—figure supplement 1* and *Video 1*.

The online version of this article includes the following source data and figure supplement(s) for figure 1:

**Source data 1.** Source data for *Figure 1*.

**Figure supplement 1.** *Pressers* and *Non-pressers* showed similar behavioral responses during cued food-seeking training and olfactory threat conditioning.

**Figure supplement 1—source data 1.** Source data for *Figure 1—figure supplement 1*.

presses (*Figure 1C*, Shapiro–Wilk normality test, $p < 0.001$, Friedman test, Friedman statistic = 35.11, $p < 0.001$; Dunn's post hoc, $p < 0.001$) and an increase in the latency to press the lever (*Figure 1D*, Shapiro–Wilk normality test, $p < 0.001$, Friedman test, Friedman statistic = 29.45, $p < 0.001$; Dunn's post hoc, $p < 0.001$) during the presentation of the food cues across trials. After rats have acquired the reward and threat associations, they were returned to the same odor arena in which they were previously habituated and exposed to a test session (day 2) (*Figure 1A*, right). The test session consisted of three different phases: (1) a *Reward Phase*, in which only the audiovisual cues signaling the availability of food were presented; (2) an *Odor Phase*, in which only the conditioned odor was presented, and (3) a *Conflict phase*, in which both the food cues and the conditioned odor were presented simultaneously (*Figure 1A*, far-right).

During the reward phase, rats spent ~40% of the time in the food area and pressed the lever for food in ~95% of the food-cue trials, without exhibiting significant defensive behaviors (*Figure 1E–J*). Introduction of the shock-paired odor during the odor phase reduced the percentage of time rats spent in the food area to ~15% (Shapiro–Wilk normality test, $p < 0.05$, Friedman test, Friedman statistic = 32.19, $p < 0.001$, Dunn's post hoc, $p < 0.001$) and increased defensive behaviors characterized by freezing (Shapiro–Wilk normality test, $p < 0.05$, Friedman test, Friedman statistic = 40.46, $p < 0.001$, Dunn's post hoc, $p < 0.001$), avoidance (Shapiro–Wilk normality test, $p < 0.05$, Friedman test, Friedman statistic = 31.67, $p < 0.001$, Dunn's post hoc, $p < 0.001$), and risk-assessment responses (Shapiro–Wilk normality test, $p < 0.05$, Friedman test, Friedman statistic = 29.86, $p < 0.001$, Dunn's post hoc, $p < 0.001$; *Figure 1E–H*). These defensive behaviors were attenuated by the introduction of food cues during the conflict phase, as evidenced by a reduction in the percentage of time avoiding the conditioned odor (*Figure 1F*, Dunn's post hoc, $p = 0.0031$) and an increase in the percentage of time approaching the food area (*Figure 1H*, Dunn's post hoc, $p < 0.001$). This indicates that the concomitant presentation of food cues and shock-paired odor induced a behavioral conflict in the animals. Interestingly, when we analyzed the percentage of rewarded presses during the conflict phase (*Figure 1J*), two behavioral phenotypes emerged: (1) rats that continued to press the lever for food in the presence of the threatening odor (*Pressers*, *Figure 1K*) and (2) rats that showed a complete suppression in lever presses in the presence of the threatening odor (*Non-pressers*, *Figure 1L*). We then separated the animals into two different groups based on whether the animals pressed the lever or not during the conflict phase and compared their behaviors during the entire test session

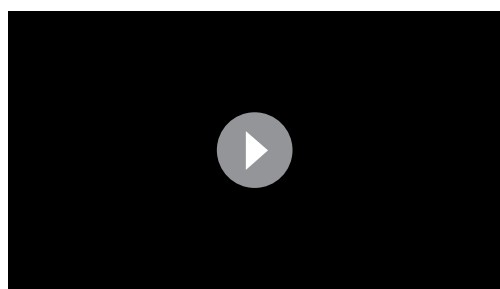

**Video 1.** Presser vs. Non-presser animals during approach–avoidance conflict task. Representative video comparing the different behavioral strategies during the conflict phase. Animals' body were label using DeepLabCut software and dots were used to track the animal position, freezing, and risk assessment.
https://elifesciences.org/articles/74950/figures#video1

(*Figure 1K–R*, *Video 1*). *Pressers* and *Non-pressers* exhibited similar behavioral responses during the reward phase (all p's > 0.05, see , *Source data 1*). However, during the odor and the conflict phases, *Pressers* showed a lower percentage of time exhibiting freezing (*Figure 1M*, two-way repeated measures analysis of variance [ANOVA], interaction – $F(2, 60) = 29.54$, p < 0.001, Bonferroni post hoc test – odor phase, p < 0.001; conflict phase, p < 0.001) and avoidance responses (*Figure 1N*, two-way repeated measures ANOVA, interaction – $F(2, 60) = 23.27$, p < 0.001, Bonferroni post hoc test – odor phase, p < 0.001; conflict phase, p < 0.001), and a greater percentage of time approaching the food area (*Figure 1P*, two-way repeated measures ANOVA, interaction – $F(2, 60) = 22.49$, p < 0.001, Bonferroni post hoc test – odor phase, p = 0.0453; conflict phase, p < 0.001) when compared to *Non-pressers* during both the odor and the conflict phases. A principal component analysis (PCA) showed that PC1 explained most of the variance of the data (>60%), with latency to press the lever and percentage of time in the food area being the two variables that most contributed to PC1 (0.54 and 0.52, respectively, see Methods for details). Because these two variables are directly associated with lever presses, the PCA results support our binary classification of rats into *Pressers* and *Non-pressers* based on whether they pressed the lever or not during the conflict phase.

Subsequent behavioral analyses demonstrated that these two individual phenotypes were not due to prior differences in reward-seeking motivation or odor–shock association because *Pressers* and *Non-pressers* showed similar lever pressing rates during the cued food-seeking training (*Figure 1—figure supplement 1A, B*; two-way repeated measures ANOVA, interaction – $F(1, 55) = 0.1065$, p = 0.7454) and threat conditioning phase (*Figure 1—figure supplement 1C*; two-way repeated measures ANOVA, $F(1, 51) = 0.265$, p = 0.608), as well as the same freezing levels (two-way repeated measures ANOVA, $F(1, 51) = 3.737$, p = 0.058) and maximum speed (two-way repeated measures ANOVA, $F(1, 51) = 6.538e007$, p = 0.999) in response to the shock-paired odor during the threat conditioning phase (*Figure 1—figure supplement 1D, E*). The two phenotypes might not be attributed to prior differences in the relative salience of the odor and the audiovisual cues because *Pressers* and *Non-pressers* spent the same time investigating the odor and exhibited the same response latency to the audiovisual cues during the preconditioning period (*Figure 1—figure supplement 1F, G*; Welch's *t*-test, $t = 0.41$, p = 0.683 and Welch's *t*-test, $t = 0.61$, p = 0.538, respectively). Although *Non-pressers* exhibited higher freezing levels during preodor trials 3 and 4 of the threat conditioning phase (*Figure 1—figure supplement 1H*, $F(5, 250) = 3.038$, p = 0.011, Bonferroni's post hoc p < 0.05), freezing responses before the first food cue and odor presentation were the same during the test day, indicating similar contextual discrimination between the two groups (*Figure 1—figure supplement 1I*, Shapiro–Wilk normality test, p < 0.05, Mann–Whitney test, $U = 248$, p = 0.113). A minute-by-minute analysis during the odor phase demonstrated that behavioral differences in freezing (two-way repeated measures ANOVA, Group, $F(1, 50) = 13.07$, p < 0.001; Interaction, $F(9, 450) = 1.327$, p = 0.220), avoidance (Group, $F(1, 50) = 20.31$, p < 0.001; Interaction, $F(9, 450) = 2.109$, p = 0.027; Bonferroni post hoc min 1 vs. min 10, p > 0.999), and time spent in the food area (Group, $F(1, 50) = 117.5$, p = 0.001; Interaction, $F(9, 450) = 0.573$, p = 0.819) between *Pressers* and *Non-pressers* were already observed in the beginning of the odor phase, and these behaviors remained constant in both groups across the entire duration of the session (*Figure 1—figure supplement 1J–L*), ruling out the possibility that the group differences were caused by extinction of the odor–shock association in *Pressers*.

Together, our results demonstrate that our conflict model is a suitable paradigm to investigate the interactions between reward- and threat-associated memories. Given that rats exhibit individual differences in food seeking and defensive responses during the test session, we next took advantage of the two observed phenotypes to examine the neuronal correlates of risk-taking (*Pressers*) and risk-avoiding (*Non-pressers*) behaviors in PL neurons.

## PL neurons respond differently to reward cues in *Pressers* vs. *Non-pressers* during the conflict test

To investigate the role of PL neurons in regulating food-approach and threat-avoidance responses, we performed single-unit recordings across the different phases of the conflict test (*Figure 2A*). We aligned the activity of PL neurons to the onset of the food cues during the reward phase and tracked the firing rates of the same cells during the conflict phase. Using the behavioral classification shown in *Figure 1J*, we separated the animals into *Pressers* or *Non-pressers* and compared changes in PL activity in response to food cues during the reward and conflict phases (*Figure 2B–V*). When PL

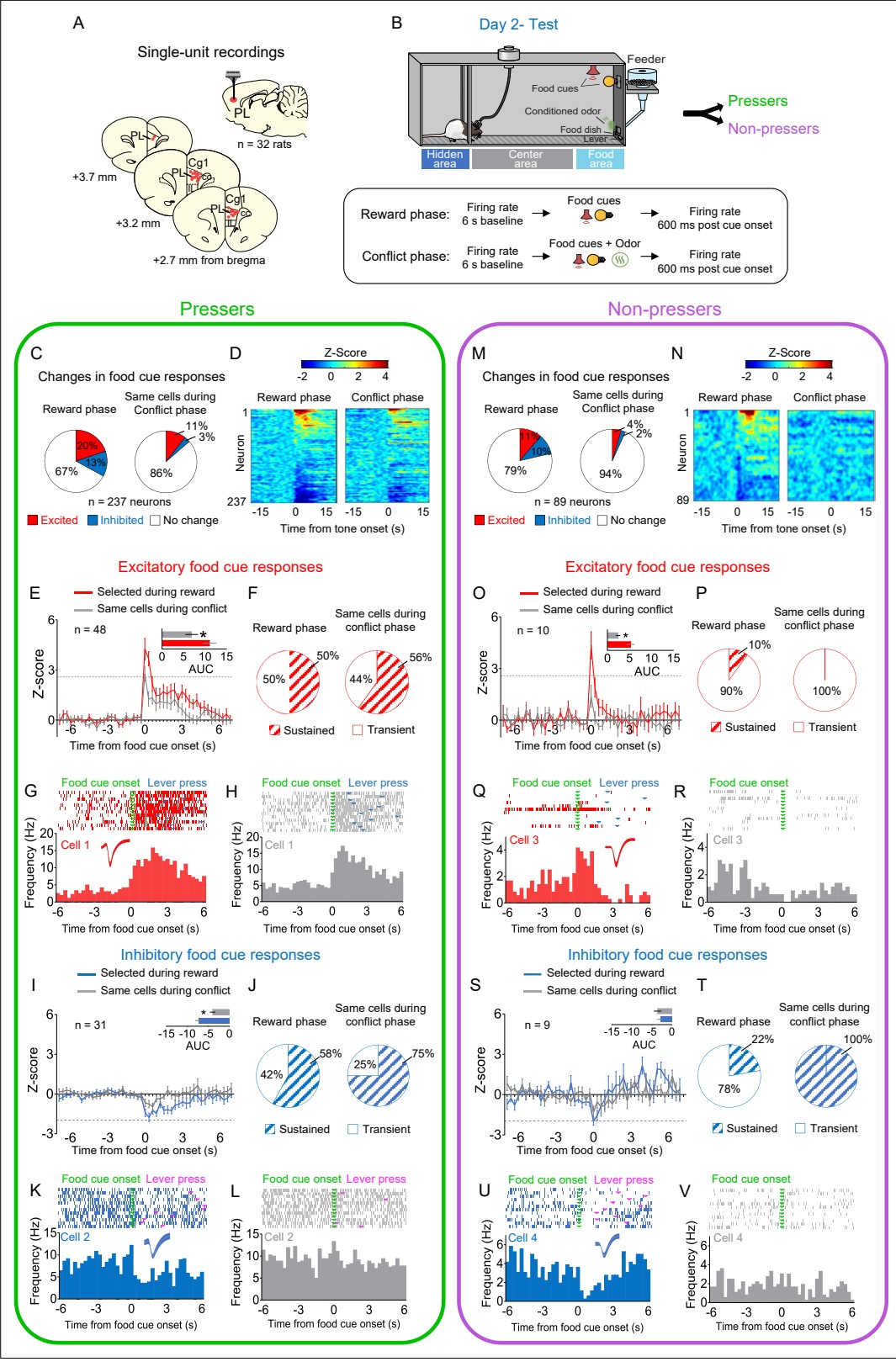

**Figure 2.** Prelimbic (PL) neurons respond differently to reward cues in *Pressers* vs. *Non-pressers* during conflict. (**A**) Diagram of the electrode placements in PL. (**B**) Schematic and timeline of PL recordings for food cue responses during of the approach–avoidance conflict test (12 food cues per phase). (**C**) Pie charts showing changes in PL firing rate in response to food cues during reward (left) vs. conflict (right) phases for *Pressers* (*n* = 237 neurons

*Figure 2 continued on next page*

*Figure 2 continued*

from 25 rats, Fisher Exact Test, responsive during reward phase: *n* = 79, responsive during conflict phase: *n* = 32, p < 0.001; excited during reward phase: *n* = 48, excited during conflict phase: *n* = 25, *p* = 0.0049; inhibited during reward phase: *n* = 31, inhibited during conflict phase: *n* = 7, p < 0.001). (**D**) Heatmap of *Z*-scored neural activities for PL neurons selected during reward phase and tracked to conflict phase. (**E**) Average peristimulus time histograms (PSTHs) for all PL neurons showing excitatory food-cue responses (*Z*-score >2.58, dotted line) during reward (red line) compared to the same cells during conflict (gray line). (**E**) Inset: differences in the positive area under the curve (AUC) between the two phases (Shapiro–Wilk normality test, p < 0.001; Wilcoxon test, *W* = −824, excitatory responses reward phase vs. conflict phase, p < 0.001). (**F**) Pie charts showing the percentage of sustained vs. transient excitatory food-cue responses in PL neurons during the reward phase with the same neurons tracked during the conflict phase. Representative PSTHs for a PL neuron showing excitatory responses to food cues during the (**G**) reward phase vs. the same neuron during the (**H**) conflict phase. (**I**) Average PSTHs for all PL neurons showing inhibitory food-cue responses (*Z*-score <−1.96, dotted line) during reward (blue line) compared to the same cells during conflict (gray line). (**I**) Inset: differences in the negative AUC between the two phases (Shapiro–Wilk normality test, p < 0.001; Wilcoxon test, *W* = 367, inhibitory responses reward phase vs. conflict phase, p < 0.001). (**J**) Pie charts showing the percentage of sustained vs. transient inhibitory food-cue responses in PL neurons during the reward phase with the same neurons tracked during the conflict phase. Representative PSTHs for a PL neuron showing inhibitory responses to food cues during the reward phase (**K**) vs. the same neuron during the conflict phase (**L**). (**M**) Pie charts showing changes in PL firing rate in response to food cues during reward (left) vs. conflict (right) phases for *Non-pressers* (*n* = 89 neurons from seven rats; Fisher Exact Test, responsive during reward phase: *n* = 19, responsive during conflict phase: *n* = 6, p < 0.0086; excited during reward phase: *n* = 10, excited during conflict phase: *n* = 4, p = 0.162; inhibited during reward phase: *n* = 9, inhibited during conflict phase: *n* = 2, p = 0.057). (**N–O**) Same as D and E, but for *Non-pressers*. (**O**) Inset: differences in the positive AUC between the two phases (paired Student's *t*-test, t = 2.34, p = 0.043). (**P–S**) Same as (**F–I**) but for *Non-pressers*. (**S**) Inset: differences in the negative AUC between the two phases (paired Student's *t*-test, t = 0.59, p = 0.569) (**T–V**). Same as (**J–L**) but for *Non-pressers*. The threshold used to identify significant differences per neurons was *Z*-score >2.58 for excitation and *Z*-score <−1.96 for inhibition. cc, corpus callosum; CG1, anterior cingulate cortex; IL, infralimbic cortex. All *p's < 0.05. All statistical analysis details are presented in **Source data 1**. See also **Figure 2—figure supplements 1–4**.

The online version of this article includes the following source data and figure supplement(s) for figure 2:

**Source data 1.** Source data for **Figure 2**.

**Figure supplement 1.** Correlation between food-cue-evoked PL activity and lever press latency during the conflict phase in *Pressers*.

**Figure supplement 1—source data 1.** Source data for **Figure 2—figure supplement 1**.

**Figure supplement 2.** Changes in prelimbic (PL) responses to reward cues selected during the conflict phase for *Pressers* and *Non-pressers*.

**Figure supplement 2—source data 1.** Source data for **Figure 2—figure supplement 2**.

**Figure supplement 3.** Distinct subpopulations of prelimbic (PL) neurons change their firing rates in response to food cues, lever pressers, and dish entries.

**Figure supplement 3—source data 1.** Source data for **Figure 2—figure supplement 3**.

**Figure supplement 4.** Changes in prelimbic (PL) spontaneous activity in *Pressers* and *Non-pressers* across the different phases of the test session.

**Figure supplement 4—source data 1.** Source data for **Figure 2—figure supplement 4**.

activity was time locked to the onset of the food cues during the reward phase, *Pressers* showed a higher number of food-cue responsive neurons than *Non-pressers* (**Figure 2C, D** vs. **Figure 2M, N**; Fisher Exact Test, 33% in *Pressers* vs. 21% in *Non-pressers*, p = 0.0418), with a similar proportion of excitatory and inhibitory responses between the two groups (Fisher Exact Test, p = 0.073 for excitatory, p = 0.571 for inhibitory). During the conflict phase, both *Pressers* and *Non-pressers* showed a significant reduction in the number of food-cue responsive neurons (**Figure 2C, D** vs. **Figure 2M, N**; Fisher Exact Test, from 33% to 14% in *Pressers*, p < 0.001; from 21% to 6% in *Non-pressers*, p = 0.0086), as well as in the magnitude of excitatory food-cue responses compared to the reward phase (**Figure 2E**, inset and **Figure 2O**, inset, Shapiro–Wilk normality test, p < 0.001, Wilcoxon test, *Pressers* – *W* = −824, p < 0.001; *Non-pressers* – *W* = −37, p = 0.032). In addition, the percentage of reduction in the number of responsive cells across the phases was similar between *Pressers* and *Non-pressers* (**Figure 2C** vs. **Figure 2M**, Fisher Exact Test, 47 out of 232 neurons for *Pressers*, 13 out of 89 neurons

for *Non-pressers*, p = 0.427), suggesting that PL neurons can distinguish between reward and conflict situations (*Figure 2G* vs. *Figure 2H* and *Figure 2Q* vs. *Figure 2R*).

Using a temporal frequency separation of the food-cue responses into transient (<600-ms duration) and sustained (≥900-ms duration) activity (*Bezdudnaya et al., 2006*), we revealed that *Pressers* display a higher proportion of sustained excitatory responses during the reward phase, when compared to *Non-pressers* (*Figure 2F* vs. *Figure 2P*, 50% in *Pressers* vs. 10% in *Non-pressers*, Fisher Exact Test, p = 0.032). In addition, *Pressers* showed a higher magnitude of inhibitory food-cue responses during the reward phase when compared to *Non-pressers* (*Figure 2I*, blue bar inset vs. *Figure 2S*, blue bar inset, Shapiro–Wilk normality test, p < 0.001, Mann–Whitney test, $U = 50$, p = 0.0045) and, in contrast to *Non-pressers*, such responses were attenuated during the conflict phase (*Figure 2I*, inset vs. *Figure 2S*, inset, *Pressers*, Shapiro–Wilk normality test, p < 0.001, Wilcoxon test – $W = 367$, p < 0.001; *Non-pressers*, paired Student's $t$-test, $t = 0.59$, p = 0.569). A correlation analysis during the conflict phase revealed that food-cue-evoked excitatory PL activity in *Pressers* was inversely correlated with lever press latency, indicating that the higher the firing rate of food-cue responsive cells after the onset of the audiovisual cues, the quicker the animals searched for rewards (*Figure 2—figure supplement 1A, B*).

Next, we time locked the activity of PL neurons to the onset of the food cues during the conflict phase. Both *Pressers* and *Non-pressers* show the same percentage of food-cue responsive neurons (*Figure 2—figure supplement 2A, B* vs. *Figure 2—figure supplement 2K, L*, 28% in *Pressers* vs. 20% in *Non-pressers*, Fisher Exact Test, p = 0.391) and the same magnitude of excitatory food-cue responses during the conflict phase (*Figure 2—figure supplement 2C, E, F* vs. *Figure 2—figure supplement 2M, O, P*; area under the curve, *Pressers* vs. *Non-pressers*, Shapiro–Wilk normality test, p < 0.001, Mann–Whitney, $U = 107$, p = 0.123). However, in *Pressers*, 42% of excitatory food-cue responses showed sustained activity during the conflict phase whereas such responses were completely absent in *Non-pressers* (*Figure 2—figure supplement 2D* vs. *Figure 2—figure supplement 2N*, Fisher Exact Test, p = 0.018).

In addition to food-cue responses, we observed a significant proportion of PL neurons that changed their firing rates in response to lever presses (23%) or rewarded food dish entries (16%, *Figure 2—figure supplement 3A, N*). A longitudinal tracking of PL activity throughout the reward phase demonstrated that most PL responsive neurons changed their activities selectively to food cues, lever presses, or food dish entries, with a smaller number of cells responding during two or more of these events (*Figure 2—figure supplement 3O*). An analysis of PL activity at a random time-point (e.g., 1 s before the food-cue onset) resulted in less than 3% of responsive cells, indicating that the proportion of PL neurons that responded to food cues, lever presses, and food dish entries was different from the proportion obtained by chance (Fisher Exact Test, all p's < 0.05, see *Source data 1*). These observations suggest that PL neurons exhibit a heterogeneous pattern of activity during reward-seeking behavior, consistent with a recent study using calcium imaging recordings from PL neurons in head-fixed mice (*Grant et al., 2021*).

To further explore whether changes in activity dynamics of PL neurons differ between *Pressers* and *Non-pressers*, we compared the spontaneous firing rate of the neurons before vs. after each phase of the test session (*Figure 2—figure supplement 4A*). While *Pressers* showed the same proportion of neurons excited and inhibited across the different phases (Fisher Exact Test, all p's > 0.05, see *Source data 1*), *Non-pressers* exhibited a significant increase in the proportion of neurons excited during the conflict phase (*Figure 2—figure supplement 4C*, Fisher Exact Test, p = 0.015 compared to odor phase, p = 0.059 compared to *Pressers*). This suggests that increased spontaneous activity in PL neurons during the conflict phase may be associated with the complete suppression in lever presses observed in *Non-pressers* (*Figure 1J*). Collectively, these results suggest that differences in the number and magnitude of excitatory food-cue responses, as well as in the spontaneous activity of PL neurons during the conflict test, may contribute to the individual differences in risky decision-making observed between the two behavioral phenotypes.

## Different subsets of PL neurons signal freezing, avoidance, and risk-assessment behaviors in both *Pressers* and *Non-pressers*

To investigate whether PL activity correlates with the expression of distinct defensive behaviors during the test session, we used a pose estimation algorithm (DeepLabCut, see Methods for details) to

identify the onset of freezing, avoidance, or risk-assessment responses and align these time points with the activity of PL neurons. We found that a small percentage of PL neurons changed their firing rates during the onset of freezing (*Figure 3A*), avoidance (*Figure 3B*), or risk-assessment (*Figure 3C*) behaviors in both *Pressers* and *Non-pressers*, with a similar proportion of excitatory and inhibitory responses being observed in the two groups (*Figure 3A–I*). Interestingly, most PL responsive neurons (80%) changed their activities exclusively during the onset of one of these three behaviors, with a reduced number of avoidance-responsive cells also responding during the onset of risk-assessment behavior (*Figure 3J–M*). Moreover, a smaller fraction of PL neurons changed their firing rates 600 ms before the onset of either freezing, avoidance, or risk-assessment responses in both *Pressers* and *Non-pressers* (*Figure 3—figure supplement 1A–M*), indicating that some PL neurons can anticipate an animal's defensive behavior during the test. Overall, these results suggest that different subsets of PL neurons signal distinct behavioral outcomes during a conflict situation, with only a reduced number of PL neurons encoding the aversive salience of environmental cues independently of the defensive response expressed by the animal.

## *Pressers* and *Non-pressers* show significant differences in delta and theta oscillations in PL

Previous studies have shown that oscillations in mPFC neuronal activity at different frequency bands correlate with distinct behavioral states in both rodents and humans (*Narayanan et al., 2013*; *Harris and Gordon, 2015*). Neural oscillations in the mPFC emerge from the network of excitatory and inhibitory synaptic connections and are thought to contribute to neural communication when subjects engage in reward and threat memory tasks (*Hyman et al., 2011*; *Likhtik and Paz, 2015*; *Park and Moghaddam, 2017*; *Widge et al., 2019*). To investigate whether *Pressers* and *Non-pressers* show significant differences in PL oscillations during conflict, we recorded local field potentials (LFPs) from PL neurons and calculated the average of power spectral density (PSD) at different frequencies across the test session. After comparing the PSD contribution for each frequency range in *Pressers* and *Non-pressers*, we observed that most of the signal originated from the delta (0–4 Hz) and theta (4–10 Hz) bands, with a much smaller contribution coming from the alpha (10–14 Hz), beta (14–35 Hz), and gamma (>35 Hz) frequencies (*Figure 4A*). We therefore focused our analyses on these two bands and found that *Pressers* displayed increased power in the delta band, whereas *Non-pressers* exhibited increased power in the theta band during the three phases of the test session (*Figure 4B, C*, Welch's *t*-test of area under the curve, all p's < 0.001, see *Source data 1*). Differences between *Pressers* and *Non-pressers* were also observed in the time–frequency domain through changes in the log of PSD for delta and theta bands across the different phases (*Figure 4D, E*, paired Student's *t*-test of area under the curve, all p's < 0.001, see *Source data 1*). These results indicate that phenotypic differences in approach–avoidance conflict are associated with distinct oscillatory frequencies in PL.

## In pressers, PL$^{GLUT}$ neurons show reduced spontaneous activity during the conflict phase

The rodent mPFC, including PL, is primarily composed of excitatory glutamatergic cells that correspond to 75–85% of the neurons in this area. In contrast, inhibitory GABAergic interneurons comprise 15–25% of the local neurons (*Santana et al., 2004*; *Gabbott et al., 2005*). Previous studies have shown that PL glutamatergic (PL$^{GLUT}$) neurons are necessary for the retrieval of conditioned threat responses (*Do-Monte et al., 2015*), whereas PL GABAergic (PL$^{GABA}$) neurons are implicated in both the encoding and the retrieval of threat associations by regulating the firing rate of PL$^{GLUT}$ neurons (*Courtin et al., 2014*; *Cummings and Clem, 2019*). In addition, during foraging in a safe context, food-associated cues activate both PL$^{GLUT}$ and PL$^{GABA}$ neurons (*Burgos-Robles et al., 2013*; *Gaykema et al., 2014*), and inactivation of PL$^{GLUT}$ neurons may increase or reduce conditioned food-seeking responses depending on the specific downstream projections that are being modulated (*Otis et al., 2017*). While these studies suggest a role for both PL$^{GLUT}$ and PL$^{GABA}$ neurons in the regulation of threat and food-seeking responses in isolation, it remains unexplored how these two subsets of PL neurons regulate the trade-off between seeking rewards and avoiding potential threats during a conflict situation. To address this question, we combined single-unit recordings with optogenetics to track the neuronal activity of photoidentified PL$^{GLUT}$ and PL$^{GABA}$ neurons during the test session.

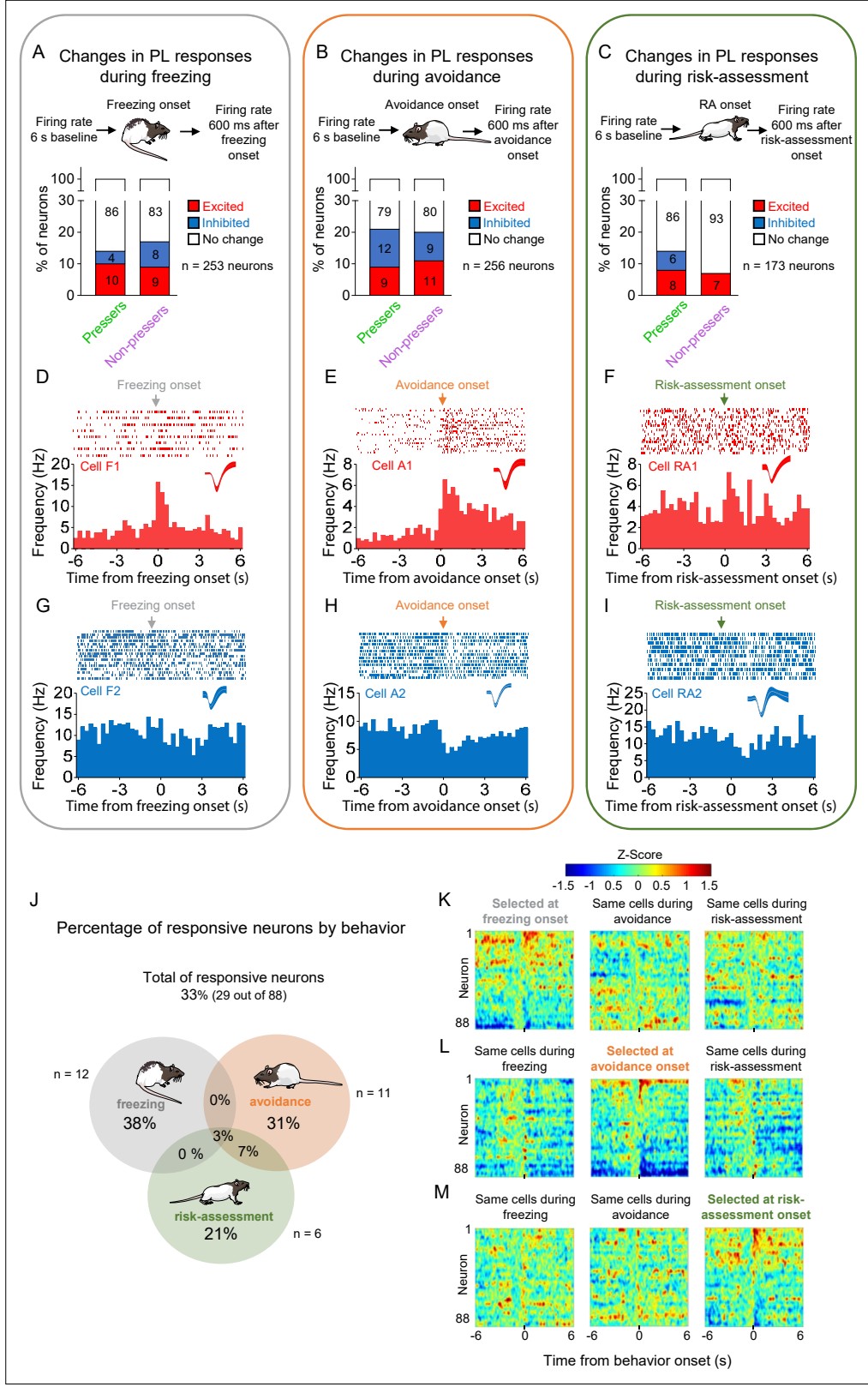

**Figure 3.** Prelimbic (PL) activity correlates with the onset of freezing, avoidance, or risk-assessment behaviors in both *Pressers* and *Non-pressers*. Both *Pressers* and *Non-pressers* showed the same number and proportion of excitatory and inhibitory PL responses during the onset of (**A**) freezing (Fisher Exact Test, responsive neurons in *Pressers*: 22 neurons, in *Non-pressers*: 15 neurons, p = 0.462), (**B**) avoidance (Fisher Exact Test, responsive

*Figure 3 continued on next page*

*Figure 3 continued*

neurons in *Pressers*: 43 neurons, in *Non-pressers*: 9 neurons, p = 0.999), or (**C**) risk-assessment (RA, Fisher Exact Test, responsive neurons in *Pressers*: 12 neurons, in *Non-pressers*: 6 neurons, p = 0.318) behaviors. Representative peristimulus time histograms (PSTHs) for distinct PL neurons showing excitatory responses at the onset of (**D**) freezing, (**E**) avoidance, or (**F**) risk-assessment behaviors. Representative PSTHs for distinct PL neurons showing inhibitory responses at the onset of freezing (**G**), avoidance (**H**), or risk-assessment (**I**) behaviors. (**J**) Venn diagram showing the percentage of all PL responsive neurons (29 out of 88 neurons) by behavior. Most of the responsive neurons responded selectively at the onset of one of the behaviors. Heatmap of *Z*-scored neural activities for PL neurons selected at the onset of freezing (**K**), avoidance (**L**), or risk-assessment behavior (**M**) with the same cells tracked during the other behaviors. The threshold used to identify significant differences per neurons was *Z*-score >2.58 for excitation and *Z*-score <−1.96 for inhibition. n.s. = nonsignificant. All statistical analysis details are presented in *Source data 1*. See also *Figure 3—figure supplement 1*.

The online version of this article includes the following source data and figure supplement(s) for figure 3:

**Source data 1.** Source data for *Figure 3*.

**Figure supplement 1.** Prelimbic (PL) activity anticipates the onset of freezing, avoidance, or risk-assessment behaviors in both *Pressers* and *Non-pressers*.

**Figure supplement 1—source data 1.** Source data for *Figure 3—figure supplement 1*.

---

For photoidentification of PL^GLUT neurons, we injected into PL a viral vector (AAV-CaMKIIα-hChR2-(H134R)-eYFP) with a gene promoter (CaMKIIα) that favors the expression of the light-activated cation channel channelrhodopsin (ChR2) in PL^GLUT neurons. This CaMKIIα labeling approach has been successfully used in previous studies (*Gradinaru et al., 2009*; *Tye et al., 2011*) and was validated here for PL neurons by showing a lack of immunocolabeling between the viral vector and the GABAergic marker GAD67 (*Figure 5A*). Rats expressing ChR2 selectively in PL^GLUT neurons were implanted with an optrode into the same region for optogenetic-mediated identification of PL^GLUT neurons at the end of the behavioral session (*Figure 5B*). Among the recorded PL cells, 36 out of 104 neurons (*n* = 5 rats) showed short-latency responses (<6 ms) and high spike reliability (Fano factor ratio >1) to laser illumination and were classified as PL^GLUT neurons (*Figure 5C, D* and Materials and methods). The <6 ms criterion was defined by using the triangle method detection (*Zack et al., 1977*) to identify the cluster division in the histogram distribution of response latencies (*Figure 5C* and Methods). The <6 ms criterion was similar or stricter than the response latency criterion used in previous photoidentification studies in vivo (*Lima et al., 2009*; *Cohen et al., 2012*; *Burgos-Robles et al., 2017*; *Allsop et al., 2018*). Photoactivation of PL^GLUT neurons can lead to indirect activation of synaptically connected neurons in the same cortical region, but these indirect responses to laser illumination take longer than 9 ms to occur (*Lima et al., 2009*). For photoidentification of PL^GABA neurons, we injected into PL a viral vector (AAV-mDlx-ChR2-mCherry) with a gene promoter (mDlx) that favors the expression of ChR2 in PL^GABA neurons. This mDlx labeling approach has been successfully used in previous studies (*Dimidschstein et al., 2016*; *Sun et al., 2020*), and was validated here for PL neurons by using two different methods: an immunohistochemical approach that resulted in significant immunocolabeling between the viral vector and the GABAergic marker GAD67 (*Figure 5E*), and an in situ hybridization approach which confirmed that ~88% of the cells labeled with the viral vector also expressed the GABAergic marker vGAT (*Figure 5—figure supplement 1A, B*). Rats expressing ChR2 selectively in PL^GABA neurons were implanted with an optrode into the same region for optogenetic-mediated identification of PL^GABA neurons at the end of the behavioral session (*Figure 5F*). Among the recorded PL cells, 69 out of 338 neurons (*n* = 19 rats) showed short-latency responses (<6 ms) and high spike reliability (Fano factor ratio >1) to laser illumination and were classified as PL^GABA neurons (*Figure 5G, H* and Materials and methods).

After separating the photoidentified cells into PL^GLUT and PL^GABA neurons, we aligned their activities to the onset of the food cues and compared changes in firing rates from the reward to the conflict phase in *Pressers* (*Figure 5I*). We observed that the proportions of excitatory and inhibitory food-cue responses for PL^GLUT and PL^GABA neurons were similar when comparing between the reward and the conflict phases as well as within each one of the phases (*Figure 5J, K*, Fisher Exact Test, all p's > 0.05, see *Source data 1*). Next, we analyzed the spontaneous activity of PL^GLUT and PL^GABA neurons and compared changes in their firing rates across the different phases of the test session (*Figure 5L*). We found that the average firing rate of PL^GLUT neurons remained the same across the different phases of

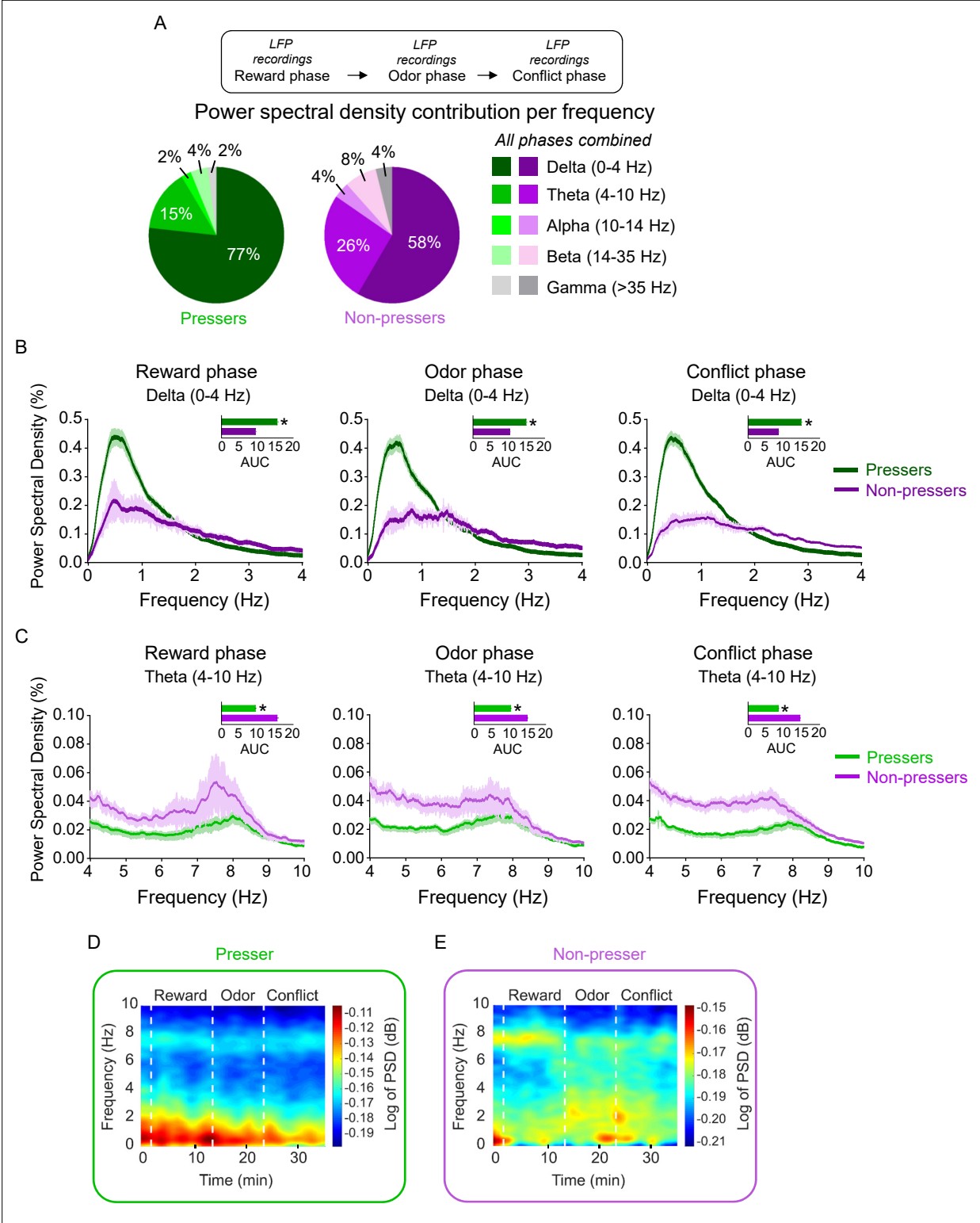

**Figure 4.** *Pressers* and *Non-pressers* show significant differences in prelimbic (PL) oscillations during the test session. (**A**) Power spectral density (PSD) contribution at different frequency bands. Average of PSD (%) in the (**B**) delta (0–4 Hz) or (**C**) theta (4–10 Hz) bands in *Pressers* (green line, n = 25 rats) and *Non-pressers* (purple line, n = 7 rats) during the (left) reward, (center) odor, and (right) conflict phases of the test session. *Pressers* showed increased power in the delta band, whereas *Non-pressers* showed increased power in the theta band during the three phases of the test session (unpaired Student's t-test comparing *Pressers* vs. *Non-pressers*, all p's < 0.001). Representative time–frequency spectrogram showing changes in the log of PSD (dB) for delta and theta bands in (**D**) *Pressers* and (**E**) *Non-pressers* across the different phases of the session. All statistical analysis details are presented

*Figure 4 continued on next page*

*Figure 4 continued*

in *Source data 1*.

The online version of this article includes the following source data for figure 4:

**Source data 1.** Source data for *Figure 4*.

the test (~5 Hz; *Figure 5M*, One-way repeated measures ANOVA, $F(2.03, 69.02) = 1.204$, $p = 0.306$), with most of the cells (57%) changing their activities in more than one session (*Figure 5N*). An average firing rate analysis across phases demonstrated that the activity of PL$^{GLUT}$ neurons did not change significantly from the reward to the odor phase (Fisher Exact Test, all p's > 0.05, see *Source data 1*), but was inhibited from the odor to the conflict phase when *Pressers* resumed searching for food (*Figure 5O*, Fisher Exact Test, odor vs. conflict, $p = 0.0046$). Similar to PL$^{GLUT}$ neurons, the average firing rate of PL$^{GABA}$ neurons also remained the same across the different phases of the test (~8 Hz, *Figure 5P*, one-way repeated measures ANOVA, $F(1.164, 79.17) = 0.013$, $p = 0.935$), with most of the cells (62%) changing their activities in more than one session (*Figure 5Q*). However, in contrast to PL$^{GLUT}$ neurons, a group analysis of the firing rates of PL$^{GABA}$ neurons did not reveal significant differences across the phases (*Figure 5R*, Fisher Exact Test, all p's > 0.05, see *Source data 1*). Because PL is comprised of different subpopulations of interneurons that inhibit each other during food seeking or defensive responses (*Gaykema et al., 2014*; *Cummings and Clem, 2019*), we cannot rule out the possibility that distinct subsets of PL$^{GABA}$ neurons were preferentially recruited during each one of the phases.

To evaluate how the spontaneous activity of the same PL neurons changed during the test session, we tracked the firing rate of PL$^{GLUT}$ and PL$^{GABA}$ neurons across the different phases. We found that all PL$^{GLUT}$ neurons that were either excited or inhibited during the reward phase responded in opposite direction or did not change their activities during the odor phase (*Figure 5—figure supplement 2A-B*), suggesting the existence of distinct subpopulations of PL$^{GLUT}$ neurons that encode reward- and threat-related information differently in our task. In contrast, no significant differences in the proportions of excitation and inhibition were observed in PL$^{GABA}$ neurons during the transition from reward to odor phase nor during the transition from odor to conflict phase for both subsets of PL neurons (*Figure 5—figure supplement 2C–F*). Furthermore, both PL$^{GLUT}$ and PL$^{GABA}$ neurons showed the same proportion of excitatory and inhibitory responses before or after the onset of freezing, avoidance, or risk-assessment behaviors (*Figure 5—figure supplement 3A–F*, Fisher Exact Test, all p's > 0.05, see *Source data 1*). These results indicate that both glutamatergic and GABAergic neurons in PL may contribute to the expression of distinct defensive responses during conflict. Together, our data suggest that a significant proportion of PL$^{GLUT}$ neurons are inhibited when rat's behavior transitions from increased defensive responses during the conditioned odor phase to increased food-seeking responses during the conflict phase.

## Photoactivation of PL$^{GLUT}$, but not PL$^{GABA}$ neurons, suppresses reward-seeking responses

To further establish whether changes in the activity of PL neurons can alter cue-triggered food-seeking responses, we used an optogenetic approach to selectively activate either PL$^{GLUT}$ or PL$^{GABA}$ neurons during a cued food-seeking test in a neutral context. We initially infused either the viral vector AAV-CaMKIIα-ChR2-eYFP (*Figure 6A*) or AAV-mDlx-ChR2-mCherry (*Figure 6E*) into PL and implanted an optrode into the same region to examine how photoactivation of PL$^{GLUT}$ or PL$^{GABA}$ neurons change local activity. Laser illumination of PL$^{GLUT}$ somata increased the firing rate of most responsive PL neurons (9 out of 20 neurons, 45%), with some neurons showing reduced activity (6 out of 20 neurons, 30%, *Figure 6B–D*). Neurons that increased their activities showed shorter response latencies (3.31 ± 1.03 ms) compared to neurons that reduced their activities (21.0 ± 3.74 ms) when analyzed in short bins of 1ms, suggesting direct responses (i.e., opsin-mediated) vs. indirect responses (i.e., multisynaptic), respectively. Conversely, although some PL$^{GABA}$ neurons showed increased activity right after the laser onset (revealed by short bins of 1ms, as shown in *Figure 5E–H*), illumination of PL$^{GABA}$ somata reduced the firing rate of all responsive PL neurons when analyzing the entire duration of the train (16 out of 22 neurons, 73%; *Figure 6F–H*), indicating a suppression in local activity.

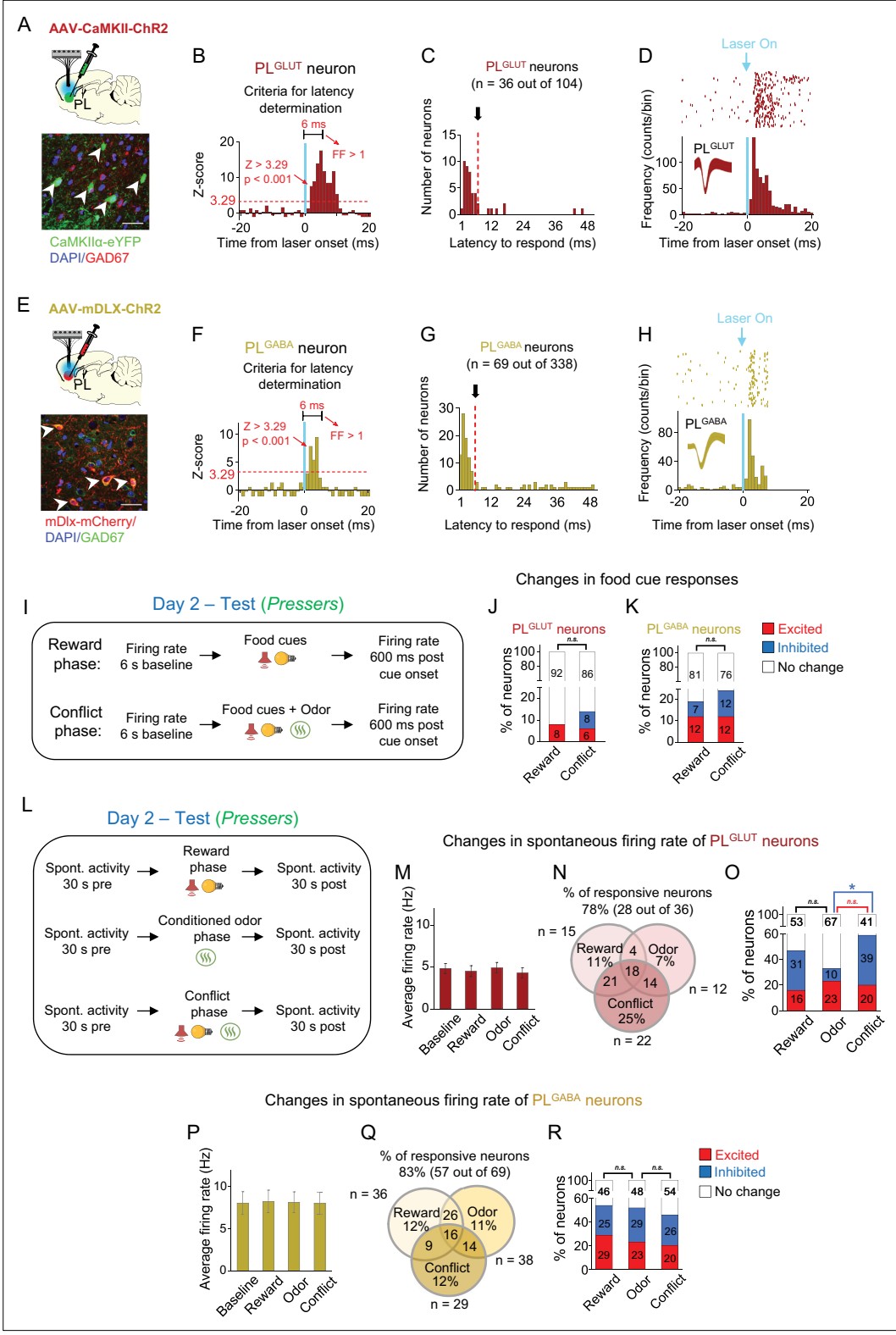

**Figure 5.** In pressers, PL[GLUT] neurons show reduced spontaneous activity during the conflict phase. (**A**) Top, schematic of viral infusion. Bottom, representative immunohistochemical micrograph showing lack of colabeling (white arrows) between the ChR2 viral construct (green, AAV-CaMKII-ChR2-eYFP) and the GABA marker GAD67 (red), confirming that the use of a CaMKII promoter enables transgene expression favoring prelimbic (PL) glutamatergic neurons. Scale bars: 25 µm. (**B–D**) Photoidentification of PL[GLUT] neurons. (**B**) Frequency histogram

*Figure 5 continued on next page*

*Figure 5 continued*

showing the latency of response to laser illumination for PL neurons (*n* = 36 photoidentified PL$^{GLUT}$ neurons out of 104 recorded cells). Triangle method detection of cluster distribution revealed a separation of latency frequencies at 6ms (see details in Methods). (**C**) Cells with photoresponse latencies <6 ms (identified as the first bin with Z-score >3.29, p < 0.001, red dotted line) and high spike reliability during the 6 ms (Fano factor [FF] ratio >1 compared to prelaser baseline) were classified as PL$^{GLUT}$ neurons (see details in Methods). (**D**) Raster plot and peristimulus time histogram showing a representative PL$^{GLUT}$ neuron responding to a 5 Hz train of laser stimulation. (**E**) Top, schematic of viral infusion. Bottom, representative immunohistochemical micrograph showing colabeling (white arrows) between the ChR2 viral construct (red, AAV-mDlx-ChR2-mCherry) and the GABA marker GAD67 (green), confirming that the use of a mDlx promoter enables transgene expression favoring PL$^{GABA}$ neurons. Scale bars: 25 µm. (**F–H**) Photoidentification of PL$^{GABA}$ neurons. (**F**) Frequency histogram showing the latency of response to laser illumination for PL neurons (*n* = 69 photoidentified PL$^{GABA}$ neurons out of 338 recorded neurons). Triangle method detection of cluster distribution revealed a separation of latency frequencies at 6 ms (see details in Methods). (**G**) Cells with photoresponse latencies <6 ms (identified as the first bin with Z-score >3.29, p < 0.001, red dotted line) and high spike reliability during the 6 ms (FF, Fano factor ratio >1 compared to prelaser baseline) were classified as PL$^{GABA}$ neurons (see details in Methods). (**H**) Raster plot and peristimulus time histogram showing a representative PL$^{GABA}$ neuron responding to a 5 Hz train of laser stimulation. Vertical blue bars: laser onset. Bins of 1 ms. (**I**) Timeline of PL recordings for food-cue responses in *Pressers* during test (12 food cues per phase). Stacked bar showing the percentage of (**J**) PL$^{GLUT}$ neurons or (**K**) PL$^{GABA}$ neurons that changed their firing rates in response to food cues from the reward phase to the conflict phase. No significant differences were observed across the phases (Fisher Exact Test, all p's > 0.05; n.s. = nonsignificant). (**L**) Timeline of PL recordings for spontaneous activity in *Pressers* during test. (**M**) Average firing rate of PL$^{GLUT}$ neurons across the different phases of test. (**N**) Venn diagram showing the percentage of responsive PL$^{GLUT}$ neurons (28 out of 36 neurons) by events. (**O**) Stacked bar showing the percentage of PL$^{GLUT}$ neurons that changed their spontaneous firing rates across the different phases of the test. PL$^{GLUT}$ neurons did not change their firing rates from the reward to the odor phase (Fisher Exact Test, inhibited in reward phase: 10 neurons, inhibited in odor phase: 3 neurons, p = 0.063), but were subsequently inhibited from the odor to the conflict phase (Fisher Exact Test, inhibited in odor phase: 3 neurons, inhibited in conflict phase: 14 neurons, p = 0.0046). (**P**) Average firing rate of PL$^{GABA}$ neurons across the different phases of test. (**Q**) Venn diagram showing the percentage of responsive PL$^{GABA}$ neurons (57 out of 69 neurons) by events. (**R**) Stacked bar showing the percentage of PL$^{GABA}$ neurons that changed their spontaneous firing rates across the different phases of the test. No significant differences were observed across the phases (Fisher Exact Test, all p's > 0.05; n.s. = nonsignificant). All statistical analysis details are presented in *Source data 1*. See also *Figure 5—figure supplements 1–3*.

The online version of this article includes the following source data and figure supplement(s) for figure 5:

**Source data 1.** Source data for *Figure 5*.

**Figure supplement 1.** Validation of the mDlx promoter used for viral vector targeting of GABAergic neurons in prelimbic (PL).

**Figure supplement 1—source data 1.** Source data for *Figure 5—figure supplement 1*.

**Figure supplement 2.** Changes in the spontaneous firing rate of PL$^{GLUT}$ and PL$^{GABA}$ neurons across the different phases of the test session.

**Figure supplement 2—source data 1.** Source data for *Figure 5—figure supplement 2*.

**Figure supplement 3.** Changes in the firing rate of PL$^{GLUT}$ and PL$^{GABA}$ neurons before or after the onset of freezing, avoidance, or risk-assessment behaviors.

**Figure supplement 3—source data 1.** Source data for *Figure 5—figure supplement 3*.

---

After investigating the local effects of photoactivating either PL$^{GLUT}$ and PL$^{GABA}$ neurons, we infused another set of animals with the same viral vectors in PL and implanted bilateral optical fibers into the same region to manipulate PL activity during the cued food-seeking test (*Figure 6I, J*). Rats expressing only eYFP in PL were used to control for any nonspecific effects of viral transduction or laser heating. To assess the effects of PL photoactivation on lever presses, we alternated 2 trials of food cues with the laser on vs. laser off conditions in a total of 12 trials (*Figure 6K–L*). Photoactivation of PL$^{GLUT}$ (CaMKII-ChR2), but not PL$^{GABA}$ (mDlx-ChR2) neurons, reduced the frequency of lever presses (*Figure 6M*, two-way repeated measures ANOVA, $F_{(10, 180)}$ = 7.009, p < 0.001; Bonferroni post hoc, CaMKII-ChR2 vs. Control, all laser on periods – p < 0.001, mDlx-ChR2 vs. Control, all laser on periods – p > 0.05) and increased the latency for the first press after the cue onset (*Figure 6N*, $F_{(10, 180)}$ = 9.931, p < 0.001; Bonferroni post hoc, CaMKII-ChR2 vs. Control, all laser on periods – p

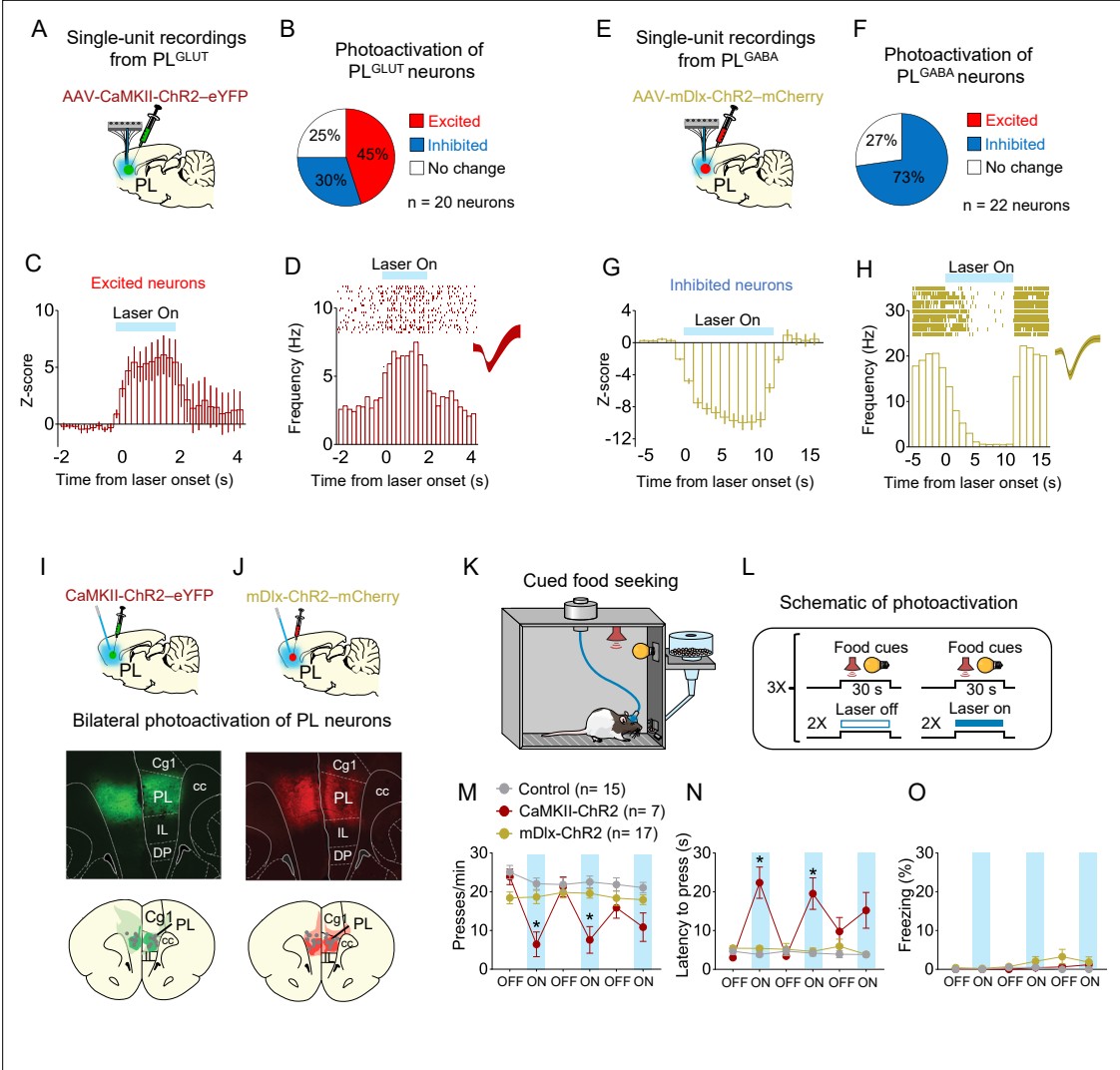

**Figure 6.** Photoactivation of PL^GLUT, but not PL^GABA, neurons suppresses reward seeking in a neutral context. (**A**) Schematic of viral infusion and recordings in prelimbic (PL). (**B**) Changes in PL firing rate with illumination of PL^GLUT neurons in rats expressing AAV-CaMKII-ChR2-eYFP in PL ($n = 20$ neurons). (**C**) Average peristimulus time histogram (PSTH) of PL neurons that were excited during laser illumination of PL^GLUT neurons. (**D**) Raster plot and PSTH of representative PL neuron showing excitatory responses to illumination in rats expressing AAV-CaMKII-ChR2-eYFP in PL. (**E**) Schematic of viral infusion and recordings in PL. (**F**) Changes in PL firing rate with illumination of PL^GABA neurons in rats expressing AAV-mDlx-ChR2-mCherry in PL ($n = 22$ neurons). (**G**) Average PSTH of PL neurons that were inhibited during laser illumination of PL^GABA neurons. (**H**) Raster plot and PSTH of representative PL neuron showing inhibitory responses to illumination in rats expressing AAV-mDlx-ChR2-mCherry in PL. Representative micrograph showing the expression of (**I**) CaMKII-ChR2-eYFP or (**J**) mDlx-ChR2-mChery in PL and schematic of optical fiber location (gray dots) in the same region (compressed across different anteroposterior levels of PL). Green or red areas represent the minimum (dark) and the maximum (light) viral expression into the PL. (**K, L**) Schematic and timeline of PL photoactivation during the cued food-seeking test in a neutral context. Optogenetic activation of PL^GLUT neurons (CaMKII-ChR2, dark red circles, $n = 7$), but not PL^GABA neurons (mDlx-ChR2, gold circles, $n = 17$), reduced the (**M**) frequency of lever presses ($F_{(10, 180)} = 7.009$, $p < 0.001$, Bonferroni post hoc, CaMKII-ChR2 vs. Control, all laser on periods – $p < 0.01$; mDlx-ChR2 vs. Control, all laser on periods – $p > 0.05$) and increased (**N**) the latency for the first press ($F(10, 180) = 9.931$, $p < 0.001$, CaMKII-ChR2 vs. Control, all laser on periods – Bonferroni post hoc, $p < 0.001$; mDlx-ChR2vs. Control, all laser on periods, $p > 0.05$). (**O**) Optogenetic activation of PL neurons did not alter freezing behavior ($F(10, 180) = 1.124$, $p = 0.346$). Blue shaded area represents laser-on trials (PL^GLUT: 5 Hz, PL^GABA: 20 Hz; 5 ms pulse width, 7–10 mW, 30-s duration). Data shown as mean ± standard error of the mean (SEM). Each circle represents the average of two consecutive trials. Two-way repeated measures analysis of variance (ANOVA) followed by Bonferroni post hoc test. All *p's < 0.05. All statistical analysis details are presented in *Source data 1*.

The online version of this article includes the following source data for figure 6:

**Source data 1.** Source data for *Figure 6*.

< 0.001, mDlx-ChR2 vs. Control, all laser on periods – p > 0.05), when compared to control group. The diminished behavioral effect observed during the third block of laser on could be the result of conformational changes in the opsin (e.g., photobleaching) or temporary depletion of synaptic vesicles following repeated laser illumination, as previously reported (*Kittelmann et al., 2013*; *Stahlberg et al., 2019*). Photoactivation of either PL$^{GLUT}$ or PL$^{GABA}$ neurons did not induce freezing behavior (*Figure 6O*, $F$(10, 180) = 1.124, p = 0.346). These results are consistent with our electrophysiological recordings in *Figure 5O* showing that increased inhibition in the firing rate of PL$^{GLUT}$ neurons correlates with augmented reward-seeking responses during conflict. Overall, these findings suggest that increasing the activity of PL$^{GLUT}$ neurons is sufficient to suppress cued reward-seeking responses in a neutral context.

## Photoinhibition of PL$^{GLUT}$ neurons in *Non-pressers* reduces freezing responses and increases food approaching during conflict

Our electrophysiological experiments in *Figure 5O* demonstrate that PL$^{GLUT}$ neurons are inhibited when rats' behavior transitions from defensive responses in the odor phase to food-seeking responses in the conflict phase. In addition, our photoactivation experiments in *Figure 6K–O* indicate that increasing the activity of PL$^{GLUT}$ neurons suppresses cued reward-seeking behavior in rats that are pressing a lever for food. We therefore hypothesized that photoinhibition PL$^{GLUT}$ neurons during conflict would attenuate defensive behaviors and rescue food-seeking responses in *Non-pressers*. To test this hypothesis, we injected a group of rats with the viral vector AAV-CaMKIIα-eNpHR-eYFP (or AAV-CaMKIIα-eYFP) into PL to express the inhibitory opsin halorhodopsin (or eYFP control) selectively in PL$^{GLUT}$ neurons (*Figure 7A*). Rats were initially exposed to a cued food-seeking test to assess the effects of photoinhibition of PL$^{GLUT}$ neurons on food-seeking responses in a neutral context. We observed that photoinhibition of PL$^{GLUT}$ neurons had no effect on lever pressing rate (two-way repeated measures ANOVA, $F$(5, 110) = 1.336, p = 0.254), latency to press the lever ($F$(5, 110) = 0.637, p = 0.671) or freezing ($F$(5, 95) = 1.395, p = 0.231) responses before threat conditioning (*Figure 7—figure supplement 1A–E*).

Animals were then threat conditioned as in *Figure 1* and on the following day exposed to the odor arena for a test session. During the conflict phase, the first pair of food cues was used to classify the animals into *Pressers* and *Non-pressers*, whereas the subsequent pairs of food cues were alternated between laser on and laser off conditions to assess the effects of illumination of PL$^{GLUT}$ neurons on approach–avoidance responses (*Figure 7B, C*). Remarkably, photoinhibition of PL$^{GLUT}$ neurons (CaMKII-eNpHR, Shapiro–Wilk normality test, all p's < 0.05, see *Source data 1*) in *Non-pressers* reduced the percentage of time rats spent freezing (*Figure 7D*, Wilcoxon test, $W$ = −64, laser off vs. laser on, p = 0.0020, Mann–Whitney test, $U$ = 18 Control vs. CaMKII-eNpHR, p = 0.319) and avoiding the odor area (*Figure 7E*, Wilcoxon test, $W$ = −21, laser off vs. laser on, p = 0.031; Mann–Whitney test, $U$ = 19.5 Control vs. CaMKII-eNpHR, p = 0.365), and increased the percentage of time rats spent approaching the food area (*Figure 7F*, Wilcoxon test, $W$ = 21, laser off vs. laser on, p = 0.031; Mann–Whitney test, $U$ = 17 Control vs. CaMKII-eNpHR, p = 0.221) during the food-cue presentation, when compared to the eYFP-control group (Wilcoxon test, Freezing: $W$ = 3, laser off vs. laser on, p = 0.812; avoidance: $W$ = 3, laser off vs. laser on, p = 0.500, time in food area: $W$ = −3, laser off vs. laser on, p = 0.500). Despite the increase in food approaching behavior, photoinhibition of the same cells had no effect on the percentage of rewarded lever presses (*Figure 7G*, Wilcoxon test, $W$ = 6, laser off vs. laser on, p = 0.250; Mann–Whitney test, $U$ = 22.5 Control vs. CaMKII-eNpHR, p = 0.697) or latency to press the lever (*Figure 7H*, Wilcoxon test, $W$ = −10, laser off vs. laser on, p = 0.125; Mann–Whitney test, $U$ = 21 Control vs. CaMKII-eNpHR, p = 0.357), when compared to the control group (Wilcoxon test, rewarded lever presses: $W$ = −1, laser off vs. laser on, p > 0.999; latency to press the lever: $W$ = 1, laser off vs. laser on, p > 0.999).

In another subset of *Non-pressers* (*Figure 7I–K*), photoactivation of PL$^{GABA}$ neurons (mDlx-ChR2, Shapiro–Wilk normality test, all p's < 0.05, see *Source data 1*) did not alter freezing (*Figure 7L*, Wilcoxon test, $W$ = 18, laser off vs. laser on, p = 0.156; Mann–Whitney test, $U$ = 14 Control vs. mDlx-ChR2, p > 0.999), avoidance (*Figure 7M*, Wilcoxon test, $W$ = 4, laser off vs. laser on, p = 0.500; Mann–Whitney test, $U$ = 12 Control vs. mDlx-ChR2, p = 0.772) time in food area (*Figure 7N*, Wilcoxon test, $W$ = −2, laser off vs. laser on, p = 0.750; Mann–Whitney test, $U$ = 11 Control vs. mDlx-ChR2, p = 0.660) rewarded lever presses (*Figure 7O*, Wilcoxon test, $W$ = 3, laser off vs. laser on, p = 0.500; Mann–Whitney test, $U$ = 13.5 Control vs. mDlx-ChR2, p > 0.999) and latency to press the lever

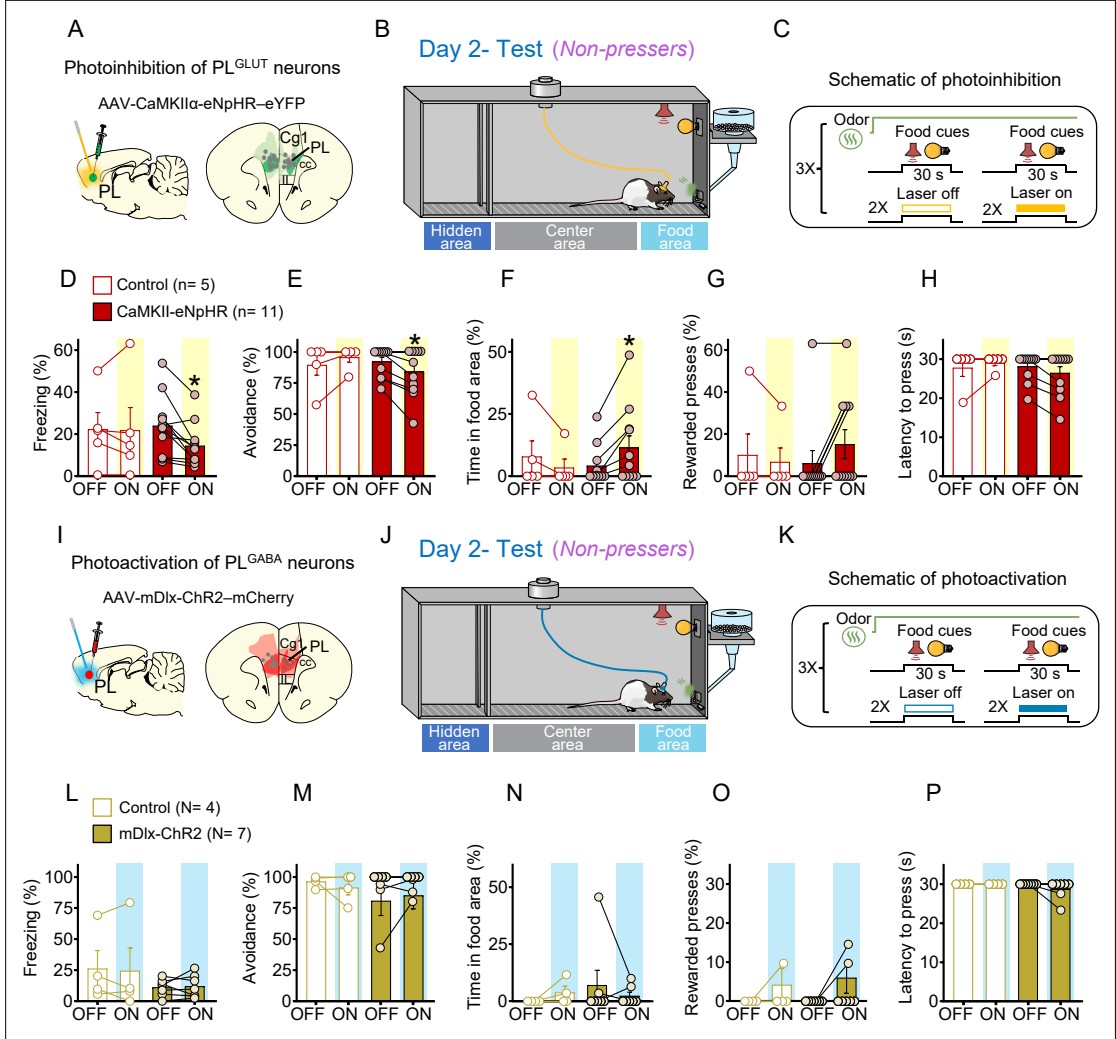

**Figure 7.** Photoinhibition of PL^GLUT neurons during conflict reduces freezing and increases food approaching in *Non-pressers*. (**A**) Schematic of AAV-CaMKII-eNpHR-eYFP virus infusion in prelimbic (PL) and location of optical fibers (gray dots) in the same region (compressed across different anteroposterior levels of PL). Green areas represent the minimum (dark) and the maximum (light) viral expression into the PL. (**B, C**) Schematic and timeline of the approach–avoidance conflict test during optogenetic inhibition of PL^GLUT neurons. Photoinhibition of PL^GLUT neurons (CaMKII-eNpHR, red bars, n = 11) during the conflict test reduced the percentage of time rats spent (**D**) freezing (Wilcoxon test, W = −64, laser off vs. laser on, p = 0.0020, Mann–Whitney test, U = 18 Control vs. CaMKII-eNpHR, p = 0.319) and (**E**) avoiding the odor area (Wilcoxon test, W = −21, laser off vs. laser on, p = 0.031; Mann–Whitney test, U = 19.5 Control vs. CaMKII-eNpHR, p = 0.365), and increased the percentage of time rats spent in the (**F**) food area (Wilcoxon test, W = 21, laser off vs. laser on, p = 0.031; Mann–Whitney test, U = 17 Control vs. CaMKII-eNpHR, p = 0.221) during the conflict test without altering (**G**) the number of lever presses (Wilcoxon test, W = 6, laser off vs. laser on, p = 0.250; Mann–Whitney test, U = 22.5 Control vs. CaMKII-eNpHR, p = 0.697) and (**H**) the latency to press (Wilcoxon test, W = −10, laser off vs. laser on, p = 0.125; Mann–Whitney test, U = 21 Control vs. CaMKII-eNpHR, p = 0.357). Laser stimulation did not alter behaviors in controls (eYFP-control virus, white bars, n = 5, Wilcoxon test, freezing: W = 3, p = 0.812, avoidance: W = 3, p = 0.500, food area: W = −3, p = 0.500, lever presses: W = −1, p = 0.999, latency to press: W = 1, p = 0.999). For all Shapiro–Wilk normality test, p < 0.05. (**I**) Schematic of AAV-mDlx-ChR-mCherry virus infusion in PL and location of optical fibers (gray dots) in the same region (compressed across different anteroposterior levels of PL). Red areas represent the minimum (dark) and the maximum (light) viral expression into the PL (**J, K**) Schematic and timeline of the approach–avoidance conflict test during optogenetic activation of PL^GABA neurons. (**L–P**) Photoactivation of PL^GABA neurons during the conflict test did not alter rats' behavior in the mDlx-ChR2 group (gold bars, n = 7) or in the control group (eYFP-control virus, white bars, n = 4, Wilcoxon and Mann–Whitney tests, all p's > 0.05). For all Shapiro–Wilk normality test, p < 0.05. PL neurons were illuminated from cue onset until the animals pressed the lever or from cue onset until the end of the 30 s cues if the animals did not press the lever (PL^GLUT: 5 Hz, PL^GABA: 20 Hz; 5 ms pulse width,7–10 mW). Data shown as mean ± standard error of the mean (SEM). Each bar represents the average of six trials alternated in blocks of 2. All *p's < 0.05. All statistical analysis details are presented in *Source data 1*. See also *Figure 7—figure supplements 1 and 2*.

The online version of this article includes the following source data and figure supplement(s) for figure 7:

**Source data 1.** Source data for *Figure 7*.

*Figure 7 continued on next page*

*Figure 7 continued*

**Figure supplement 1.** Photoinhibition of PL^GLUT neurons did not affect cued food-seeking responses in a neutral context.

**Figure supplement 1—source data 1.** Source data for *Figure 7—figure supplement 1*.

**Figure supplement 2.** Photoinhibition of PL^GABA neurons in Pressers does not alter defensive responses and food seeking during conflict.

**Figure supplement 2—source data 1.** Source data for *Figure 7—figure supplement 2*.

(*Figure 7P*, Wilcoxon test, $W = -3$, laser off vs. laser on, p = 0.500; Mann–Whitney test, $U = 10$ Control vs. mDlx-ChR2, p = 0.490), when compared to the control group (eYFP-control virus, white bars, $n = 4$, Wilcoxon test, freezing: $W = -2$, laser off vs. laser on, p = 0.875; avoidance, $W = 0$, p > 0.999, time in food area: $W = 3$, p = 0.500, rewarded lever presses: $W = 1$, p = 0.500, latency to press the lever: all animals reached maximum latency). Photoactivation of PL^GABA neurons in *Pressers* also did not affect defensive responses and food-seeking behavior during the conflict test (*Figure 7—figure supplement 2A–H*, Repeated measures ANOVA, all p's > 0.05, see *Source data 1*). Taken together, these results demonstrate that reduced activity in PL^GLUT neurons during conflict situations decreases defensive responses and biases rats' behavior toward food seeking.

## Discussion

Using a novel approach–avoidance conflict test, we identified two distinct behavioral phenotypes during the combined presentation of reward- and threat-paired cues: (1) rats that continued to press a lever for food (*Pressers*), and (2) rats that exhibited a complete suppression in food-seeking responses (*Non-pressers*). Single-unit recordings revealed that PL neurons regulate reward-approach vs. threat-avoidance responses during situations of uncertainty, when rats use previously associated memories to guide their decisions. We observed that increased risk-taking behavior in *Pressers* was associated with a larger number of food-cue responses in PL neurons, which showed sustained excitatory activity that persisted during the conflict phase, when compared to *Non-pressers*. In addition, PL^GLUT neurons showed reduced spontaneous activity during risky reward seeking and photoactivation of these cells in a neutral context was sufficient to suppress lever-press responses. Accordingly, photoinhibition of PL^GLUT neurons at the onset of the food cues in *Non-pressers* reduced defensive responses and increased food-approaching during the conflict phase, consistent with our observation that a small fraction of PL neurons changed their activity at the onset of freezing, avoidance, or risk-assessment responses. Altogether, these results suggest that under memory-based conflict situations, reduced or increased activity in PL^GLUT neurons can favor the behavioral expression of food-approaching or threat-avoidance responses, respectively.

During our approach–avoidance conflict test, *Pressers* and *Non-pressers* showed similar levels of lever pressing before the conflict phase (e.g., cued food-seeking training, threat conditioning, and reward phases). This observation suggests that these two individual phenotypes most likely emerged during the test session and were independent of prior differences in sucrose preference or food-seeking motivation. Similarly, because both groups exhibited the same percentage of freezing to the shock-paired odor during the olfactory threat conditioning session, the increased defensive behaviors and the reduced food-seeking responses observed in *Non-pressers* during the test session were unlikely due to higher acquisition of conditioned threat responses. Furthermore, other internal factors such as shock sensitivity or pain tolerance cannot be accounted for the individual differences observed in our experiments because both groups reacted equally to the unconditioned stimulus (i.e., velocity measured as maximum speed after the footshocks) and, different from other conflict tasks using footshocks as a punishment during the conflict test (*Geller, 1960*; *Vogel et al., 1971*; *Oberrauch et al., 2019*), in our model rats were not exposed to footshocks during the conflict phase. Therefore, the most plausible interpretation for the behavioral differences observed in our task is that *Pressers* and *Non-pressers* have allocated distinct motivational significance to the food- or shock-paired cues during the test session.

Individual differences in risky decision-making have also been reported in other studies using rodent models of behavioral conflict involving footshock punishment (*Simon et al., 2009*; *Jean-Richard-Dit-Bressel et al., 2019*; *Bravo-Rivera et al., 2021*), reversal learning (*Bari et al., 2010*), or variations in reward probability (*Ainslie, 1975*; *St Onge and Floresco, 2009*; *Dellu-Hagedorn et al.,*

2018), although the neural mechanisms underlying such differences are less clear. Evidence indicates that some of the neurobiological bases of individual variation in stimulus–reward response depend on differences in dopamine levels in subcortical circuits (*Tomie et al., 2000*; *Flagel et al., 2007*; *Flagel et al., 2011*), which are regulated by top-down mechanisms involving the mPFC (*Ferenczi et al., 2016*; *Haight et al., 2017*; *Serrano-Barroso et al., 2019*). Accordingly, our neural correlate analyses of risk-taking vs. risk-avoiding behaviors in the PL subregion of the mPFC revealed some clear differences between the two phenotypes, suggesting that PL neurons participate in behavioral selection when rats' decision depends on the conflicting memories of reward and threat. Both *Pressers* and *Non-Presses* showed a reduction in the number and magnitude of food-cue responses from reward to conflict phases, indicating that PL neurons can differentiate between situations involving motivational conflict and those that do not.

One intriguing finding in our study was the observation that *Pressers* showed a larger number of sustained excitatory food-cue responses during the conflict phase, when compared to *Non-pressers*. Because PL neurons are known for encoding the value of reward-predictive cues (*Sharpe and Killcross, 2015*; *Otis et al., 2017*), the increase in the number and magnitude of food-cue responses observed in *Pressers* might result in a greater allocation of attention to reward cues, which would explain the persistent reward-seeking responses observed in this group during motivational conflict. In support of this interpretation, reward-paired cues can acquire motivational salience in some subjects and become sufficient to elicit reward-seeking responses in both rodents (*Robinson and Flagel, 2009*; *Robinson et al., 2014*) and humans (*Smith et al., 2011*; *Jensen and Walter, 2014*). Consistently, *Pressers* also showed a larger number of food-cue responses in PL before the conflict phase (i.e., reward phase), although the percentage of rewarded presses and the latency to press the lever during the reward phase were similar between the two groups.

Another possible interpretation for the differences in food-cue responses in *Pressers* and *Non-pressers* is the reduced excitatory food-cue responses in *Non-pressers,* which may be mediated by cue-evoked activity in inhibitory inputs to PL during the conflict phase. While the source of this inhibition is unclear, a potential candidate are GABAergic neurons in the ventral tegmental area (VTA^GABA), which correspond to 35% of the cells in this region and send significant projections to PL (*Nair-Roberts et al., 2008*; *Breton et al., 2019*). Previous studies have shown that VTA^GABA neurons change their firing rates in response to reward-predicting cues (*Cohen et al., 2012*), and chemogenetic activation of these cells suppress the activity of local dopaminergic neurons (*van Zessen et al., 2012*), reduces cue-evoked sucrose-seeking responses (*Wakabayashi et al., 2019*), and induces conditioned place aversion in rodents (*Tan et al., 2012*). Future studies need to determine whether this regulation of rewarding and aversive responses by VTA^GABA neurons can also be attributed to their long-range inhibitory projections to PL neurons, particularly during conflict situations.

Differences in risk-taking and risk-avoiding behaviors were also reflected on LFP frequencies in PL neurons in the beginning of the test session, with *Pressers* and *Non-pressers* displaying increased power in the delta or theta bands, respectively. These findings are in corroboration with previous studies showing that increased delta power activity in the mPFC is associated with both reward seeking and preparatory attention (*Horst and Laubach, 2013*; *Totah et al., 2013*; *Emmons et al., 2016*), whereas augmented theta power in the mPFC or synchronized theta activity between mPFC and BLA is correlated with the expression of avoidance responses or the consolidation of threat memories, respectively (*Popa et al., 2010*; *Padilla-Coreano et al., 2019*). More specifically, increased synchrony between mPFC and BLA activity in the theta frequency range has been reported for animals that successfully differentiate between aversive and safe cues (or environments) during a differential threat conditioning task (or an open field arena) (*Likhtik et al., 2014*; *Stujenske et al., 2014*). In addition, prior studies have shown that 4 Hz LFP oscillations in the mPFC and BLA were strongly synchronized during conditioned freezing episodes (*Courtin et al., 2014*; *Dejean et al., 2016*; *Karalis et al., 2016*), and these sustained 4 Hz oscillations in the mPFC were independent of hippocampal low-theta oscillations, suggesting that they were internally generated in the mPFC during the expression of freezing behavior (*Karalis et al., 2016*). Consistent with these findings, in our study *Non-pressers* showed increased theta activity and marked 4 Hz oscillations in PL neurons, which were associated with better discrimination between reward and threat cues and increased freezing responses during the test session, when compared to *Pressers*.

Increased risk-taking behavior in *Pressers* was associated with a higher number of PL[GLUT] neurons showing reduced spontaneous activity during the conflict phase. In contrast, risk-avoiding responses in *Non-pressers* were associated with increased spontaneous activity during conflict. While this set of results suggest that distinct patterns of PL activity are associated with risk-taking or risk-avoiding behaviors in conflict situations, our optogenetic manipulation provided a causal role for PL[GLUT] neurons in the regulation of approach–avoidance conflict. For instance, the reduction in food-seeking responses during photoactivation of PL[GLUT] neurons indicates that increased activity in PL pyramidal cells is sufficient to recapitulate the reward-seeking suppression observed during conflict. Our findings agree with previous studies showing that increased activity in mPFC neurons, including PL, attenuates reward-seeking responses in a neutral context (*Berglind et al., 2007*; *Chen et al., 2013*; *Ferenczi et al., 2016*; but see *Warthen et al., 2016*), an effect that has been attributed, at least in part, to downstream projections to the paraventricular nucleus of the thalamus (PVT) (*Otis et al., 2017*). Notably, PVT neurons are necessary for the retrieval of both reward- and threat-associated memories (for a review see *Do Monte et al., 2016*; *Millan et al., 2017*; *McGinty and Otis, 2020*; *Penzo and Gao, 2021*), and activity in PVT neurons has recently been shown to be associated with the regulation of approach–avoidance responses during situations of conflict (*Choi and McNally, 2017*; *Choi et al., 2019*; *Engelke et al., 2021*), suggesting a potential target by which PL glutamatergic neurons may exert their effects.

Considering that *Pressers* showed a higher number of sustained excitatory food-cue responses than *Non-pressers*, it is counterintuitive that photoactivation of PL[GLUT] neurons during the food-cue onset resulted in reduced food-seeking responses. However, it is important to note that our optogenetic manipulation not only altered the activity of food-cue responsive neurons, but mostly the global activity of other PL[GLUT] neurons. Thus, it is possible that increased activity in the firing rate of PL[GLUT] neurons may result in reduced signal-to-noise ratio during the food-cue onset (*Kroener et al., 2009*; *McGinley et al., 2015*), and consequently decreased food-seeking responses. In contrast, we speculate that by reducing their spontaneous firing rates during conflict situations, PL[GLUT] neurons become more likely to fire in response to food cues due to an increase in the signal-to-noise ratio, thereby resulting in persistent reward-seeking responses during the conflict phase as we propose in our schematic in *Figure 8*.

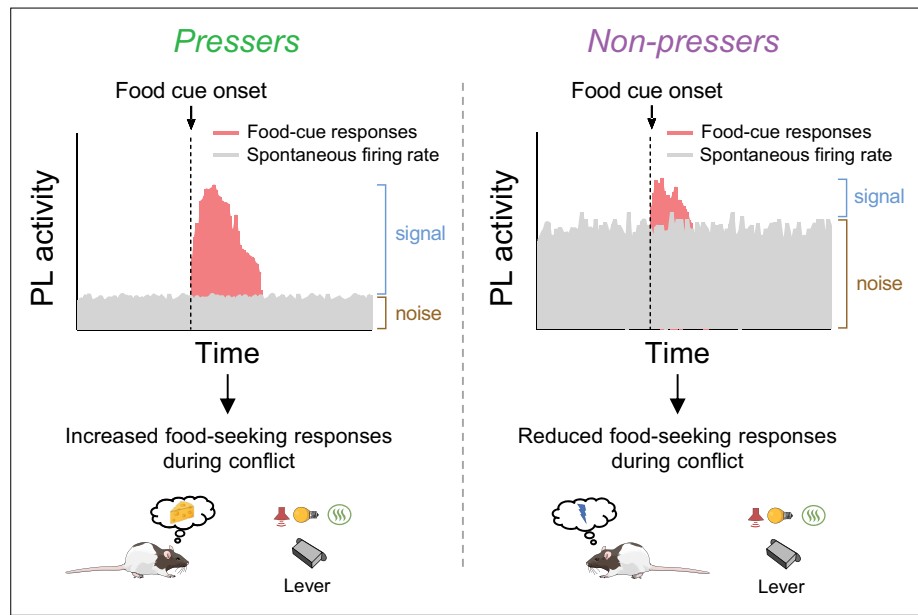

**Figure 8.** Schematic showing differences in food-cue responses and spontaneous firing rate of prelimbic (PL neurons in *Pressers* and *Non-pressers*). Left, *Pressers* showed reduced spontaneous firing rate and increased food-cue responses in PL neurons during the conflict test, which may have resulted in higher signal-to-noise ratio and increased food-seeking responses. Right, *Non-pressers* showed increased spontaneous firing rate and reduced food-cue responses in PL neurons during the conflict test, which may have resulted in lower signal-to-noise ratio and reduced food-seeking responses.

Additionally, our findings showing that inactivation of PL^GLUT neurons increases food-approaching responses in *Non-pressers* suggest that PL activity is indispensable to inhibit reward pursuit in the presence of threat-associated cues. The lack of effects on lever pressing indicates that other parallel brain regions may be modulating the suppression of operant lever-press responses during conflict. Alternatively, photoinhibition of PL^GLUT neurons was not large enough to produce a more global effect on risky behavior (i.e., completely restore lever presses). Collectively, these results add to a growing literature indicating that PL neurons are necessary to guide appropriate food-seeking behavior in tasks that rely on discrimination among environmental cues (*Marquis et al., 2007*; *Sangha et al., 2014*; *Moorman and Aston-Jones, 2015*) or decision-making tasks involving risk of punishment in which animals need to (1) adapt choice behavior during shifts in risk contingencies (*Orsini et al., 2018*), (2) regulate behavioral flexibility (*Radke et al., 2015*; *Capuzzo and Floresco, 2020*), or (3) suppress reward seeking in response to conditioned aversive stimuli (*Kim et al., 2017*; *Piantadosi et al., 2020*). Moreover, our results are in accordance with previous findings demonstrating that inactivation of PL neurons, or their inputs from BLA, increases risk-taking behavior in a conflict task in which rats needed to refrain from consuming sucrose to avoid a footshock (*Burgos-Robles et al., 2017*; *Verharen et al., 2019*).

Previous studies have shown that PL neurons fire in response to shock-paired cues and such activity is highly correlated with the expression of freezing responses (*Burgos-Robles et al., 2009*; *Sotres-Bayon et al., 2012*; *Kim et al., 2013*; *Courtin et al., 2014*). Adding to these findings, our recordings demonstrated that the activity of a small number of PL neurons changed immediately before or after the onset of freezing responses, with the same proportion of freezing-responsive cells being classified as PL^GLUT or PL^GABA neurons (~6–14%). At first sight, the lack of effects on freezing behavior following optogenetic activation of PL^GLUT neurons seems at odds with our recordings. It also seems to disagree with previous studies showing that electrical stimulation or optogenetic induction of 4 Hz oscillations in PL increases conditioned freezing responses (*Vidal-Gonzalez et al., 2006*; *Courtin et al., 2014*) by synchronizing the neural activity between PL and BLA regions (*Karalis et al., 2016*). However, one important difference between our study and others is that photoactivation of PL^GLUT neurons in our experiments was performed in naive rats, in the absence of shock-paired cues. Thus, the increased freezing responses following PL activation reported in previous studies appear to be dependent on the preexistence of a conditioned threat memory.

Overall, our results outline the neural correlates of risk-taking and risk-avoiding behaviors in PL and reveal an important role for PL^GLUT neurons in coordinating memory-based risky decision-making during conflict situations. Further studies will focus on identifying the PL downstream/upstream circuits that regulate reward-approaching and threat-avoidance responses, as well as the potential genetic and epigenetic factors that could contribute to the observed behavioral phenotypes. Elucidating the underlying mechanisms that mediate risk-taking vs. risk-avoiding responses during situations of uncertainty may help to provide understanding of response selection and adaptive behaviors, and may have clinical relevance to many psychiatric disorders (*Aupperle and Paulus, 2010*; *Kirlic et al., 2017*). Whereas persistent avoidance of presumed threats is the cardinal symptom of anxiety disorders (*Treanor and Barry, 2017*), seeking reward despite negative consequences is a hallmark of both eating and substance use disorders in humans (*Volkow et al., 2012*).

## Materials and methods

### Key resources table

| Reagent type (species) or resource | Designation | Source or reference | Identifiers | Additional information |
|---|---|---|---|---|
| Strain, strain background (rat, male) | *Rattus norvegicus* | Charles River | Strain code: 006 | Male Long–Evans hooded |
| Recombinant DNA reagent | AAV-CaMKIIα-eNpHR-eYFP | University of North Carolina Viral Vector Core | AAV Stock Vector Karl Deisseroth | 3rd Gen Opto Inhibition: eNpHR 3.0 |
| Recombinant DNA reagent | AAV-CaMKIIα-hChR2(H134R)-eYFP | University of North Carolina Viral Vector Core | AAV Stock Vector Karl Deisseroth | Opto Excitation: ChR2 |
| Recombinant DNA reagent | AAV-CaMKIIα-eYFP | University of North Carolina Viral Vector Core | AAV Stock Vector Karl Deisseroth | Control Fluorophores |

*Continued on next page*

*Continued*

| Reagent type (species) or resource | Designation | Source or reference | Identifiers | Additional information |
|---|---|---|---|---|
| Recombinant DNA reagent | pAAV-mDlx-ChR2-mCherry-Fishell-3 | Addgene and University of North Carolina Viral Vector Core | Addgene plasmid #83,898 | Packaged by UNC vector core (Serotype 5) |
| Antibody | Anti-GAD67 raised in mouse polyclonal | Millipore-Sigma | Cat No. MAB5406B | 1:400 |
| Antibody | Goat Anti-Mouse IgG H&L (Alexa Fluor 488) polyclonal | Abcam | Cat No. ab150113 | 1:200 |
| Antibody | Goat Anti-Mouse IgG H&L (Alexa Fluor 594) polyclonal | Abcam | Cat No. ab150116 | 1:200 |
| Other | Opal 520 | Akoya Biosciences | Cat No. FP1487001KT | 1:1000 Fluorescent dye |
| Other | Opal 620 | Akoya Biosciences | Cat No. FP1495001KT | 1:1000 Fluorescent dye |
| Sequence-based reagent | RNA scope probe mCherry | Advanced Cell Diagnostics | Cat No. 431201-C3 | |
| Sequence-based reagent | RNA scope probe vGAT | Advanced Cell Diagnostics | Cat No. 424,541 | |
| Commercial Assay or Kit | RNA scope Multiplex Fluorescent Detection Kit v2 | Advanced Cell Diagnostics | Cat No. 323,110 | |
| Chemical compound, drug | VECTASHIELD Antifade Mounting Medium with DAPI | Vectorlabs | Cat No. H-1200–10 | |
| Software, algorithm | Matlab | Mathworks | RRID:SCR_001622 | https://it.mathworks.com/products/matlab.html |
| Software, algorithm | NeuroExplorer | Plexon | RRID:SCR_001818 | https://plexon.com/products/neuroexplorer/ |
| Software, algorithm | Offline Sorter | Plexon | RRID:SCR_000012 | https://plexon.com/products/offline-sorter/ |
| Software, algorithm | CinePlex Behavioral Research System | Plexon | | https://plexon.com/plexon-systems/cineplex-behavioral-research-system/ |
| Software, algorithm | GraphPad Prism | GraphPad | RRID:SCR_002798 | https://www.graphpad.com/scientific-software/prism/ |

## Animals

All experimental procedures were approved by the Center for Laboratory Animal Medicine and Care of The University of Texas Health Science Center at Houston. The National Institutes of Health guidelines for the care and use of laboratory animals were strictly followed to minimize any potential discomfort and suffering of the animals. Male Long–Evans hooded adult rats (Charles Rivers Laboratories) with 3–5 months of age and weighing 300–450 g at the time of the experiment were used. Rats were single housed and after a 3-day acclimation period handled and trained to press a lever for sucrose as described below. Animals were kept in a 12 hr light/12 hr dark cycle (light from 7:00 to 19:00) and maintained on a restricted diet of 18 g of standard laboratory rat chow provided daily at the end of experimentation. Animals were given ad libitum access to water. Animals' weights were monitored weekly to ensure all animals maintained their weight under food restriction. During pre- and postsurgery phases, animals were given ad libitum access to food for a total of 7 days.

## Surgeries

Rats were anesthetized with 5% isoflurane in an induction chamber. Animals were positioned in a stereotaxic frame (Kopf Instruments) and anesthesia was maintained with 2.5% isoflurane delivered through a facemask. A heating pad was positioned below the body of the animal and both temperature and respiration were monitored during the entire surgery. Veterinary lubricant ointment was applied on the eyes to avoid dryness during the surgery. Animals received a subcutaneous injection of the local anesthetic bupivacaine (0.25%, 0.3 ml) at the incision site. Iodine and ethanol (70%) were alternately applied for asepsis of the incision site. The surgery procedures varied according to the type of implantation/injection (see below). For injection-only surgeries, the incision was stitched after the injection by using surgical suture (Nylon, 3–0). For implantation surgeries, the implants were fixed to the skull using C&B metabond (Parkell), ortho acrylic cement, and four to six anchoring screws. After surgery, animals received a subcutaneous injection of meloxicam (1 mg/kg) and a topical triple antibiotic was applied to the incision area.

## Viral vector injection

Viral injections were performed using a microsyringe (SGE, 0.5 µl) with an injection rate of 0.04 µl/min plus an additional waiting time of 12 min to avoid backflow. The adeno-associated virus (AAV) was bilaterally injected at a volume of 0.4 µl per side. The AAV-CaMKIIα-eNpHR-eYFP vector was used to inhibit glutamatergic neurons, whereas AAV-mDlx-ChR2-mCherry or AAV-CaMKIIα-ChR2-eYFP vectors were used to activate either GABAergic or glutamatergic neurons, respectively. The use of mDlx or CaMKIIα promoters enabled transgene expression favoring either GABAergic or glutamatergic neurons, as previously shown (*Gradinaru et al., 2009*; *Tye et al., 2011*; *Dimidschstein et al., 2016*; *Sun et al., 2020*) and was confirmed by our immunohistochemical and RNAscope assessment (*Figure 5—figure supplement 1*). The viral construct AAV-CaMKIIα-eYFP was used to control for any nonspecific effects of viral infection or laser heating. All plasmid or viral vectors were obtained from Addgene or University of North Carolina Viral Vector Core. For implantation of optrodes, the following coordinates from bregma were used for virus injection: PL, +2.7 mm AP, ±0.7 mm ML, −3.8 mm DV at a 0° angle. For PL soma illumination, an optical fiber (0.39 NA, 200 nm core, Inper) was implanted in each hemisphere targeting PL neurons using the following coordinates from bregma: +2.7 mm AP, ±1.5 mm ML, −4.0 mm DV at a 15° angle.

## Single-unit electrodes

An array of 16 or 32 microwires was unilaterally implanted targeting the PL using the following coordinates from bregma: +2.7 mm AP, ±0.8 mm ML, −3.9 mm DV. Three different electrode configurations were used: (1) 32-channel silicon probes (Buzsaki32-CM32 or A1 × 32-5 mm-25-177-CM32, Neuro Nexus Technologies, USA), (2) Micro-Wire Arrays of 16 or 32 channels (Bio-Signal Technologies Ltd, USA); or (3) custom designed electrodes with 2 × 8 grid with 150 µm of space between wires, 200 µ of space between rows, with 35 µm diameter wires (Innovative Neurophysiology Inc, USA). For photoidentification of GABAergic or glutamatergic neurons, a Hermes 32 channels optrode array was used (200 nm core, Bio-Signal Technologies Ltd). Optrodes were unilaterally implanted at the same coordinates described above after the infusion of 0.6 µl of AAV-mDlx-ChR2-mCherry or AAV-CaMKIIα-ChR2-eYFP vectors. In all cases, the ground wire was wrapped around a grounding screw previously anchored into the skull. Two insulated metal hooks were implanted bilaterally into the cement to allow firmly attachment of the array connector to the cable during recording.

## Odor preparation

A 99% amyl acetate solution (Sigma-Aldrich) was diluted in propylene glycol (Bluewater Chemgroup, Inc) to a 10% solution and presented to the rats during the different stages and phases of the olfactory threat conditioning test. A customized olfactometer (Med Associates) was used to control the flow of air into the animal's chamber. Before being mixed with the 10% amyl acetate solution, the air was passed through a desiccant and a charcoal filter to remove any moisture and odors, and was finally rehydrated with distilled water before being delivered into the chamber through a thermoplastic PVC-based tube (Tygon) attached to an odor port located in the odor area.

## Behavioral tasks

### Lever-press training

Rats were placed in a plexiglass, standard operant box (34 cm high × 25 cm wide × 23 cm deep, Med Associates), and trained to press a lever for sucrose on a fixed ratio of one pellet for each press. Next, animals were trained in a variable interval schedule of reinforcement that was gradually reduced across the days (one pellet every 15, 30, or 60 s) until the animals reached a minimum criterion of 10 presses/min. All sessions lasted 30 min and were performed on consecutive days. Sucrose pellet delivery, variable intervals, and session duration were controlled by an automated system (ANY-maze, Stoelting). Lever-press training lasted approximately 1 week, after which animals were assigned to surgery or cued food-seeking training. A small number of rats failed to reach the lever press criteria and were excluded from the experiments (<3%).

### Cued food-seeking training

Rats previously trained to press a lever for sucrose were trained to learn that each lever press in the presence of an audiovisual cue (tone: 3 kHz, 75 dB; light: yellow, 2.8 W; 30-s duration) resulted in the

delivery of a sucrose pellet into a nearby dish. Reward cue conditioning also took place in the standard operant boxes. While the light cue helps to direct the animals toward the lever during the beginning of the training phase, the tone assures that the animals will not miss the presentation during the trial and provides the temporal precision required for single-unit recordings. After ~4 consecutive days of training (24 trials per day, pseudorandom intertrial interval of ~120-s, 60-min session), rats learned to discriminate the food-associated cue as indicated by a significant increase in press rate during the presence of the audiovisual cues, when compared to the 30 s immediately before the cue onset (cue-off, see *Figure 1—figure supplement 1A*). The cued food-seeking training was completed when animals reached 50% of discriminability index (presses during cue-on period minus presses during cue-off period divided by the total number of presses).

After the cued food-seeking training was completed, rats with single-unit electrodes were exposed to an additional training session in which the audiovisual cue ceased immediately after the animals pressed the lever and a single sucrose pellet was delivered into the dish. This extra training reduced the rat's response to a single press and dish entry per cue, thereby enabling us to correlate each food-seeking event with the neuronal firing rate by avoiding overlapping between consecutive events (e.g., lever presses). The single-pellet training took place in the same plexiglass rectangular arena subsequently used for the odor test (40 cm high × 60 cm wide × 26 cm deep, Med Associates, see schematic in *Figure 1A*, right). The arena consisted of a hidden area (40 cm high × 20 cm wide × 26 cm deep) separated from an open area by a plexiglass division. An 8-cm slot located in the center of the division enabled the animal to transition between both sides of the arena. For behavioral quantification, the open area was subdivided into a center area and a food area (40 cm high × 12 cm wide × 26 cm deep), the latter containing a lever, a dish, and an external feeder similar to the food-seeking operant box.

## Habituation day

Animals were placed in the odor arena and exposed to 12 audiovisual cues (30-s duration, pseudorandom intertrial intervals of between 25 and 40 s) followed by 10 min of presentation to the neutral odor alone (10% amyl acetate) and an additional 12 audiovisual cues similar to the first cues but in the presence of the neutral odor delivered in the food area of the arena. Each lever press in the presence of the audiovisual cue resulted in the delivery of a sucrose pellet into the dish, and the audiovisual cue ceased immediately after the animal pressed the lever.

## Threat conditioning day

One day after the habituation day, rats were placed in a plexiglass, standard operant box similar to the cued food-seeking training box, but with the grid floor previously attached to a shock generator system. Rats were habituated to one nonreinforced odor presentation (10% amyl acetate, 30-s duration) followed by five odor presentations that coterminated with a foot shock (0.7 mA, 1-s duration, 258–318-s intertrial intervals). An olfactometer system was used to deliver the odor into the box (see Odor preparation session), whereas an exhaustor system was used to remove it during the intertrial intervals. Between each odor presentation, audiovisual cues (30-s duration) signaling the availability of sucrose were presented to the animals. Each lever press during the audiovisual cues resulted in the delivery of a sucrose pellet into the dish. Shock grids and floor trays were cleaned with 70% ethanol between each rat. No rats were excluded from the analyses due to distinct levels of freezing following the threat conditioning session.

## Test day

One day after the threat conditioning session, rats were returned to the same arena used during the habituation and exposed to the exact same protocol. The first phase of the test session was called *reward phase* (12 min) and the animals were exposed to 12 food cues. The second phase was called *odor phase* and the animals were exposed to 10 min of conditioned odor (10% amyl acetate) alone. The last phase was called *conflict phase* (12 min) and the animals were exposed to 12 food cues in the presence of the conditioned odor. An odor dispersion sensor (200B miniPID, Aurora Scientific) revealed that the odor took approximately 2.21 ± 0.28 s to reach detectable concentrations (56 particles per billion; *Punter, 1983*) in the arena after the olfactometer onset, and approximately 19.59 ± 0.97 s to be completely removed from the arena after the olfactometer offset and concomitant

activation of the exhaustor fan. Due to the low temporal resolution to control the delivery of the odor in the arena, the odor was maintained constant during the entire duration of the odor and conflict phases. In order to press the lever for sucrose during the conflict phase, rats had to approach the conditioned odor presented in the food area. After the end of the conflict phase, the odor was extracted from the arena with the exhaustor fan and the floor and walls of the arena were cleaned thoroughly with 70% ethanol solution.

### Behavioral tracking

Both the standard operant boxes and the testing arenas were equipped with video cameras and a behavior tracking software (ANY-maze, Stoelting) which were used to record the animal's behavior and control the delivery of sucrose, foot shock, tone, light, and odor in the apparatuses. Avoidance responses were characterized by the time spent in the hidden area of the arena. Freezing responses were characterized by the complete absence of movements except those needed for respiration. Risk-assessment responses were characterized by a body stretching movement to peep out toward the food area while in the hidden area and were used as a measure of risk-assessment behavior (*Blanchard et al., 2011*).

For single-unit recording analyses, the detection of freezing, avoidance, and risk assessment behaviors were performed using the open source tool DeepLabCut, a machine learning software that tracks complex patterns of behavior from videos (*Mathis et al., 2018*). After a video has been analyzed, the data were saved to a.csv file that contained the *x* and *y* location of each rat's body part in pixels, as well as the analysis of the expected accuracy (i.e., likelihood) of the tracked positions across time. After DeepLabCut has calculated the positions and the likelihood, we used three different Python codes to identify each one of the three behaviors. For freezing behavior, the code used DeepLabCut's position data and determined if the rat was still for more than 500 ms. The animal was considered to be still if the position in question was within 1.05 pixels of each other. For avoidance behavior, the code used DeepLabCut's position data to determine the location of the rat in the arena and based on the center of its head to identify when the animals entered the hidden area of the arena. Finally, for risk assessment behavior, the code used DeepLabCut's position data to identify the nose, ears, center of the head, and spine to determine whether the rat was located in the hidden area of the arena with its body stretched and the head looking through the open division of the apparatus. Each of these codes generated a.xlsx file that contained the onset and the total duration of each behavioral episode (see Single-unit analyses for more details).

### Optogenetic stimulation during behavior

Bilateral optical cables (200 μm core, 0.37 NA, 2.5 mm ceramic ferrule, Inper) were connected to a blue laser (diode-pumped solid-state, 473 nm, 150 mW output, OptoEngine) or a yellow laser (diode-pumped solid-state, 593.5 nm, 300 mW output, OptoEngine) by using a patch cord (200 μm, 0.39 NA, FC/PC connector, Inper) through a dual rotary joint (200 μm core, Doric lenses). During the stimulation, the optical cables were coupled to the previously implanted optical fibers by using a ceramic sleeve (2.5 mm, Precision Fiber Products). An optogenetic interface (Ami-2, Stoelting) and an electrical stimulator (Master 9, A.M.P. Instruments) were used to control the onset of the laser, pulse width, train duration, and frequency. The power density estimated at the tip of the optical fiber was 7–10 mW for illumination of PL somata (PM-100D, Power Energy Meter, Thor Labs).

### Single-unit recording

A 64-channel neuronal data acquisition system (Omniplex, Plexon) integrated with a high-resolution video-tracking system (Cineplex, Plexon) was used for electrophysiological recordings from freely behaving animals. Both videos and neuronal recordings were combined within the same file, thereby facilitating the correlation of behavior with neuronal activity. An electrical isolation, Faraday cage was made and connected to the grounding port of the data acquisition system. The system was connected to the head-mounted electrode/optrode by using a digital headstage cable (32 channels, Plexon), a motorized carrousel commutator (Plexon), and a digital headstage processor (Plexon). Rats were habituated to the headstage cable daily for approximately 1 week before the beginning of the experiments. Extracellular waveforms exceeding a voltage threshold were band-pass filtered (500–5000 Hz), digitized at 40 kHz, and stored onto disk. Automated processing was performed using

a valley-seeking scan algorithm and then visually evaluated using sort quality metrics (Offline Sorter, Plexon, see Single-unit analyses).

## Photoidentification of PL neurons during recordings

During neuronal photoidentification, we recorded from rats expressing channelrhodopsin (ChR2) in PL neurons previously implanted with an optrode in the same region. An optical cable connected to a blue laser was attached to the headstage cable and coupled to the previously implanted optical fiber by using a ceramic sleeve. At the end of the behavioral session, 10 trains of 10-s blue laser pulses (5 ms pulse width, 5 Hz) were delivered by a Master-9 programmable pulse stimulator, which also sent flags to the data acquisition system to mark the time of the laser events.

Neurons were considered to be responsive to photoactivation if they showed a significant increase in firing rate above baseline (20 ms, Z-score >3.29, p < 0.001) and higher reliability within the 6 ms after laser, similar to previous studies (*Lima et al., 2009*; *Pi et al., 2013*; *Burgos-Robles et al., 2017*; *Engelke et al., 2021*). To identify the threshold separation for the frequency distribution of response latencies to laser illumination, we implemented the triangle method detection (*Zack et al., 1977*). This calculation is particularly effective for left-skewed distributions as in our sample. We considered bins with response latency values from 0 to 12 ms and excluded those with larger values as they would most likely reflect indirect stimulation via collateral activity. We computed the distance normal to the line along with the minimum and maximum values in the histogram. The threshold was defined as the maximum distance between the histogram and the line (i.e. a normalized level within 0 and 1), which in our analysis resulted in 5.8 ms (rounded to 6 ms bin). In addition, to measure the reliability of neural responses to photoactivation, we calculated the Fano factor (FF), defined as the variance-to-mean ratio of spike counts (*Churchland et al., 2011*), to characterize the variability of neuronal responses 6 ms before (FF before) and 6 ms after (FF after) the laser pulses for each train of illumination (10 trains of 50 laser pulses, 200-ms pulse interval). When the variance in the counts equals the mean count, FF was equal to 1. Afterwards, we computed the 'overall FF ratio' between the 'FF after' divided by the 'FF before' to compute the reliability of each cell to the laser onsets. Only neurons showing an overall FF higher than 1, which indicates reliable responses to laser illumination compared to baseline, were included as photoidentified cells.

A small number of laser-generated photoelectric artifacts (~10% of the channels in less than 10% of the rats) were observed during the photoactivation. However, they were easily distinguished from the action potentials by their descending voltage signals of high amplitude, pulse shapes distinct from the regular waveforms, isolated spatial distribution in the PCAs, and occurrence restricted to the period of laser activation, which resulted in lack of activity during the behavioral session.

## Optogenetic manipulation of PL neurons during behavior

During the cued food-seeking test, rats expressing ChR2 or eNpHR in PL were bilaterally illuminated in the same region by using a blue (5-ms pulse width, 5 Hz for CaMKIIα or 20 Hz for mDlx) or a yellow laser (constant illumination), respectively. The laser was activated at cue onset and persisted throughout the entire 30 s of the audiovisual cue presentation. Rats were exposed to two consecutive cues with laser off followed by 2 consecutive cues with laser on in a total of 12 cues (pseudorandom intertrial intervals of between 25 and 40 s). To assess the effects of PL illumination on rat's defensive behavior, PL neurons of rats expressing ChR2 or eNpHR were bilaterally illuminated during six distinct epochs of 30 s during the *odor phase* by using a blue (5-ms pulse width, 20 Hz) or a yellow laser (constant illumination), respectively. To assess the effects of PL illumination on food-seeking responses during the conflict phase, rats were exposed to two consecutive cues with laser off followed by 2 consecutive cues with laser on in a total of 12 cues (pseudorandom intertrial intervals of between 25 and 40 s). The laser was activated at cue onset and persisted on until the animal pressed the bar or the 30 s of the audiovisual cue was completed.

## Histology

Animals were transcardially perfused with KPBS followed by 10% buffered formalin. Brains were processed for histology as previously described (*Do-Monte et al., 2013*). At the end of the recording sessions, a microlesion was made by passing anodal current (0.3 mA for 15 s) through the active wires to deposit iron in the tissue. After perfusion, brains were extracted from the skull and stored in a 30% sucrose/ 6% ferrocyanide solution to stain the iron deposits. Only rats with the presence of eYFP or

mCherry labeling and the track of the electrode wires or optical fiber tips located exclusively in PL were included in the statistical analyses.

## Immunohistochemistry

Rats previously infused with AAV-mDlx-ChR2-mCherry or AAV-CaMKIIα-ChR2-eYFP were transcardially perfused with 300 ml of KPBS followed by 500 ml of 4% paraformaldehyde. Brains were removed from the skull, transferred to a 20% sucrose solution in KPBS for 24 h, and stored in a 30% sucrose solution in KPBS for another 24 hr. Next, coronal PL sections (40 µm thick) were cut in a cryostat (CM 1860, Leica), blocked in 20% normal goat serum and 0.3% Triton X-100 in KPBS at room temperature for 1 hr. For identification of GABAergic neurons, PL sections were incubated with anti-GAD67 serum raised in mouse (1:400; Millipore-Sigma) at 4°C for 48 hr. After sections were washed in KPBS for five times, sections were incubated with a secondary anti-mouse antibody (1:200, Alexa Fluor 488 or Alexa Fluor 594, Abcam) for 2 hr. Sections were washed with KPBS, mounted in Superfrost Plus slides, and coverslipped with antifading mounting medium (Vectashield, Vectorlabs). Images were generated by using a microscope (Nikon, Eclipse NiE Fully Motorized Upright Microscope) equipped with a fluorescent lamp (X-Cite, 120 LED) and a digital camera (Andor Zyla 4.2 PLUS sCMOS).

## In situ hybridization

Single molecule fluorescent in situ hybridization (RNAscope Multiplex Fluorescent Detection Kit v2, Advanced Cell Diagnostics) was used following the manufacturer protocol for fixed-frozen brains sample. Brain samples were sectioned at a thickness of 20 µm in a cryostat (CM1860, Leica). Sections were collected onto superfrost plus slides (Fisher Scientific) and transferred to a −80°C freezer. To prepare for the assay, brain sections were serially dehydrated with EtOH (50%, 75%, and 100%, each for 5 min) and then incubated in hydrogen peroxide for 10 min. Target retrieval was performed with RNAscope target retrieval reagents at 99°C for 5 min. The sections were then pretreated with Protease III (RNAScope) for 40 min at 40°C. RNAscope probes (Advanced Cell Diagnostics) for mCherry (Cat No. 431201-C3) and vGAT (Cat No. 424541) were hybridized at 40°C for 2 hr, serially amplified, and revealed with horseradish peroxidase, Opal Dye/TSA Plus fluorophore (Akoya Biosciences), and horseradish peroxidase blocker. Sections were coverslipped with antifading mounting medium with DAPI (Vectashield, Vectorlabs) and kept in the refrigerator. Images were generated by using an epifluorescent microscope (Nikon, Eclipse NiE Fully Motorized Upright Microscope) equipped with a fluorescent lamp (X-Cite, 120 LED) and a digital camera (Andor Zyla 4.2 PLUS sCMOS). Expression of mCherry mRNA (red, Opal 620) and GAD67 mRNA (green, Opal 520) was determined by using an automated fluorescent threshold detector (NIS-Elements). Colabeled cells were manually counted by an experimenter by measuring either the percentage of mCherry-positive neurons in PL that were also labeled with GAD67, or the percentage of GAD67-positive neurons in PL that were also labeled with mCherry.

## Data analyses

### Behavioral quantification and statistical analysis

Rats were recorded with digital video cameras (Logitech C920) and behavioral responses were measured by using an automated video-tracking system (ANY-maze) or machine learning (DeepLabCut). Presses per minute were calculated by measuring the number of presses during the 30 s cue multiplied by two. All graphics and numerical values reported in the figures are presented as mean ± standard error of the mean. Given that the different phases of the test have different duration, we have normalized the data in percentage to be able to compare the behavior of the animals across the different phases of the test. Shapiro–Wilk normality test was performed before all the statistical analyses to determine parametrical or nonparametrical statistical tests, and reported in Results when the values showed nonnormal distribution. For normal data, statistical significance was determined with paired Student's $t$-test, Welch's $t$-test, repeated measures ANOVA followed by Bonferroni post hoc comparisons (Prism 7), and $Z$-test or Fisher Exact Test, as indicated in Results, figure legends, and *Source data 1*. For nonparametric data, Wilcoxon test (paired groups) or Mann–Whitney test (unpaired groups) w performed. Effects sizes for pairwise comparisons or ANOVA were calculated by using Cohen's $d$ or partial eta squared, respectively (*Cohen, 1988*; *Richardson, 2011*). Effect sizes <0.2 were considered small, effects sizes between 0.2 and 0.8 were considered medium, and effect

sizes > 0.8 were considered large (*Cohen, 1988*). Sample size (*n*) was based on estimations by power analysis with a level of significance of 0.05 and an expected effect size of 0.8.

## PCAs on behaviors

The PCA method (*Jolliffe, 2002*) was implemented to further understand the relationships between approaching, avoidance, freezing, and latency to press during the conflict phase for both *Pressers* and *Non-pressers*. This method was used to reduce the dimensionality of our multivariate data while preserving as much of the relevant information as possible. Briefly, we performed a linear transformation of the data to a new coordinate system such that the new set of variables, the principal components, were linear functions of the original variables. These variables were also uncorrelated, and the greatest variance by any projection of the data came to lie on the first coordinate, the second greatest variance on the second coordinate, and so on. Afterwards, we analyzed the explained variance (%) of the first two PCs (i.e., PC1 and PC2) together with the relevance of each analyzed behavior to define which variables better explained the differences between *Pressers* and *Non-pressers* (see Results for additional details).

## Single-unit analyses

Single units were selected based on three principal components and waveform features such as valley-to-peak and amplitude measurements. The principal component scores for unsorted waveforms were computed and plotted in a three (or two)-dimensional principal component space. Clusters containing similar valid waveforms were manually defined. After manually clustering similar valid waveforms, a group of spikes were considered from a single neuron if the waveforms formed a discrete, isolated, cluster in the principal component space. A Commercial software (NeuroExplorer, NEXT Technologies) combined with Matlab (MathWorks) scripts, and Python scripting were implemented to calculate the spontaneous firing rate, changes in neural activity in response to food cues, lever presses, food dish entries, as well as the neural correlates of freezing, avoidance, and risk-assessment behaviors. The spontaneous firing rate was calculated by comparing the frequency of spike trains during the last 30 s of the food-seeking phase, odor phase, or conflict phase against the 30 s prior to the beginning of each session. Food cue, lever press, and food dish responses were calculated by implementing Matlab scripts as *Z*-scores normalized to 20 precue bins of 300 ms. Neurons showing a *Z*-score >2.58 (p < 0.01) during the first two bins following the onset of the aligned event were classified as excitatory responses, whereas neurons showing a *Z*-score <−1.96 (p < 0.05) during the same first two bins were classified as inhibitory responses. A temporal frequency separation was used to classify the food-cue responses according to the pattern of activity, similar to a previous study (*Bezdudnaya et al., 2006*). Neurons showing a transient increase in firing rate (<600-ms duration) were classified as transient activity, whereas neurons showing a sustained increase in firing rate (≥900-ms duration) were classified as sustained activity (*Z*-score >2.58 during the first 3 s after food-cue onset).

To analyze freezing, avoidance, and risk-assessment responses, the time onsets for each behavior were filtered by selecting only the events that lasted more than 1 s and were not preceded by the same behavior during the previous 6 s (baseline). The final list of time onsets was entered into the single-unit recording files to create the events and temporally align them with the neuronal recordings. To increase the number of events during our analyses, we combined the behavioral responses emitted during the odor and conflict phases. Only animals exhibiting at least six events for each behavior were included. We used an interval criterium of 600 ms to select the neurons that responded close in time to the onset of the analyzed behavior (before or after), thereby avoiding potential neural activity contamination caused by other types of behavioral responses.

## Acknowledgements

We thank Dr. Roger Janz for helping us with the packaging of the AAV-mDLX-ChR2-mCherry viral construct, Ryia Albert and Sharon Gordon for their technical assistance, and Dr. Robson Scheffer-Teixeira for his support with the statistical analyses. We also thank current and former members of the Do Monte and Quirk Labs for their valuable comments on the manuscript, and the Mind the Graph team for creating the schematic drawings presented in the manuscript. This work was supported by NIH grants R00-MH105549 and R01-MH120136, a Brain & Behavior Research Foundation grant (NARSAD Young Investigator), and a Rising STARs Award from UT System to FHD-M.

## Additional information

### Funding

| Funder | Grant reference number | Author |
|--------|----------------------|--------|
| National Institute of Mental Health | R00-MH105549 | Fabricio H Do Monte |
| National Institute of Mental Health | R01-MH120136 | Fabricio H Do Monte |
| Brain and Behavior Research Foundation | NARSAD Young Investigator | Fabricio H Do Monte |
| University of Texas Health Science Center at Houston | Rising STARs Award | Fabricio H Do Monte |

The funders had no role in study design, data collection, and interpretation, or the decision to submit the work for publication.

### Author contributions

Jose A Fernandez-Leon, Data curation, Formal analysis, Investigation, Methodology, Validation, Visualization, Writing – original draft, Writing – review and editing; Douglas S Engelke, Conceptualization, Data curation, Formal analysis, Investigation, Methodology, Project administration, Supervision, Validation, Visualization, Writing – original draft, Writing – review and editing; Guillermo Aquino-Miranda, Formal analysis, Investigation, Methodology, Validation, Visualization, Writing – original draft, Writing – review and editing; Alexandria Goodson, Formal analysis, Methodology, Software; Maria N Rasheed, Methodology; Fabricio H Do Monte, Conceptualization, Data curation, Formal analysis, Funding acquisition, Investigation, Methodology, Project administration, Resources, Supervision, Validation, Visualization, Writing – original draft, Writing – review and editing

### Author ORCIDs

Jose A Fernandez-Leon http://orcid.org/0000-0001-7166-9738
Douglas S Engelke http://orcid.org/0000-0002-3962-5003
Guillermo Aquino-Miranda http://orcid.org/0000-0001-6185-4112
Fabricio H Do Monte http://orcid.org/0000-0002-1079-0064

### Ethics

All experimental procedures were approved by the Center for Laboratory Animal Medicine and Care of The University of Texas Health Science Center at Houston. All of the animals were handled according to approved Institutional Animal Care and Use Committee (IACUC) protocols (AWC-19-0103). The National Institutes of Health guidelines for the care and use of laboratory animals were strictly followed to minimize any potential discomfort and suffering of the animals.

### Decision letter and Author response

Decision letter https://doi.org/10.7554/eLife.74950.sa1
Author response https://doi.org/10.7554/eLife.74950.sa2

## Additional files

### Supplementary files

• Transparent reporting form
• Source data 1. Statistical Analyses Table 1.

### Data availability

All data generated or analyzed during this study are included in the manuscript and supporting file; Source Data files have been provided for all main figures and supplementary data. We also have included detailed statistical analyses in Source data 1.

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
