## [Editor Report]

This paper offers a novel behavioural perspective showing how opposing motivational states interact to influence behaviour differentially across individuals. It uses a variety of cutting-edge tools to dissect the microcircuits of the prefrontal cortex. This report is novel, timely, and important. It will be of broad interest to neuroscientists studying fear, reward, motivation, and decision making and is relevant to understanding neural processes in stress and anxiety-related disorders.

---

## [Decision Letter]

**Decision letter after peer review:**

[Editors’ note: the authors submitted for reconsideration following the decision after peer review. What follows is the decision letter after the first round of review.]

Thank you for submitting the paper "Neural correlates and determinants of approach-avoidance conflict in the prelimbic prefrontal cortex" for consideration by *eLife*. Your article has been reviewed by 3 peer reviewers, one of whom is a member of our Board of Reviewing Editors, and the evaluation has been overseen by a Senior Editor. The following individual involved in review of your submission has agreed to reveal their identity: Shelly B Flagel (Reviewer #2).

Comments to the Authors:

After consultation between the editors and the reviewers, we have decided that this work, in its present form, will not be considered further for publication by *eLife*. During the review process a number of concerns were raised (see appended reviews for details) regarding the statistics, the lack of direct comparisons between the Pressers and Non-Pressers, the behavioural designs and the conclusions regarding the neural findings. These concerns require further analyses that may substantially change the message of the findings. Additional data may also be necessary to address some of the concerns. That said, the editors and reviewers recognized the importance and potential impact of the findings and would be willing to reconsider the paper for publication provided it was adequately revised to address each reviewer concerns. If the authors decided to chose this option, the manuscript will be treated as a new submission and must be accompanied by a point-by-point response to review. Although we will endeavor to secure the reviewers, we cannot guarantee this.

*Reviewer #1:*

The paper by Fernandez-Leon examined the role of PL glutamate and GABAergic neurons during a conflict-based behavioural task. The task consisted of lever press during an audio-visual compound in the presence of an aversively conditioned odour. The behavioural data indicated that two cohort of animals were generated – pressers and non-pressers. Pressers continued to press the lever (reward-seek) in the presence of the aversively conditioned odour (albeit to a lesser degree) whereas the non-pressers ceased pressing. Single unit recordings revealed a reduction in the number of food-cue responsive neurons under conflict (compared to no conflict). Different subsets of PL neurons were shown to signal freezing, avoidance and risk-assessment during conflict. The data show reduced spontaneous activity in PL glutamatergic neurons when animals lever press under conflict. Activation of these neurons using ChR2 under the control of the CaMKII promotor attenuated food-seeking behaviour in a neutral context in pressers. Inhibiting the same neurons in non-pressers reduced defensive behaviours often seen to cues conditioned with shock and increased food-based conditioned behaviours.

The strengths of the paper are numerous and include the novel behavioural design that pits reward up against aversion. Examining the distinct conflict phenotypes throughout the paper was also excellent. The integration of single-cell recordings, LFPs, and optogenetics were considerable strengths allowing to dissect the glutamatergic vs GABAergic microcircuits in the PL during this behaviour. The discussion of the results in the context of the existing literature was excellent.

Despite the clear strengths of the paper, some weaknesses exist. A closer examination of the single units is warranted. The claim that putative classification of PL neurons into glutamatergic and GABAergic based on waveform and spike timing given the optotagging results seems premature. The optotagging analysis needs additional data including an eYFP control to show what, if any, effect light stimulation alone has on neural responses. Some consideration of whether the two behavioural phenotypes are due to differences under conflict or due to perception is also needed.

Additional neural analyses are necessary to understand what the PL neurons are doing:

1) For example in Figure 2 panels E, G, H,O, Q, R – the signal seems to be present at food cue onset but it is the sustained portion of that signal that is different between the pressers and non-pressers.

2) Some single unit analyses are needed here that better capture the individual profile of neurons. The current population based analyses are not sufficient to understand if there is any heterogeneity in the neural firing. For example the heatplots suggest that some neurons show the sustained firing but others do not. It may be important to pull those out as they may be doing different things across the different patterns of stimulation (reward cues only, odour, conflict).

3) A statistical analysis of whether these proportions can be obtained by chance is needed.

Optotagging studies require additional investigation to justify the strong claims:

1) I wasn't convinced by the claim that the optotagging proves that waveform and spike timing analyses are inappropriate. I believe this requires more extensive examination or should be toned down or removed from the paper.

2) The 6ms and 12ms optotaging cut off seemed to be quite long and additional justification is needed. A cluster analysis of the optotagged cells and their responses may help to see how responses are distributed in time as well as a response reliability analysis (as per Lima et al., 2009). This may need to be coupled with pharmacology showing which signals are synaptic and which directly induced by the light.

3) The optotagging studies did not include an eYFP control and to ensure that there are no light artifacts (which I have seen). This control is needed.

4) It is unclear why stimulation with ChR2 using the Dlx promotor does not lead to excitation of GABAergic neurons. This requires clarification.

The reward-cues were trained as discriminant in an operant and this is different to the Pavlovian associations in the aversive conditioning. To what extent do the neural response reflect this distinction and do the authors expect that a similar neural profile would be seen if the reward cues were trained in a Pavlovian procedure?

To what extent could the distinct of the pressers vs non-pressers be due to individual differences in the relative salience of the odour vs audio-visual cues rather conflict?

The data in Figure 7 suggest that only two/three rats were tested in some conditions (e.g., panels O, P). If this is the case then the study is underpowered and more data need to be collected, if that is because the points overlap, then the authors should represent the circle as adjacent so that the reader and see the n clearly from the figure.

*Reviewer #2:*

This manuscript includes a series of studies to assess the role of prelimbic neurons in mediating behavior during an approach-avoidance conflict task. The authors used a novel task to assess the ability of rats to remember cues previously associated with either reward (food) or threat (footshocks) to make a behavioral decision. In doing so, they uncovered two behavioral phenotypes: "Pressers", who continued to press a lever for food during conflict; and "Non-Pressers", who exhibited a suppression of food-seeking behavior in face of conflict. A combination of optogenetics and single-unit recordings were used to assess the neural mechanisms underlying this individual variability in reward-seeking behavior during conflict. The authors report that increased risk-taking behavior in "Pressers" is associated with reward-cue-elicited responses in the prelimbic cortex and reduced spontaneous activity in prelimbic glutamatergic neurons during conflict. Further, activation of prelimbic glutamatergic, but not GABAergic, neurons attenuated reward-seeking responses selectively in "Pressers"; and inhibition of prelimbic glutamatergic neurons increased reward-approach behavior and decreased freezing behavior during conflict in "Non-Pressers".

These experiments were well-designed, the methods were appropriate to address the questions at hand, and the manuscript is well-written. The ethologically-relevant approach-avoidance task is novel and will be of interest to the field. In particular, the ability to capture distinct behavioral phenotypes and individual differences using this test will allow further investigation of the neural determinants of reward-seeking and threat-avoiding behavior during conflict.

As currently presented, there are some concerns regarding the statistical analyses and whether they support all of the authors' claims. As the individual differences component of the manuscript is particularly novel and of interest, it is a bit concerning that these analyses include a sample size of 25 "Pressers" and 7 "Non-Pressers". In relation, it is not clear that the neural responses of these two behavioral phenotypes were ever directly compared. For example, in Figure 2 and Supplementary Figure 2, the area under the curve for neural responses during reward and conflict are presented independently for the two phenotypes and direct comparisons to assess group differences and/or interactions are not apparent. Similarly, it is not clear why data only from "Non-Pressers" is shown in Figure 7, as the methods suggest that both "Pressers" and "Non-Pressers" were used for this experiment. Further, in general, it is difficult to deduce which statistical analyses support the claims made in the manuscript text, as the analyses are only presented in the Figure legends and in Source data 1 and don't always seem congruent.

The two behavioral phenotypes that are reported are novel and of great potential interest, thereby warranting more detailed analyses and description. For one, the characterization of the behavioral phenotypes is not described in detail in the methods or data analyses. Was this based only on the percentage of rewarded presses during the conflict phase? What it a median split based on this value? Is the distribution of "Pressers" and "Non-Pressers" consistent across experiments, as it seems to be skewed towards "Pressers" in the current data set. In addition, while these phenotypes that emerge during conflict do appear to be independent of other behavior in the current study, additional analyses, like principal components analysis, would help support this claim by showing whether there is a relationship (e.g. clustering or reduction of variables) between certain behavioral responses and whether this relationship differs between the phenotypes. That is, such an analysis would allow one to better determine if this behavioral outcome measure is indeed independent of others.

Given the skewed sample sizes, it would be good to include effect sizes to further support the claims made based on these data.

It is quite difficult to determine which statistical effects are being reported in the Figure Legends and how they correspond to those presented in Supplementary Source data 1 and the claims made in the text. It would be beneficial to put supporting statistics following claims made in the primary text.

In relation to the point above, justification is needed for the Wilcoxon Test. While it seems to be appropriate for some datasets, for others it is questionable. For example, for the data presented in Figure 7, it would be important to report interactions between control and eNpHR groups for laser on vs. laser off conditions.

What is the justification for showing almost all behavioral data as percentages rather than raw values of time or other metrics?

Figure 5 legend, panel O – Fisher's exact test for the Reward Phase seems to be lacking.

*Reviewer #3:*

Fernandez-Leon et al. investigate the role of the pre-limbic area (PL) in regulating approach avoidance behavior in situations of learned motivational conflict where animals experience both cues that predict an aversive outcome as well as cues that signal the availability of food. This region has been implicated in threat responding and food seeking separately but has not previously been examined in situations of conflict. The authors employ an individual differences approach, subdividing animals based on their food seeking behavior in the presence of conflicting cues that signal food availability and footshock and use a combination of in vivo recordings and optogenetic manipulations to identify a role for specific cell types in the PL in regulating risky behaviors in aversive contexts. This manuscript adds to the growing literature on neural mechanisms of processing approach-avoidance conflict.

This work has many strengths. Examining approach and avoidance in a conflict paradigm, rather than separately, provides a more ethological study of the neural basis of these behaviors as, beyond the confines of a laboratory, action selection commonly occurs in the face of multiple competing cues. Subdividing animals into 'pressers' and 'non-pressers' based on individual differences in engagement in food seeking behavior is an excellent strategy to gain insight into the behavioral function of these cells. Recognizing that not engaging in food seeking does not necessarily reflect failure to complete the task but rather a bias toward avoidance behavior is insightful and important. The authors suggest a number of interesting and potentially important differences in PL neural activity between pressers and non-pressers. For example, pressers (i.e. rats that continue to seek food in the presence of an aversive cue) have both more food-cue responsive neurons and greater magnitude of excitatory and inhibitory responses to food-cues, a difference that is sustained when food-cues are presented in the presence of an aversive cue. Pressers and non-pressers also had marked differences in oscillatory frequency, an intriguing finding that warrants further investigation. Optogenetic experiments nicely establish causality with precise temporal resolution.

The design of the behavioral paradigm somewhat limits the ability the ability to draw certain conclusions. During testing, food-cues were presented discretely while the shock-cue was constant preventing direct comparison of responding to appetitive and aversive cues that would have been highly interesting. Furthermore, during the test session, reward cues are always presented first followed by the addition of the shock-cue. This, and the extended shock-cue presentation under extinction conditions makes it difficult to entirely rule out alternative interpretations for differences between pressers and non-pressers, for example, more rapid extinction of fear memory in pressers than non-pressers. Beyond this, the lack of direct statistical comparison of neural activity in pressers and non-pressers undermines the strength of the central conclusions of this paper.

The authors hypothesize that stimulating glutamatergic PL neurons decreases signal to noise ratio between cells that are active during food seeking and those that are not, thus resulting in a decrease in food-seeking. This is interesting and plausible proposal to be further explored in future research.

I have a number of concerns about the behavioral paradigm. Although both food and shock cues were presented as discrete 30s cues during conditioning, in the test phase, the shock associated cue was presented as a constant cue, first alone for 10 minutes and then with discrete 30s food associated cues overlayed. No rationale is provided for this specific design. I struggle to think of benefits for such a design compared to all discrete cue presentations, and see clear disadvantages. For example, it is not possible to compare responding to food and shock cues and the sustained exposure to the shock cue may result in extinction. Further, the cues are in different modalities and not counterbalanced across animals: the food cue is always a tone & light and the shock cue is always odor. The manuscript also does not provide definitive behavioral evidence to show learning of the shock-odor association. Although for food-seeking discrimination between cue on and cue off periods is shown, the same is not shown for fear conditioning. In the absence of an unconditioned stimulus (CS-), it would be good to at least show discrimination from the pre-cue period (or an equivalent time where there is no food seeking or shock cue present) when comparing pressers and non-pressers in order to confirm that freezing and defensive behaviors are specific to learned odor association and not a generalized fear response. This is especially important given that lever pressing during food cues during the threat conditioning day is suppressed, indicating that the animals show a generalized increase in freezing.

Given the sustained exposure to the shock cue under extinction conditions during test, a potential interpretation of the difference between pressers and non-pressers is that pressers may extinguish faster than non-pressers leading to behavioral differences in the odor phase. Enhanced extinction could reflect differences in underlying learning that are occluded on the conditioning day by high levels of freezing or an alteration in the mechanisms of extinction. It is notable that pressers and non-pressers do show differences in freezing on the test day. It would be good to look at freezing and pressing cue by cue as is done in Supplemental Figure 1C for the threat conditioning.

Wherever neural activity is compared between pressers and non-pressers, statistical testing should be conducted and reported. For example, the section starting on line 183 includes many statements comparing pressers and non-pressers with no statistical analysis to support this and Figure 1 only presents analyses within each group. If the authors wish to directly compare pressers and non-pressers, a more appropriate analysis (e.g. a two-way ANOVA) should be used.

Can the authors explain why they chose to stimulate PGLUT neurons in shock naïve animals? The paper is looking to show the involvement of these cells in approach avoidance in situations of conflict, but this design simply assesses food seeking.

Starting on line 578 the authors note that activation of PGLUT neurons decreases food seeking responses despite Pressers showing increased excitatory food cue responses. They hypothesize that optogenetic activation of PGLUT neurons also activates cells not active during food seeking, decreasing signal to noise during the cue and therefore decreases food seeking behavior (FIGURE 8). Although a reasonable hypothesis, there is no direct evidence provided to support this. This claim could be strengthened by showing the magnitude of excitatory and inhibitory responses in PGLUT neurons specifically rather than across PL neurons generally like in Figure 2 E/F and O/P.

The title of Figure 5 is misleading in its exclusive focus on the conflict phase. In fact, this is true for the reward only period also. This should be amended to more accurately convey the data presented in the figure. The associated text in the Results section (e.g. line 418-420) also presents the interpretation that increased firing of PLGLUT neurons associates with increased reward seeking during conflict. While strictly speaking this is not incorrect, not also discussing that the same effect is observed in the reward only period misconstrues this data by potentially suggesting an effect specific to conflict when in fact it would appear to be a general phenomenon reflecting reward seeking. The appropriate interpretation should be clarified throughout the manuscript.

In general, more nuanced analysis of the temporal dynamics of neural responses relative to behavior would be of interest and could more convincingly establish the link between neural activity and behavior.

[Editors’ note: further revisions were suggested prior to acceptance, as described below.]

Thank you for submitting your work entitled "Neural correlates and determinants of approach-avoidance conflict in the prelimbic prefrontal cortex" for further consideration by *eLife*. Your revised article has been evaluated by Kate Wassum (Senior Editor), a Reviewing Editor, and the original reviewers.

The reviewers and editors agreed that the authors have done a commendable job responding to prior concerns and suggestions. The amount of data and analyses included is impressive, and the responses were not only thorough but also thoughtful. In reading the manuscript, it was often difficult to recognize and appreciate the most meaningful effects. Additional clarity could be provided regarding statistical analyses included in the text and how they relate to the graphical illustrations, which, at times, seems incongruent. To help the authors revise their manuscript, the Reviewing Editor has outlined the key comments that need additional attention.

Essential revisions:

1) Regarding the neural analyses comment #3 by Reviewer 1, the question pertained to whether the number of neurons obtained for a given population differed from chance. This is not a question about whether neural modulation was different from baseline for the neurons. It's generally answered with a chi squared test of independence. Is the population of neurons obtained one that is different from a population that could be obtained by chance?

2) Some of the most interesting, innovative, and impactful aspects of this work is the behavioral characterization based on individual differences. For this reason, the authors should consider reframing the Discussion, starting with a description of the behavioral phenotypes and what they might be capturing before describing the neuronal findings.

3) Pg. 10; lines 243-251: The authors state that Pressers showed a higher number of food-cue responsive neurons than Non-Pressers during the reward phase, and that both Pressers and Non-Pressers showed a significant reduction in the number of food-cue responsive neurons during the conflict phase. In the latter case, it is unclear if the reduction from 33% to 14% in Pressers is significantly different from the reduction of 21% to 6% in Non-Pressers (i.e. is there an interaction with phenotype in this reduction). It seems that this would be the case and that it would be important to recognize, but it is written as though both show an equivalent reduction in the number of food-cue responsive neurons during conflict. Direct phenotype comparisons appear to be lacking in subsequent analyses of the same dataset, which limits the conclusions that can be drawn. In this regard, additional clarity is needed regarding what comparisons are being made and reported and what the authors can or cannot conclude based on these comparisons. Another example comes from the data described for the inset in Figure 2I and 2S, for which it is stated that "Pressers show a higher magnitude of inhibitory food-cue responses during the reward phase, and, in contrast to Non-Pressers, such responses were attenuated during the conflict phase". However, it is not clear from the statistics included or the text that the authors are directly comparing Pressers and Non-Pressers and/or which statistics speak to this comparison.

4) Clarification points:

i) Pg. 11, line 270: The authors should replace "the same number" with "the same percentage". This also raises the concern as to whether or not it is appropriate to compare percentages when the denominator is 237 in Pressers and 89 in Non-Pressers. That is, how does recording from 237 neurons vs. 89 neurons affect the interpretation of the results given that the likelihood of detecting differences is presumably greater with 237 neurons.

ii) Pg. 12, lines 293-297: Similar to the point above, is the Fisher's Exact Test reported here only comparing the proportions of neurons between phases and not between behavioral phenotypes?

iii) Figure 6M, 6N. It looks as though the significant effects of laser diminish over time for the CaMKII-ChR2 group. Can the authors comment on this in the manuscript?

iv) A description of the principal components analysis used to assess the behavioral data seems to be lacking in the primary text.

5) Additional analyses:

i) Please include insets of the AUC analyses reported in the text, this applies for 4B, C, D and E.

ii) Pg. 17, lines 405-412: It is stated that proportions of excitatory and inhibitory food cue response for PL(Glut) and PL(GABA) neurons were similar during the reward and conflict phase (Figure 5J-K). However, it does not appear that an analysis was conducted to compare proportions between different neuronal subtypes. That is, it seems like it would be meaningful to be able to say that the proportion of excited glutamatergic neurons in response to the food cue during the conflict phase was less than the proportion of GABAergic neurons excited during the conflict phase (should that prove to be the case). Again, as written, it is difficult to discern what analyses were or were not conducted and which are being described.

Please also include a Key Resources Table.

---

## [Author Response]

[Editors’ note: the authors resubmitted a revised version of the paper for consideration. What follows is the authors’ response to the first round of review.]

Comments to the Authors:After consultation between the editors and the reviewers, we have decided that this work, in its present form, will not be considered further for publication by eLife. During the review process a number of concerns were raised (see appended reviews for details) regarding the statistics, the lack of direct comparisons between the Pressers and Non-Pressers, the behavioural designs and the conclusions regarding the neural findings. These concerns require further analyses that may substantially change the message of the findings. Additional data may also be necessary to address some of the concerns. That said, the editors and reviewers recognized the importance and potential impact of the findings and would be willing to reconsider the paper for publication provided it was adequately revised to address each reviewer concerns. If the authors decided to chose this option, the manuscript will be treated as a new submission and must be accompanied by a point-by-point response to review. Although we will endeavor to secure the reviewers, we cannot guarantee this.Reviewer #1:The paper by Fernandez-Leon examined the role of PL glutamate and GABAergic neurons during a conflict-based behavioural task. The task consisted of lever press during an audio-visual compound in the presence of an aversively conditioned odour. The behavioural data indicated that two cohort of animals were generated – pressers and non-pressers. Pressers continued to press the lever (reward-seek) in the presence of the aversively conditioned odour (albeit to a lesser degree) whereas the non-pressers ceased pressing. Single unit recordings revealed a reduction in the number of food-cue responsive neurons under conflict (compared to no conflict). Different subsets of PL neurons were shown to signal freezing, avoidance and risk-assessment during conflict. The data show reduced spontaneous activity in PL glutamatergic neurons when animals lever press under conflict. Activation of these neurons using ChR2 under the control of the CaMKII promotor attenuated food-seeking behaviour in a neutral context in pressers. Inhibiting the same neurons in non-pressers reduced defensive behaviours often seen to cues conditioned with shock and increased food-based conditioned behaviours.The strengths of the paper are numerous and include the novel behavioural design that pits reward up against aversion. Examining the distinct conflict phenotypes throughout the paper was also excellent. The integration of single-cell recordings, LFPs, and optogenetics were considerable strengths allowing to dissect the glutamatergic vs GABAergic microcircuits in the PL during this behaviour. The discussion of the results in the context of the existing literature was excellent.

We are glad the reviewer recognizes the strengths of our work. Thank you!

Despite the clear strengths of the paper, some weaknesses exist. A closer examination of the single units is warranted. The claim that putative classification of PL neurons into glutamatergic and GABAergic based on waveform and spike timing given the optotagging results seems premature. The optotagging analysis needs additional data including an eYFP control to show what, if any, effect light stimulation alone has on neural responses. Some consideration of whether the two behavioural phenotypes are due to differences under conflict or due to perception is also needed.

We thank the reviewer for the thoughtful comments. Below we provide a detailed explanation about how we have addressed each one of these points.

Additional neural analyses are necessary to understand what the PL neurons are doing:1) For example in Figure 2 panels E, G, H,O, Q, R – the signal seems to be present at food cue onset but it is the sustained portion of that signal that is different between the pressers and non-pressers.

We thank the reviewer for the recommendation. We have followed the reviewer’s recommendation and performed some additional neuronal analyses to explore the dynamics of PL tone responses in *Pressers* vs. *Non-Pressers*. We have now separated the food cue responses into transient (< 600 ms) and sustained (≥ 900 ms) activity and quantified the number of neurons in each category. We observed that *Pressers* exhibited a higher number of sustained excitatory food cue responses during the reward (new Figure 2F vs. 2P) and the conflict phases (new Supp Figure 2 D vs. Supp Figure 2N). We have modified the Figures, the Methods, and the Results section accordingly. The new sentences in the Results section now reads:

“Using a temporal frequency separation of the food cue responses into transient (< 600 ms duration) and sustained (≥ 900 ms duration) activity (Bezdudnaya et al., 2006), we revealed that Pressers display a higher proportion of sustained excitatory responses during the reward phase, when compared to Non-Pressers (Figure 2F vs. 2P, Fisher Exact Test, p = 0.032).”

“…in Pressers, 42% of excitatory food-cue responses showed sustained activity during the conflict phase whereas such responses were completely absent in Non-Pressers (Supplementary Figure 2D vs. Supplementary Figure 2N, Fisher Exact Test, p = 0.0181).”

2) Some single unit analyses are needed here that better capture the individual profile of neurons. The current population based analyses are not sufficient to understand if there is any heterogeneity in the neural firing. For example the heatplots suggest that some neurons show the sustained firing but others do not. It may be important to pull those out as they may be doing different things across the different patterns of stimulation (reward cues only, odour, conflict).

Following the reviewer’s recommendation, we have performed some additional analyses to better identify the neural firing heterogeneity of PL neurons. In addition to the separation of food cue responses into transient and sustained activity as described above, we have analyzed how PL activity changes during lever presses and food dish entries and quantified the number of cells that responded to one or more of these events (see new Supplementary Figure 3). The new sentence in the Results section reads:

“In addition, we observed that a significant proportion of PL recorded neurons changed their firing rates in response to lever presses (23%) or rewarded food dish entries (16%, Supplementary Figure 3A-N). A longitudinal tracking of PL activity throughout the reward phase demonstrated that most PL responsive neurons changed their activities selectively to food cues, lever presses, or food dish entries, with a smaller number of cells responding during two or more of these events (Supplementary Figure 3O). This observation suggests that PL neurons exhibit a heterogeneous pattern of activity during reward-seeking behavior, consistent with a recent study using calcium imaging recordings from PL neurons in head-fixed mice (Grant et al., 2021).”

3) A statistical analysis of whether these proportions can be obtained by chance is needed.

We thank the reviewer for raising this point and we apologize for not being clear enough with the responsiveness criteria. For all the single-unit recording data, we applied a Z-score calculation using 20 bins as the baseline to define whether a neuron was responsive or not after the onset of an event or behavior, meaning that all neurons classified as responsive passed the significant criteria of Z = 2.58 (p < 0.01) for excitation and Z = -1.96 (p < 0.05) for inhibition, therefore not by chance. To clarify this point, we have improved the description in the Methods section and added the following sentence in the figure legends of all the single-unit recording figures: “The threshold used to identify significant differences per neurons was Z-score > 2.58 for excitation and Z-score < -1.96 for inhibition.”

In addition to the Z-score classification for individual cells, we have used the appropriate statistical analysis to compare differences between phases or groups, as indicated in the Results section, Figure Legends, Methods, and Supplementary Source data 1.

Optotagging studies require additional investigation to justify the strong claims:1) I wasn't convinced by the claim that the optotagging proves that waveform and spike timing analyses are inappropriate. I believe this requires more extensive examination or should be toned down or removed from the paper.

We have followed the reviewer’s recommendation and removed the comparison between optogenetic classification and putative analyses from the manuscript to avoid deviating the attention of the readers from the main focus of the study.

2) The 6ms and 12ms optotaging cut off seemed to be quite long and additional justification is needed. A cluster analysis of the optotagged cells and their responses may help to see how responses are distributed in time as well as a response reliability analysis (as per Lima et al., 2009). This may need to be coupled with pharmacology showing which signals are synaptic and which directly induced by the light.

We have followed the reviewer’s recommendation and performed additional analyses to clarify the response latency criteria used for the optogenetic classification of PL cells. Using the triangle method detection, we identified a cluster division in the histogram distribution at 6 ms after the laser onset (Figure 5C and 5G; see below the new description added to the Methods section). This 6 ms response latency adopted in our manuscript is similar or stricter than several previous studies using in vivo photoidentification in rodents:

Allsop et al. 2018, Cell (PMID: 29731170): < 8 ms for mPFC neurons

Beyeler et al. 2016, Neuron (PMID: 27041499): < 9 ms BLA-NAc or < 6 ms BLA-CeA

Nieh et al. 2015, Cell (PMID: 25635460): < 8 ms Lateral Hypothalamus

Burgos-Robles et al., 2017, Nature Neurosc (PMID: 28436980): < 12ms BLA-mPFC Kravitz et al. 2013 Brain Res (PMID: 23178332): < 15 ms dorsomedial striatum.

Cohen et al. 2012, Nature (PMID: 22258508): < 8 ms VTA

Keller et al. 2018, J Neurophys (PMID: 29589814): < 15 ms auditory cortex

Fadok et al. 2017, Nature (PMID: 28117439): < 10 ms central amygdala

Wolff et al. 2014, Nature (PMID: 24814341): < 7 ms central amygdala

After reading carefully the recommended Lima et al. 2009 article, we noticed that they have calculated spike reliability to laser illumination based on the 40 ms after laser onset, which could have resulted in the inclusion of spikes from neighboring cells that were indirectly activated by collateral projections. To avoid this possibility, we have limited our analyses to the first 6 ms after the laser onset and compared it with the 6 ms before by using a Fano factor ratio to estimate the spike reliability during these two windows of time (please see below the new detailed description added to the Methods section). For PL^GLUT^ neurons classification, 3 out of 39 PL neurons showing short latency responses of 6 ms failed the reliability test and were excluded from our analyses, resulting in 36 photoidentified PL^GLUT^ neurons. For the PL^GABA^ neurons classification, 7 out of 76 PL neurons showing short-latency responses of 6 ms failed the reliability test and were excluded from our analyses, resulting in 69 photoidentified PL^GABA^ neurons. Although the main findings and the interpretation of the behavioral tests did not significantly change after adopting this stricter classification, we have updated all the corresponding figures and statistical analyses in the manuscript to reflect the new number of photoidentified cells (see updated Figure 5, Supplementary Figure 7, Supplementary Figure 8).

Updated Methods section:

“To identify the threshold separation for the frequency distribution of response latencies to laser illumination, we implemented the Triangle Method detection (Zack et al., 1977). […] Only neurons showing an overall FF higher than 1, which indicates reliable responses to laser illumination compared to baseline, were included as photoidentified cells.”

3) The optotagging studies did not include an eYFP control and to ensure that there are no light artifacts (which I have seen). This control is needed.

Optogenetic-induced light artifacts was not a confounding in our photoidentification experiments for several reasons: (i) photoelectric artifacts were very rare (1 or 2 channels in less than 10% of the rats), most likely due to the large distance between the optical fiber tip and the contact area of the wires (~300 μm) with the Hermes 32 channels optrode arrays (Biosignal Technologies); (ii) the small number of photoelectric artifacts generated by laser activation were easily distinguished from the action potentials. They were characterized by descending voltage signals of high amplitude and pulse shapes visibly distinct from the regular waveforms, which resulted in different spatial distribution in the principal component analyses; and most importantly, (iii) the occurrence of photoelectric artifacts was restricted to the period of laser illumination and their frequency corresponded approximately to the number of laser pulses delivered during the optogenetic stimulation (~500). Because all photoidentification sessions occurred by the end of the behavioral test and were recorded within the exact same file, real spikes triggered by the laser populated a cluster of thousands of spikes previously recorded during the behavioral session. In contrast, the photoelectric artifacts resulted in a separate cluster with a much smaller number of “spikes” and absence of activity during the behavioral session. Please see the representative example in Author response image 1 for a comparison between a sorted photoidentified neuron (yellow, >18 thousand spikes recorded along the entire session) and a photoelectric artifact (green, 501 “spikes” recorded along the entire session but observed exclusively during the 10 trains of illumination) in the same channel.

**Author response image 1. sa2fig1:** 

Although the photoelectric artifact was not an issue during our experiments, we have followed the reviewer’s recommendation of testing a few control animals to completely rule out the possibility that the laser illumination could generate a signal that resembles an action potential, which would lead to an inaccurate classification of the cells. For this purpose, we used two control rats implanted with the same model of optrodes but without channelrhodopsin expression in PL. We then exposed the animals to the same behavioral and laser illumination protocol used in the previous experiments. We found that none of the 14 recorded cells show any type of response when aligned to the onset of the laser (see the PSTH graphics Author response image 2). We have now introduced a new paragraph in the Methods section to explain how we have distinguished the photoelectric artifacts.

4) It is unclear why stimulation with ChR2 using the Dlx promotor does not lead to excitation of GABAergic neurons. This requires clarification.

We apologize for not being clear in the results description. Although the photoactivation of PL^GABA^ neurons increases the firing rate of ChR2-expressing cells right after the pulses of illumination, it results in subsequent inhibition during the pulse

intervals. Whereas the increased firing triggered by the laser can be visualized by aligning the onset of each laser pulse in small bins of 1 ms (as shown in Figure 5H and in the representative example “aligned to pulses” Author response image 3), a global inhibitory effect is revealed by aligning the onset of each train and analyzing the entire duration of the train including the intervals between the pulses of illumination (as shown in Figure 6E-H and the representative example “aligned to train onset” Author response image 3) .

**Author response image 3. sa2fig3:** 

We have fixed the Results section to better reflect these findings. It now reads:

“…although some PL^GABA^ cells showed increased activity right after the laser onset (as shown in Figure 5E-H), illumination of PL^GABA^ somata reduced the firing rate of all responsive PL neurons when analyzing the entire duration of the train (16 out of 22 neurons, 73%; Figure 6F-H), indicating a suppression in local activity.”

The reward-cues were trained as discriminant in an operant and this is different to the Pavlovian associations in the aversive conditioning. To what extent do the neural response reflect this distinction and do the authors expect that a similar neural profile would be seen if the reward cues were trained in a Pavlovian procedure?

This is an interesting question that we have asked ourselves before. Because both *Pressers* and *Non-Pressers* were trained under the exact same protocols, the observed differences in PL firing rate between the two groups cannot be attributed to the operant vs. Pavlovian nature of the cues. Considering that the differences in food cue responses between *Pressers* and *Non-Pressers* were observed as soon as 300 ms after the food cue onset (much earlier than the animals moved to the food area to approach the lever), we predict that a similar distinction in PL activity would be also observed if the animals were trained using a Pavlovian conditioning test for the reward cues. This is a separate question that we are planning to explore in the future.

To what extent could the distinct of the pressers vs non-pressers be due to individual differences in the relative salience of the odour vs audio-visual cues rather conflict?

We have performed some additional analyses to investigate how *Pressers* and *Non-Pressers* responded to the audiovisual and olfactory stimuli before the conditioning session by analyzing the response latency to audiovisual cues, as well as the amount of time investigating the odor cue, during the pre-conditioning phase (before the first shock presentation; see new Supplementary Figure 1G-H). We did not observe any significant differences between the two groups, suggesting that *Pressers* and *Non-Pressers* attributed the same salience to the audiovisual and olfactory cues before the fear conditioning session. We therefore concluded that the two individual phenotypes emerged during the test day, particularly during the odor and conflict phases, as we have acknowledged in the manuscript.

The data in Figure 7 suggest that only two/three rats were tested in some conditions (e.g., panels O, P). If this is the case then the study is underpowered and more data need to be collected, if that is because the points overlap, then the authors should represent the circle as adjacent so that the reader and see the n clearly from the figure.

Thank you for raising this point and we apologize for not being sufficiently clear. We had a short note in the figure legends to indicate that the number of data points could appear fewer than the number of rats due to some overlapping in the values. However, we have now figured out a way to distribute the circles side by side in the graphic so that the readers can clearly see all the values in the figure.

Reviewer #2:This manuscript includes a series of studies to assess the role of prelimbic neurons in mediating behavior during an approach-avoidance conflict task. The authors used a novel task to assess the ability of rats to remember cues previously associated with either reward (food) or threat (footshocks) to make a behavioral decision. In doing so, they uncovered two behavioral phenotypes: "Pressers", who continued to press a lever for food during conflict; and "Non-Pressers", who exhibited a suppression of food-seeking behavior in face of conflict. A combination of optogenetics and single-unit recordings were used to assess the neural mechanisms underlying this individual variability in reward-seeking behavior during conflict. The authors report that increased risk-taking behavior in "Pressers" is associated with reward-cue-elicited responses in the prelimbic cortex and reduced spontaneous activity in prelimbic glutamatergic neurons during conflict. Further, activation of prelimbic glutamatergic, but not GABAergic, neurons attenuated reward-seeking responses selectively in "Pressers"; and inhibition of prelimbic glutamatergic neurons increased reward-approach behavior and decreased freezing behavior during conflict in "Non-Pressers".These experiments were well-designed, the methods were appropriate to address the questions at hand, and the manuscript is well-written. The ethologically-relevant approach-avoidance task is novel and will be of interest to the field. In particular, the ability to capture distinct behavioral phenotypes and individual differences using this test will allow further investigation of the neural determinants of reward-seeking and threat-avoiding behavior during conflict.

We are glad the reviewer recognizes the strengths of our work. Thank you!

As currently presented, there are some concerns regarding the statistical analyses and whether they support all of the authors' claims. As the individual differences component of the manuscript is particularly novel and of interest, it is a bit concerning that these analyses include a sample size of 25 "Pressers" and 7 "Non-Pressers". In relation, it is not clear that the neural responses of these two behavioral phenotypes were ever directly compared. For example, in Figure 2 and Supplementary Figure 2, the area under the curve for neural responses during reward and conflict are presented independently for the two phenotypes and direct comparisons to assess group differences and/or interactions are not apparent. Similarly, it is not clear why data only from "Non-Pressers" is shown in Figure 7, as the methods suggest that both "Pressers" and "Non-Pressers" were used for this experiment. Further, in general, it is difficult to deduce which statistical analyses support the claims made in the manuscript text, as the analyses are only presented in the Figure legends and in Source data 1 and don't always seem congruent.

We thank the reviewer for the thoughtful comments. Below we provide a detailed explanation about how we have addressed each one of these points.

The two behavioral phenotypes that are reported are novel and of great potential interest, thereby warranting more detailed analyses and description. For one, the characterization of the behavioral phenotypes is not described in detail in the methods or data analyses. Was this based only on the percentage of rewarded presses during the conflict phase? What it a median split based on this value?

To separate the two behavioral phenotypes, we have performed a binary classification of the animals based exclusively on the number of lever presses during the conflict phase. This was the only criterion used to separate the animals into Pressers and Non-Pressers. We have clarified this point in the Results section, which now reads: “We then separated the animals into two different groups based exclusively on whether the animals pressed the lever or not during the conflict phase and compared their behaviors during the entire test session (Figure 1K-R, Video 1).”

Is the distribution of "Pressers" and "Non-Pressers" consistent across experiments, as it seems to be skewed towards "Pressers" in the current data set.

No, the distribution of the two phenotypes was not consistent across the groups. Although we observed more *Pressers* in the experiments shown in Figure 1 (78% *Pressers* vs. 22% *Non-Pressers*), during the subsequent optogenetic experiments in Figure 7 and Supplementary Figure 11 we have observed less *Pressers* than Non-Pressers in the PL^GLUT^ photoinhibition groups (26% *Pressers* vs. 74% *Non-Pressers)* or similar proportions of *Pressers* and *Non-Pressers* in the PL^GABA^ photoactivation groups (51% *Pressers* vs. 49% *Non-Pressers*). Therefore, we cannot predict the number of *Pressers* and *Non-Pressers* before the test day.

In addition, while these phenotypes that emerge during conflict do appear to be independent of other behavior in the current study, additional analyses, like principal components analysis, would help support this claim by showing whether there is a relationship (e.g. clustering or reduction of variables) between certain behavioral responses and whether this relationship differs between the phenotypes. That is, such an analysis would allow one to better determine if this behavioral outcome measure is indeed independent of others.

We thank the reviewer for this suggestion. We went ahead and performed principal component analyses (PCA) using the percentage of time the animals spent approaching, avoiding, or freezing, as well as the latency to press the lever during the conflict phase. You can find in Author response images 4 and 4 the explained variance (%) for all four PCs together with the importance of each behavior for PC1 and PC2.

**Author response image 4. sa2fig4:** 

Note that the variables Latency and Approach are the ones that most contributed to PC1, but their relative contributions to the variance are relatively low (0.54 and 0.52). In contrast, for PC2, the variable Avoidance was the one that contributed more, but PC2 explained only ~15% of the observed variance. In conclusion, our analyses showing that Latency and Approach are the two variables that better explain the differences between *Pressers* and *Non-Pressers* reinforce our binary classification based on whether the animals pressed the lever or not during the conflict phase. We have now added the PCA analyses in the Result section (page 8).

Given the skewed sample sizes, it would be good to include effect sizes to further support the claims made based on these data.

We have now added an effect size calculation (Cohen’s or partial eta squared tests) for the comparisons between *Pressers* and *Non-Presses,* as recommended by the Reviewer. See the new analyses in the Source data 1. In addition, we added a description of the effect size calculation in the Methods section.

It is quite difficult to determine which statistical effects are being reported in the Figure Legends and how they correspond to those presented in Source data 1 and the claims made in the text. It would be beneficial to put supporting statistics following claims made in the primary text.

We apologize for not including all the statistical analyses in the Results section. We limited the statistical analyses to the figure legends and Source data 1 as an attempt to make the text cleaner for the readers. However, we recognize that not having the statistical analyses in the Results section may make the assessment of the comparisons difficult. We have now included all the statistical analyses to the Results section, added new comparisons between the two phenotypes, and revised all the previous analyses to make sure that they are congruent with the text description.

In relation to the point above, justification is needed for the Wilcoxon Test. While it seems to be appropriate for some datasets, for others it is questionable. For example, for the data presented in Figure 7, it would be important to report interactions between control and eNpHR groups for laser on vs. laser off conditions.

We have performed a normality test (Shapiro-Wilk test) for all the behavioral tests. We have now included the normality test analyses in the Results section and also described it in the Methods session. We observed that the results shown in Figure 7, as well as in other figures in the manuscript, don’t show a normal distribution. We therefore performed the appropriate non-parametric tests (Wilcoxon for paired samples and Mann-Whitney for non-paired samples) to compare the different groups and conditions. The statistical details are now better explained in the Methods section.

What is the justification for showing almost all behavioral data as percentages rather than raw values of time or other metrics?

Because the different phases of the test session (reward, odor, and conflict) have different durations, we have normalized the data in percentage to be able to compare the animals across the phases. We have clarified this point in the Methods section.

Figure 5 legend, panel O – Fisher's exact test for the Reward Phase seems to be lacking.

We apologize for the confusion that the legend has caused. The statistical analysis was presented in the figure legends, but the way we have presented it may have caused this confusion. The Fisher’s exact tests were performed by comparing reward phase vs. odor phase and subsequently odor phase vs. conflict phase. We have fixed the description in the figure legends and also added a description in the Results section. It now reads: “An average firing rate analysis across phases demonstrated that the activity of PL^GLUT^ neurons didn’t change significantly from the reward to the odor phase (Fisher Exact Test, all p’s > 0.05), but was inhibited from the odor to the conflict phase when Pressers resumed searching for food (Figure 5O, Fisher Exact Test, odor vs conflict, p = 0.0046)”.

Reviewer #3:Fernandez-Leon et al. investigate the role of the pre-limbic area (PL) in regulating approach avoidance behavior in situations of learned motivational conflict where animals experience both cues that predict an aversive outcome as well as cues that signal the availability of food. This region has been implicated in threat responding and food seeking separately but has not previously been examined in situations of conflict. The authors employ an individual differences approach, subdividing animals based on their food seeking behavior in the presence of conflicting cues that signal food availability and footshock and use a combination of in vivo recordings and optogenetic manipulations to identify a role for specific cell types in the PL in regulating risky behaviors in aversive contexts. This manuscript adds to the growing literature on neural mechanisms of processing approach-avoidance conflict.This work has many strengths. Examining approach and avoidance in a conflict paradigm, rather than separately, provides a more ethological study of the neural basis of these behaviors as, beyond the confines of a laboratory, action selection commonly occurs in the face of multiple competing cues. Subdividing animals into 'pressers' and 'non-pressers' based on individual differences in engagement in food seeking behavior is an excellent strategy to gain insight into the behavioral function of these cells. Recognizing that not engaging in food seeking does not necessarily reflect failure to complete the task but rather a bias toward avoidance behavior is insightful and important. The authors suggest a number of interesting and potentially important differences in PL neural activity between pressers and non-pressers. For example, pressers (i.e. rats that continue to seek food in the presence of an aversive cue) have both more food-cue responsive neurons and greater magnitude of excitatory and inhibitory responses to food-cues, a difference that is sustained when food-cues are presented in the presence of an aversive cue. Pressers and non-pressers also had marked differences in oscillatory frequency, an intriguing finding that warrants further investigation. Optogenetic experiments nicely establish causality with precise temporal resolution.

We are glad the reviewer recognizes the strengths of our work. Thank you!

The design of the behavioral paradigm somewhat limits the ability the ability to draw certain conclusions. During testing, food-cues were presented discretely while the shock-cue was constant preventing direct comparison of responding to appetitive and aversive cues that would have been highly interesting. Furthermore, during the test session, reward cues are always presented first followed by the addition of the shock-cue. This, and the extended shock-cue presentation under extinction conditions makes it difficult to entirely rule out alternative interpretations for differences between pressers and non-pressers, for example, more rapid extinction of fear memory in pressers than non-pressers. Beyond this, the lack of direct statistical comparison of neural activity in pressers and non-pressers undermines the strength of the central conclusions of this paper.The authors hypothesize that stimulating glutamatergic PL neurons decreases signal to noise ratio between cells that are active during food seeking and those that are not, thus resulting in a decrease in food-seeking. This is interesting and plausible proposal to be further explored in future research.

We thank the reviewer for the thoughtful comments. Below we provide a detailed explanation about how we have addressed each one of these points.

I have a number of concerns about the behavioral paradigm. Although both food and shock cues were presented as discrete 30s cues during conditioning, in the test phase, the shock associated cue was presented as a constant cue, first alone for 10 minutes and then with discrete 30s food associated cues overlayed. No rationale is provided for this specific design. I struggle to think of benefits for such a design compared to all discrete cue presentations, and see clear disadvantages. For example, it is not possible to compare responding to food and shock cues and the sustained exposure to the shock cue may result in extinction.

We want to thank the reviewer for raising this important point. We recognize that we have not been very clear explaining the rationale for the use of a constant odor in the previous version of the manuscript, but we have now clarified it in the Methods section. The main reason why we have decided to proceed with a constant odor presentation during our test protocol is the limitation to effectively introduce (and extract) the odor into (from) the chamber with accurate temporal precision. Although we have used a TTL-controlled olfactometer to insert the odor into the chamber and an exhaustor fan to remove it, after performing a series of pilot experiments we have learned that using an odor as a discrete cue in freely behaving animals is extremely challenging. By using an odor dispersion sensor (200B miniPID, Aurora Scientific) to quantify the amount of odor particles in the chamber right after the onset of the odor cues, we found that the amyl acetate odor takes approximately 2.21 ± 0.28 s to reach detectable concentrations in the arena (56 particles per billion; Punter, 1983). Our pilot tests also revealed that the delay and the variability to detect the odor after the olfactometer activation were even larger when we looked at the animals’ behavior (e.g., sniffing movements) because the distance of the animals from the odor port varied across the trials. Using the same odor dispersion sensor, we observed a large delay for the odor particles to be completely removed from the chamber after the offset of the olfactometer and the concomitant activation of the extractor fan (19.59 ± 0.97 s). This lack of temporal precision made it impossible for us to use the olfactory stimulus as a discrete cue during the conflict test, particularly because the single-unit recording experiments require high temporal resolution. Therefore, we have decided to use the odor as a constant cue, similar to an approach-avoidance conflict test with predator odor cues that our group has recently published (Engelke et al., 2021). Using a similar protocol with a constant odor presentation will allow us to perform future comparison between learned and innate defensive responses during the conflict test. It is important to note that, despite the lack of fine temporal resolution during the odor delivery, the presentation of 30 s odor cues during the acquisition phase resulted in significant expression of defensive behaviors during the odor test phase in the following day. The lack of freezing responses before the first food cue and odor presentation observed in the two groups (see Supplementary Figure 1I) suggests that the animals did not show signals of threat memory generalization to the context, as clarified in the Results section now.

To address the reviewer’s concern about the possibility of extinction taking place during the constant odor presentation, we have performed a minute by minute analyses of the behavioral responses during the odor phase for both groups. Similar to what we have reported in Figure 1M, N, P, we observed that *Pressers*, compared to *Non-Pressers*, showed a lower percentage of time exhibiting freezing and avoidance responses and a greater percentage of time approaching the food area. These behavioral differences were already observed in the very beginning of the odor phase and remained constant from the beginning to the end of the session for both groups, indicating that the animals did not extinguish their defensive responses across the session (see new Supplementary Figure 1J-L). We have added a similar description in the Results section (Page 9).

Further, the cues are in different modalities and not counterbalanced across animals: the food cue is always a tone & light and the shock cue is always odor.

While an auditory or a visual cue can be easily controlled with high temporal precision, an olfactory cue cannot. Thus, due to the lack of temporal control over the odor presentation (see more details in the answer above), we opted for not counterbalancing the cue modalities because millisecond temporal precision is necessary for electrophysiological recordings in vivo.

The manuscript also does not provide definitive behavioral evidence to show learning of the shock-odor association. Although for food-seeking discrimination between cue on and cue off periods is shown, the same is not shown for fear conditioning. In the absence of an unconditioned stimulus (CS-), it would be good to at least show discrimination from the pre-cue period (or an equivalent time where there is no food seeking or shock cue present) when comparing pressers and non-pressers in order to confirm that freezing and defensive behaviors are specific to learned odor association and not a generalized fear response.

The odor discrimination question is another important point that we would like to clarify to the reviewer. Using previous odor threat conditioning studies as a reference, we have performed a series of pilot tests in our lab with the intention of finding the best training protocol for the animals to discriminate a non-shock paired odor (CS-) from a shock-paired odor (CS+) during the conditioning phase. Despite trying different intertrial intervals, shock intensities, and rat strains, we failed to validate a protocol in which the animals would show defensive responses exclusively to the CS+ presentation during the acquisition phase (see an example in Author response image 5 using amyl acetate as CS+ and peppermint as CS-, intertrial intervals of ~5 min, 30s cue duration, 0.5 mA shock intensity, 1s duration). We therefore concluded that, at least in our hands, rats cannot discriminate well the CS+ from the CS- during the conditioning session.

**Author response image 5. sa2fig5:** 

Following the reviewer’s recommendation, we have now quantified the amount of freezing during the pre-odor presentation during the conditioning and test phases. It turns out that both groups show high freezing levels during the pre-odor trials of the conditioning phase, but very low freezing levels before the first food cue and odor presentation during the test session on the next day. We have added two new panels in Supplementary Figure 1 and a description of the findings in the Results section, which now reads: *“*Although Non-Pressers exhibited higher freezing levels during pre-odor trials 3 and 4 of the threat conditioning phase (Supplementary Figure 1H, F(5, 250) = 3.038, p = 0.011, Bonferroni's post-hoc p < 0.05), freezing responses before the first food cue and odor presentation were the same during the test day, indicating similar contextual discrimination between the two groups (Supplementary Figure 1I, Shapiro Wilk normality test, p < 0.05, Mann-Whitney test, U = 248, p = 0.113).”

This is especially important given that lever pressing during food cues during the threat conditioning day is suppressed, indicating that the animals show a generalized increase in freezing.

Please notice that a suppression in lever presses during the conditioning phase does not necessarily mean fear generalization. Lever press suppression has been also reported in several previous studies using discrete visual or auditory cues. However, this suppression has been attributed to conditioned inhibition in response to the unconditioned stimulus (i.e., footshocks), rather than threat generalization, because pre- CS lever presses (i.e., context) were not altered during the test session in the following day (Rescorla, 1968; Quirk et al., 2000; Bouton et al., 2008; McDannald and Galarce, 2011; Sierra-Mercado et al., 2011). Accordingly, a behavioral analysis of our results during the reward phase in the next day revealed that rats pressed the lever in >95% of the trials (Figure 1J), with response latencies during the reward phase (Figure 1I, 5.98 ± 0.73s) being similar to those observed during the pre-conditioning trials (Figure 1D habituation trial, 6.77 ± 1.20; Paired Student’s t-test, p = 0.579).

Given the sustained exposure to the shock cue under extinction conditions during test, a potential interpretation of the difference between pressers and non-pressers is that pressers may extinguish faster than non-pressers leading to behavioral differences in the odor phase. Enhanced extinction could reflect differences in underlying learning that are occluded on the conditioning day by high levels of freezing or an alteration in the mechanisms of extinction. It is notable that pressers and non-pressers do show differences in freezing on the test day. It would be good to look at freezing and pressing cue by cue as is done in Supplemental Figure 1C for the threat conditioning.

As reported above, we have performed a minute by minute analyses of the behavioral responses during the odor phase for both groups. Similar to what we have demonstrated in Figure 1M, N, P, *Pressers* compared to *Non-Pressers* showed a lower percentage of time exhibiting freezing and avoidance responses and a greater percentage of time approaching the food area. These behavioral differences were observed since the very beginning of the odor phase and the defensive responses remained constant from the beginning to the end of the session for both groups, indicating that the animals did not extinguish their defensive responses across the session (see new Supplementary Figure 1J-L). We have now added a similar explanation in the Results section (Page 9).

Wherever neural activity is compared between pressers and non-pressers, statistical testing should be conducted and reported. For example, the section starting on line 183 includes many statements comparing pressers and non-pressers with no statistical analysis to support this and Figure 1 only presents analyses within each group. If the authors wish to directly compare pressers and non-pressers, a more appropriate analysis (e.g. a two-way ANOVA) should be used.

The requested comparison between *Pressers* and *Non-Pressers* in Figure 1 were performed during the first version of the manuscript and represented by the symbol $, as described in the figure legends. We apologize for not including all the statistical analyses in the Results section. We limited the statistical analyses to the figure legends and Source data 1 as an attempt to make the text cleaner for the readers. However, we recognize that not having the statistical analyses in the Results section may make the assessment of the comparisons difficult. We have now moved all the statistical analyses to the Results section, added new comparisons between the two phenotypes, and revised all the previous analyses to make sure that they are congruent with the text description.

To determine the most appropriate statistical analyses, we have performed a normality test (Shapiro-Wilk’s test) for all the behavioral experiments. We have added more information to the Methods section to clarify this point. The new paragraph now reads: “Shapiro-Wilk normality test was performed to determine parametrical or non-parametrical statistical tests. For normal data, statistical significance was determined with paired or unpaired Student’s t test, repeated-measures ANOVA followed by Bonferroni post-hoc comparisons (Prism 7), and Z-test or Fisher’s exact test, as indicated in Source data 1S. For non-parametric data, Wilcoxon test (paired groups) or Mann- Whitney test (unpaired groups) were performed. Sample size was based on estimations by power analysis with a level of significance of 0.05 and an effect size of 0.8.”

Can the authors explain why they chose to stimulate PGLUT neurons in shock naïve animals? The paper is looking to show the involvement of these cells in approach avoidance in situations of conflict, but this design simply assesses food seeking.

The rationale for this experiment is based on the observation that *Pressers* showed an inhibition of spontaneous activity in PL^GLUT^ neurons from the odor to the conflict phase. We then hypothesized that reverting this inhibition by increasing the firing rate of PL^GLUT^ neurons would reduce lever presses. However, before testing the effects of PL^GLUT^ photoactivation during the conflict test, we first wanted to make sure that the stimulation had no effect on cued food seeking per se. Otherwise, it would be impossible to interpret the results during the conflict test. Surprisingly, we observed that stimulating PL^GLUT^ neurons in a neutral context was already sufficient to reduce cued food seeking. We therefore aborted the PL^GLUT^ photoactivation experiment during the conflict test and focused on inhibiting the same cells in *Non-Pressers* during conflict as an attempt to increase reward-seeking responses.

Starting on line 578 the authors note that activation of PGLUT neurons decreases food seeking responses despite Pressers showing increased excitatory food cue responses. They hypothesize that optogenetic activation of PGLUT neurons also activates cells not active during food seeking, decreasing signal to noise during the cue and therefore decreases food seeking behavior (FIGURE 8). Although a reasonable hypothesis, there is no direct evidence provided to support this. This claim could be strengthened by showing the magnitude of excitatory and inhibitory responses in PGLUT neurons specifically rather than across PL neurons generally like in Figure 2 E/F and O/P.

We appreciate the reviewer’s recommendation, but due to the reduced number of PL^GLUT^ neurons showing food cue responses, we could not perform the same calculation as in Figure 2E/F and O/P. We have now tuned down our interpretation in the discussion session by clarifying that this is a speculation.

“Thus, it is possible that increased activity in the firing rate of PL^GLUT^ neurons may result in reduced signal-to-noise ratio during the food cue onset (Kroener et al., 2009; McGinley et al., 2015), and consequently decreased food-seeking responses. In contrast, we speculate that by reducing their spontaneous firing rates during conflict situations, PL^GLUT^ neurons become more likely to fire in response to food cues due to an increase in the signal-to-noise ratio, thereby resulting in persistent reward-seeking responses during the conflict phase as we propose in our schematic in Figure 8.”

The title of Figure 5 is misleading in its exclusive focus on the conflict phase. In fact, this is true for the reward only period also. This should be amended to more accurately convey the data presented in the figure. The associated text in the Results section (e.g. line 418-420) also presents the interpretation that increased firing of PL^GLUT^ neurons associates with increased reward seeking during conflict. While strictly speaking this is not incorrect, not also discussing that the same effect is observed in the reward only period misconstrues this data by potentially suggesting an effect specific to conflict when in fact it would appear to be a general phenomenon reflecting reward seeking. The appropriate interpretation should be clarified throughout the manuscript.

We recognize that this was not the most accurate way to report the results in the first version of the manuscript. However, after applying the new photoidentification criteria suggested by Reviewer 1, the PL^GLUT^ photoidentified cells have changed and the significant effect that was initially observed when comparing reward and odor phases has now disappeared (please see updated Figure 5O). Because the effect is still present when comparing the odor and conflict phases, we have kept the same description we had initially.

In general, more nuanced analysis of the temporal dynamics of neural responses relative to behavior would be of interest and could more convincingly establish the link between neural activity and behavior.

We have followed the reviewer’s recommendation and performed some additional analyses of neuronal activity relative to rats’ behavioral responses during the test day. We aligned the firing rate of the cells to the onset of freezing, avoidance, and risk-assessment responses and selected those neurons that significantly responded 600 ms before the onset of each behavior (see new Supplementary Figure 6). The new sentence in the Results section now reads: “Moreover, a smaller fraction of PL neurons changed their firing rates 600 ms before the onset of either freezing, avoidance or risk assessment responses in both Pressers and Non-Pressers (Supplementary Figure 5A-M), indicating that some PL neurons can anticipate animal’s behavior during the test.”

[Editors’ note: what follows is the authors’ response to the second round of review.]

Essential revisions:1) Regarding the neural analyses comment #3 by Reviewer 1, the question pertained to whether the number of neurons obtained for a given population differed from chance. This is not a question about whether neural modulation was different from baseline for the neurons. It's generally answered with a chi squared test of independence. Is the population of neurons obtained one that is different from a population that could be obtained by chance?

We thank the reviewer for the recommendation. To verify whether PL neurons responsive to food cues, as well as PL neurons responsive to other events, change their firing rates specifically to the onset of the audio-visual cues (and not by chance at any random timepoint), we performed an additional analysis to quantify the proportion of cells that were either excited or inhibited 1 second before the food cue onset using the exact same Z-score criteria performed for other analyses in the manuscript. We observed that <2% of the 237 neurons recorded from *Pressers* and <3% of the 89 neurons recorded from *Non-Pressers* changed their firing rates when aligned to 1 second before the food cue onset. Notably, the percentage of responsive neurons increased significantly to 33% in *Pressers* and 21% in *Non-Pressers* when the same cells were aligned to the onset of the food cues, indicating that the proportion of food-cue responsive neurons in both groups was different from the proportion of responsive cells obtained by change at a random timepoint (Fisher Exact Test, all p’s < 0.0001). Similarly, the proportion of PL neurons that responded to lever presses (23%) and dish entries (16%) was also significantly different from that obtained by chance (<3%, all p’s < 0.0001).

Another possible way to test whether the number of PL responsive neurons differ from chance would be comparing the percentage of cells that showed excitatory, inhibitory, or no chances in response to the food cues with the percentage that could be obtained by chance (33%), assuming that excitatory, inhibitory, or no changes are equally likely to occur. However, this method may not be very informative because a series of previous studies have demonstrated that the prelimbic cortex is a very heterogenous region and only a small fraction (~30%) of PL neurons responds to conditioned cues. We therefore used the first method to address this point and added a new sentence in the Results section to report the new comparison. The new sentence reads:

“An analysis of PL activity at a random timepoint (e.g., 1 second before the food cue onset) resulted in less than 3% of responsive cells, indicating that the proportion of PL neurons that responded to food cues, lever presses, and food dish entries was significantly different from the proportion obtained by chance (Fisher Exact Test, all p’s < 0.05).”

2) Some of the most interesting, innovative, and impactful aspects of this work is the behavioral characterization based on individual differences. For this reason, the authors should consider reframing the Discussion, starting with a description of the behavioral phenotypes and what they might be capturing before describing the neuronal findings.

We have followed the reviewer’s recommendation and fixed the beginning of the discussion session to highlight the individual differences in behavior. The updated first paragraph of the discussion session now reads:

“Using a novel approach-avoidance conflict test, we identified two distinct behavioral phenotypes during the combined presentation of reward- and threat-paired cues: (i) rats that continued to press a lever for food (Pressers), and (ii) rats that exhibited a complete suppression in food-seeking responses (Non-Pressers). Single-unit recordings revealed that PL neurons regulate reward-approach vs. threat-avoidance responses during situations of uncertainty, when rats use previously associated memories to guide their decisions. We observed that increased risk-taking behavior in Pressers was associated with a larger number of food-cue responses in PL neurons, which showed sustained excitatory activity that persisted during the conflict phase, when compared to Non-Pressers. In addition, PL^GLUT^ neurons showed reduced spontaneous activity during risky reward seeking and photoactivation of these cells in a neutral context was sufficient to suppress lever press responses. Accordingly, photoinhibition of PL^GLUT^ neurons at the onset of the food cues in Non-Pressers reduced defensive responses and increased food approaching during the conflict phase, consistent with our observation that a small fraction of PL neurons changed their activity at the onset of freezing, avoidance or risk-assessment responses. Altogether, these results suggest that under memory-based conflict situations, reduced or increased activity in PL^GLUT^ neurons can favor the behavioral expression of food approaching or threat-avoidance responses, respectively.”

3) Pg. 10; lines 243-251: The authors state that Pressers showed a higher number of food-cue responsive neurons than Non-Pressers during the reward phase, and that both Pressers and Non-Pressers showed a significant reduction in the number of food-cue responsive neurons during the conflict phase. In the latter case, it is unclear if the reduction from 33% to 14% in Pressers is significantly different from the reduction of 21% to 6% in Non-Pressers (i.e. is there an interaction with phenotype in this reduction). It seems that this would be the case and that it would be important to recognize, but it is written as though both show an equivalent reduction in the number of food-cue responsive neurons during conflict. Direct phenotype comparisons appear to be lacking in subsequent analyses of the same dataset, which limits the conclusions that can be drawn. In this regard, additional clarity is needed regarding what comparisons are being made and reported and what the authors can or cannot conclude based on these comparisons.

We have performed a group comparison analysis and found that *Pressers* and *Non-Pressers* exhibited similar reductions in the number of responsive neurons from the reward to the conflict phases. We have added the new analysis in the Results section, which now reads:

“In addition, the percentage of reduction in the number of responsive cells across the phases was similar between Pressers and Non-Pressers (Figure 2C vs. Figure 2M, Fisher Exact Test, 47 out of 232 neurons for Pressers, 13 out of 89 neurons for Non-Pressers, p = 0.427), suggesting that PL neurons can distinguish between reward and conflict situations (Figure 2G vs. 2H and 2Q vs. 2R).”

Another example comes from the data described for the inset in Figure 2I and 2S, for which it is stated that "Pressers show a higher magnitude of inhibitory food-cue responses during the reward phase, and, in contrast to Non-Pressers, such responses were attenuated during the conflict phase". However, it is not clear from the statistics included or the text that the authors are directly comparing Pressers and Non-Pressers and/or which statistics speak to this comparison.

Thanks for raising this point. We have clarified the text now by indicating that a direct statistical comparison was made between *Pressers* and *Non-Pressers.* The new sentence now reads:

“In addition, Pressers showed a higher magnitude of inhibitory food-cue responses during the reward phase when compared to Non-Pressers (Figure 2I blue bar inset vs. 2S blue bar inset, Shapiro-Wilk normality test, p < 0.001, Mann-Whitney test, U = 50, p = 0.0045).”

4) Clarification points:i) Pg. 11, line 270: The authors should replace "the same number" with "the same percentage".

We have incorporated the suggestion. Thank you!

This also raises the concern as to whether or not it is appropriate to compare percentages when the denominator is 237 in Pressers and 89 in Non-Pressers. That is, how does recording from 237 neurons vs. 89 neurons affect the interpretation of the results given that the likelihood of detecting differences is presumably greater with 237 neurons.

For all the between-group comparisons performed in the manuscript, we have used the most appropriate statistical analyses for samples of unequal size and variance. Because unpaired Student’s t test comparison has been criticized by some authors, we have now replaced it by Welch’s t test, which is more recommended for group comparisons of unequal variance. The final results in Supp. Figure 1F-G, where it has been used before, remained the same after the modification.

ii) Pg. 12, lines 293-297: Similar to the point above, is the Fisher's Exact Test reported here only comparing the proportions of neurons between phases and not between behavioral phenotypes?

We have performed both comparisons. In addition to describing the statistical differences in Source data 1, we have now added the statistical results to the text and figure legend of Supp. Figure 5.

iii) Figure 6M, 6N. It looks as though the significant effects of laser diminish over time for the CaMKII-ChR2 group. Can the authors comment on this in the manuscript?

Thanks for raising this point. We have included a new sentence in the manuscript to provide some potential explanations for the diminished effect of laser illumination observed by the end of the session. The text now reads:

“The diminished behavioral effect observed during the third block of laser on could be the result of conformational changes in the opsin (e.g. photobleaching) or temporary depletion of synaptic vesicles following repeated laser illumination, as previously reported (Kittelmann et al., 2013; Stahlberg et al., 2019).”

iv) A description of the principal components analysis used to assess the behavioral data seems to be lacking in the primary text.

We added a brief explanation of the principal component analysis in the Methods section together with a reference about the topic. The new paragraph now reads:

“Principal component analyses on behaviors. The principal component analysis (PCA) method (Jolliffe, 2002) was implemented to further understand the relationships between approaching, avoidance, freezing, and latency to press during the conflict phase for both Pressers and NonPressers. This method was used to reduce the dimensionality of our multivariate data whilst preserving as much of the relevant information as possible. Briefly, we performed a linear transformation of the data to a new coordinate system such that the new set of variables, the principal components, were linear functions of the original variables. These variables were also uncorrelated, and the greatest variance by any projection of the data came to lie on the first coordinate, the second greatest variance on the second coordinate, and so on. Afterwards, we analyzed the explained variance (%) of the first two PCs (i.e. PC1 and PC2) together with the relevance of each analyzed behavior to define which variables better explained the differences between Pressers and Non-Pressers (see Results section for additional details).”

5) Additional analyses:i) Please include insets of the AUC analyses reported in the text, this applies for 4B, C, D and E.

We have created new inset graphics for all the AUC quantifications and performed the appropriate statistical comparisons, which were reported in the text and in details in Source data 1. Please see the new inset graphics in the updated Figure 4.

ii) Pg. 17, lines 405-412: It is stated that proportions of excitatory and inhibitory food cue response for PL(Glut) and PL(GABA) neurons were similar during the reward and conflict phase (Figure 5J-K). However, it does not appear that an analysis was conducted to compare proportions between different neuronal subtypes. That is, it seems like it would be meaningful to be able to say that the proportion of excited glutamatergic neurons in response to the food cue during the conflict phase was less than the proportion of GABAergic neurons excited during the conflict phase (should that prove to be the case). Again, as written, it is difficult to discern what analyses were or were not conducted and which are being described.

We have performed all the possible comparisons (within and between phases) and have not observed any significant differences in the proportion of PL^GABA^ vs PL^GLUT^ neurons that were responsive to food cues. The p values for each one of the eight comparisons can be found now in Source data 1. Following the reviewer’s recommendation, we have now rephrased the sentence to better reflect the performed comparisons. The new text now reads:

“We observed that the proportions of excitatory and inhibitory food cue responses for PL^GLUT^ and PL^GABA^ neurons were similar when comparing between the reward and the conflict phases as well as within each one of the phases (Figure 5J-K, Fisher Exact Test, all p’s > 0.05, see Source data 1).”

Please also include a Key Resources Table.

A Key Resources Table was included in the submission.